*Resource*

# MemPrep, a new technology for isolating organellar membranes provides fingerprints of lipid bilayer stress

John Reinhard [1,2], Leonhard Starke[3], Christian Klose[4], Per Haberkant[5], Henrik Hammarén [6], Frank Stein [5], Ofir Klein [7], Charlotte Berhorst[1,2], Heike Stumpf[1,2], James P Sáenz [8], Jochen Hub [3], Maya Schuldiner [7] & Robert Ernst [1,2 ✉]

## Abstract

**Biological membranes have a stunning ability to adapt their composition in response to physiological stress and metabolic challenges. Little is known how such perturbations affect individual organelles in eukaryotic cells. Pioneering work has provided insights into the subcellular distribution of lipids in the yeast *Saccharomyces cerevisiae*, but the composition of the endoplasmic reticulum (ER) membrane, which also crucially regulates lipid metabolism and the unfolded protein response, remains insufficiently characterized. Here, we describe a method for purifying organelle membranes from yeast, MemPrep. We demonstrate the purity of our ER membrane preparations by proteomics, and document the general utility of MemPrep by isolating vacuolar membranes. Quantitative lipidomics establishes the lipid composition of the ER and the vacuolar membrane. Our findings provide a baseline for studying membrane protein biogenesis and have important implications for understanding the role of lipids in regulating the unfolded protein response (UPR). The combined preparative and analytical MemPrep approach uncovers dynamic remodeling of ER membranes in stressed cells and establishes distinct molecular fingerprints of lipid bilayer stress.**

**Keywords** ER Stress; UPR; Lipid Bilayer Stress; MemPrep; Organelle Lipidomics
**Subject Categories** Membranes & Trafficking; Organelles

## Introduction

Biological membranes are complex assemblies of lipids and proteins. Their compositions and properties are dynamically regulated in response to stress and as well as more subtle physical and metabolic cues (Harayama and Riezman, 2018; Ernst et al, 2018). A prominent example is the homeoviscous adaptation, where the lipid composition is adapted to temperature for maintaining membrane fluidity and membrane phase behavior (Sinensky, 1980; Ernst et al, 2016; Harayama and Riezman, 2018). Even mammals, which maintain a constant body temperature, can readily adjust their membrane composition in response to dietary perturbation with major impact on collective bilayer properties such as fluidity, thickness, surface charge or compressibility (Bigay and Antonny, 2012; Levental et al, 2020). Eukaryotic cells face the challenge of maintaining the properties of the plasma membrane and several coexisting organelle membranes each featuring characteristic lipid compositions as well as constantly exchanging membrane material with other organelles. Despite recent advances to manipulate and follow membrane properties (John Peter et al, 2022; Renne et al, 2022; Tsuchiya et al, 2023; preprint: Jiménez-Rojo et al, 2022), we know little about how stressed cells coordinate membrane adaptation between organelles whilst maintaining organelle identity and functions.

The endoplasmic reticulum (ER) spans eukaryotic cells as a continuous membrane network, including the nuclear envelope and the peripheral ER consisting of flat cisternae and narrow tubules (Phillips and Voeltz, 2016). It is a hotspot for lipid biosynthesis (Zinser et al, 1991; Henry et al, 2012) and provides an entry site for soluble and transmembrane proteins to the secretory pathway. The flux of proteins and lipids through the secretory pathway is controlled by the unfolded protein response (UPR) (Travers et al, 2000; Walter and Ron, 2011). When the protein folding capacity of the ER is overwhelmed, the type I membrane protein Ire1 responds to an increase in protein misfolding by multimerizing thereby facilitating the association and activation of its cytosolic kinase/RNase domains (Walter and Ron, 2011). The cleavage of the *HAC1* mRNA in *Saccharomyces cerevisiae* (from here on "yeast") by the activated RNase domain is the committed step for UPR activation and leads to the formation of the Hac1 transcription factor, which

[1]Saarland University, Medical Biochemistry and Molecular Biology, Homburg, Germany. [2]Saarland University, Preclinical Center for Molecular Signaling (PZMS), Homburg, Germany. [3]Saarland University, Theoretical Physics and Center for Biophysics, Saarbrücken, Germany. [4]Lipotype GmbH, Dresden, Germany. [5]EMBL Heidelberg, Proteomics Core Facility, Heidelberg, Germany. [6]EMBL Heidelberg, Genome Biology, Heidelberg, Germany. [7]Weizmann Institute of Science, Department of Molecular Genetics, Rehovot, Israel. [8]Technische Universität Dresden, B CUBE, Dresden, Germany. ✉E-mail: robert.ernst@uks.eu

regulates hundreds of UPR target genes (Travers et al, 2000; Ho et al, 2020). Once activated, the UPR lowers global protein synthesis, whilst upregulating the ER-luminal folding machinery, ER-associated protein degradation, and lipid biosynthesis enzymes (Travers et al, 2000; Walter and Ron, 2011). Intriguingly, Ire1 can also sense ER membrane aberrancies referred to as lipid bilayer stress via a hydrophobic mismatch-based mechanism (Halbleib et al, 2017). This responsiveness of the UPR to lipid bilayer stress is evolutionarily conserved (Volmer et al, 2013; Hou et al, 2014; Ho et al, 2018; Pérez-Martí et al, 2022). A diverse set of lipid metabolic perturbations trigger the UPR: inositol depletion (Cox et al, 1997; Promlek et al, 2011; Halbleib et al, 2017), increased lipid saturation (Pineau et al, 2009; Surma et al, 2013), increased sterol levels (Feng et al, 2003; Pineau et al, 2009), misregulated sphingolipid metabolism (Han et al, 2010), and a disrupted conversion of phosphatidylethanolamine (PE) to phosphatidylcholine (PC) (Thibault et al, 2012; Ho et al, 2020; Ishiwata-Kimata et al, 2022). Even prolonged proteotoxic stresses, misfolded membrane proteins, and exhaustion of the culture medium have been associated with UPR activation via this membrane-based mechanism (Promlek et al, 2011; Tran et al, 2019b; Väth et al, 2021; Ishiwata-Kimata et al, 2022). Little is known how these conditions of lipid bilayer stress affect the molecular composition of the ER membrane.

Advances in quantitative lipidomics (Ejsing et al, 2009) have provided deep insights into the flexibility and adaptation of the cellular lipidome to various metabolic and physical stimuli in both yeast and mammals (Klose et al, 2012; Casanovas et al, 2015; Levental et al, 2020; Surma et al, 2021). However, unless these analytical platforms are paired with powerful techniques for isolating organellar membranes from stressed and unstressed cells, they lack the subcellular resolution, which is essential to understand how lipid metabolism is organized between organelles. Even though tremendous and pioneering efforts have been invested in characterizing organellar membranes from yeast (Zinser and Daum, 1995; Schneiter et al, 1999; Klemm et al, 2009; Surma et al, 2011; Reglinski et al, 2020), we still lack comprehensive and quantitative information on the ER membrane. This is probably due to the extensive membrane contact sites formed between the ER and other organelles, making isolation technically challenging (English and Voeltz, 2013; Scorrano et al, 2019).

Here, we describe a protocol for the isolation of highly enriched organellar membranes, MemPrep. We demonstrate its utility by the successful isolation of both ER and vacuolar membranes from yeast and provide a molecular toolkit to make this method applicable to other organelles. Using quantitative lipidomics, we reveal specific characteristics of these membranes. In the ER, we observe a high proportion of monounsaturated fatty acyl chains and a low level of ergosterol. The vacuole membrane, on the other hand, is virtually devoid of phosphatidic acid (PA). By analyzing the lipid composition of the stressed ER, we establish molecular fingerprints of lipid bilayer stress, provide evidence for a general contribution of low membrane compressibility to UPR activation, and identify a potential role of anionic lipids as negative regulators of the UPR. Despite these common denominators, our data demonstrate that lipid bilayer stress comes in different flavors provided by vastly distinct lipid and protein compositions of the ER membrane. The MemPrep approach sets the stage for a better understanding of the organelle-specific membrane adaptations to metabolic, proteotoxic, and physical stresses in the future.

# Results

## Creation of a rapid and clean approach for yeast organelle isolation, MemPrep

In the past, systematic organelle isolation from yeast has been carried out predominantly by differential sedimentation and density centrifugation (Zinser and Daum, 1995; Schneiter et al, 1999). Affinity purification methods that work well in mammalian cells cannot be translated easily into yeast work, especially when the organelle-of-interest forms extensive membrane contact sites (Takamori et al, 2006; Klemm et al, 2009; Abu-Remaileh et al, 2017; Ray et al, 2020; Fasimoye et al, 2023). We sought to create a versatile, yeast-specific affinity purification method, MemPrep, for obtaining highly enriched organelle membrane fractions. We reasoned that important aspects of MemPrep would be the capacity to bind organellar membranes with high specificity and the ability to release them selectively after intense washing. Hence, we constructed a tagging cassette that can equip an open reading frame in yeast with a sequence encoding for a C-terminal bait tag comprising a myc epitope, a recognition site for the human rhinovirus (HRV) 3C protease, and the 3xFLAG epitope (Fig. 1A). Following proof of concept of the validity of MemPrep (see below) and to enable our approach to be widely used by the yeast community regardless of which organelle is of interest, we created a systematic collection of strains in which every yeast protein is tagged with the bait sequence (see some examples for each organelle in Appendix Fig. S1A). To do this, we used the SWAp Tag (SWAT) approach (Yofe et al, 2016; Meurer et al, 2018; Weill et al, 2018) coupled with automated library creation strategies (Tong and Boone, 2006; Weill et al, 2018). The library or any individual strain is freely distributed.

## MemPrep yields highly enriched ER membrane vesicles

To showcase MemPrep, we initially focused on the largest organelle in the cell, the ER, which is a particularly challenging target. It forms physical contact sites with almost every other membrane-bound organelle (English and Voeltz, 2013), and previous attempts to isolate ER membranes suffered from significant mitochondrial contaminations (Schneiter et al, 1999; Reglinski et al, 2020). An ideal bait protein should be a highly abundant transmembrane protein, feature an accessible C-terminus, and localize exclusively to a single organelle. Initially, we used Rtn1 as a bait, which is a structural determinant of the tubular ER. Rtn1 is a small (~33 kDa) and highly abundant reticulon protein (~37,100 copies per cell), which stabilizes membrane curvature in the tubular ER (Ghaemmaghami et al, 2003; Voeltz et al, 2006). Rtn1 has four predicted transmembrane helices with both N- and C-terminus facing the cytosol and a C-terminal amphipathic helix, which inserts into the cytosolic leaflet of the ER membrane to generate a high spontaneous membrane curvature (De Craene et al, 2006; Hu et al, 2008).

Several experimental factors are important to ensure the successful isolation of ER membranes (Fig. 1A): First, cells are mechanically disrupted, thereby minimizing potential artifacts from the ongoing lipid metabolism and ER stress that would occur during the enzymatic digestion of the cell wall under reducing conditions (Zinser and Daum, 1995). Second, after a differential

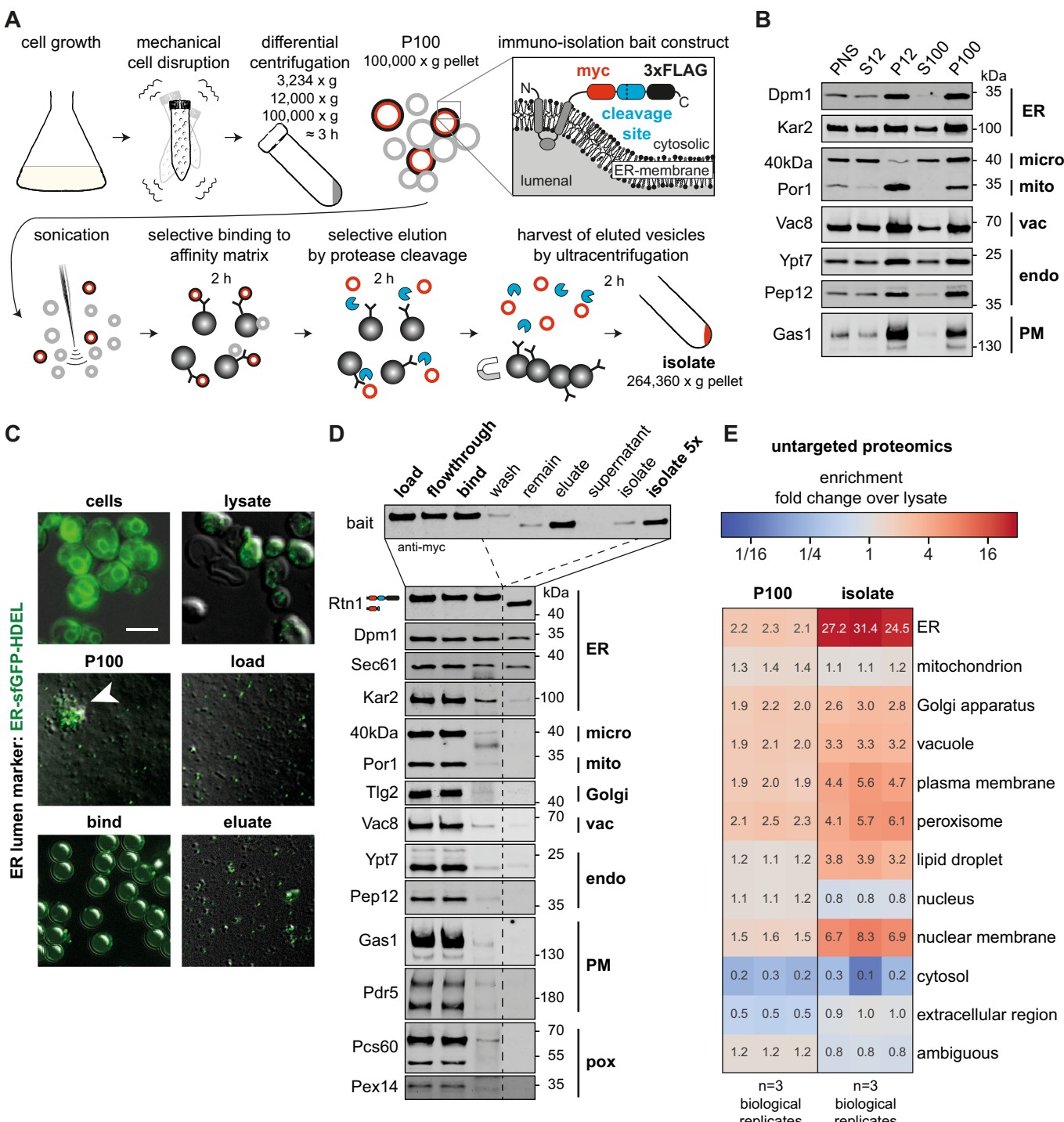

centrifugation to obtain a crude microsome fraction, organellar fragments were disrupted by sonication. This is because a major fraction of the ER membrane surface forms physical contacts to other organelles (Phillips and Voeltz, 2016), which would hamper the subsequent ER isolation. We reasoned that large ER fragments are more likely to contain such contact sites, while enough small ER fragments might facilitate higher purity if membrane mixing can be avoided. Third, we employ harsh washing conditions with urea-containing buffers to remove peripherally attached proteins and to

destabilize membrane contact sites. Fourth, the isolated membrane vesicles are selectively eluted from the affinity matrix thereby providing a straightforward coupling to various mass spectrometry-based analytical platforms following previous paradigms (Klemm et al, 2009).

Enrichments of organellar membranes relies first on differential centrifugation and only then on affinity purification. To decide on the exact fraction best to utilize for membrane pulldowns, we performed immunoblotting experiments after differential

**Figure 1.   Immunoisolation of the ER via MemPrep.**

(A) Schematic representation of the immunoisolation protocol. Cells are cultivated in SCD$_{complete}$ medium and mechanically disrupted by vigorous shaking with zirconia/glass beads. Differential centrifugation at 3234 × $g$, 12,000 × $g$, and 100,000 × $g$ yields crude microsomes in the P100 fraction originating from different organelles. The immunoisolation bait tag installed at the C-terminal end of Rtn1 is depicted in the inlay (myc-tag, human rhinovirus (HRV) 3C protease cleavage site, 3xFLAG tag). Sonication segregates clustered vesicles and decreases the vesicle size. ER-derived vesicles are specifically captured by anti-FLAG antibodies bound to Protein G on magnetic beads. After rigorous washing, the ER-derived vesicles are selectively eluted by cleaving the bait tag with the HRV3C protease (blue sectors). The eluted ER-derived vesicles (red circles) are harvested and concentrated by ultracentrifugation. (B) Distribution of the indicated organellar markers in the fractions of a differential centrifugation procedure: Supernatant after 3234 × $g$ centrifugation (post nuclear supernatant, PNS), supernatant after 12,000 × $g$ centrifugation (S12), pellet after 12,000 × $g$ centrifugation (P12), supernatant after 100,000 × $g$ centrifugation (S100), pellet after 100,000 × $g$ centrifugation (P100). Dpm1 and Kar2 are ER markers, the 40 kDa protein (40 kDa) is a marker for light microsomes (Zinser et al, 1991), Por1 is a marker of the outer mitochondrial membrane, Vac8 is a vacuolar marker, Ypt7 and Pep12 mark endosomes, and Gas1 serves as plasma membrane marker. In total, 7.8 µg total protein loaded per lane. (C) Overlay of fluorescence micrographs and differential interference contrast images of cells expressing an ER-luminal marker (ER-sfGFP-HDEL) and fractions from immunoisolation. Intact cells (cells) show typical ER staining. Mechanical cell disruption leads to fragmentation and release of intracellular membranous organelles (lysate). The crude microsomal fraction (P100) contains aggregates of GFP-positive and GFP-negative vesicles (white arrowhead). Segregation by sonication yields more homogenous size distribution of vesicles (load). Individual ER-luminal marker containing vesicles are bound to the surface of much larger magnetic beads (bind). Selective elution by protease cleavage releases vesicles from the affinity matrix (eluate). Scale bar for all panels: 5 µm. (D) Immunoblot analysis of immunoisolation fractions for common organellar markers (ER endoplasmic reticulum, micro light microsomal fraction, mito mitochondria, Golgi Golgi apparatus, vac vacuole, endo endosomal system, PM plasma membrane, pox peroxisomes). Overall, 0.2% of each fraction loaded. (E) Untargeted protein mass spectrometry analysis showing enrichment of P100 and isolate fractions over whole-cell lysate. The enrichment of proteins over the cell lysate (fold change) is based on uniquely annotated subcellular locations and provided for each biological replicate. The illustrated numbers represent the median enrichment for unique annotated genes from $n = 3$ biological replicates. Source data are available online for this figure.

centrifugation (Fig. 1B). Membrane markers for the ER (Dpm1), mitochondria (Por1), endosomes (Ypt7, Pep12), the vacuole (Vac8), and the plasma membrane (Gas1) were all enriched in the pellets of a centrifugation at 12,000 × $g$ (P12) and 100,000 × $g$ (P100), while the light microsomal 40 kDa protein (40 kDa) was predominantly found in the P100 fraction (Fig. 1B). The marker for the outer mitochondrial membrane (Por1) was significantly enriched in P12 relative to P100 (Fig. 1B). To minimize contaminations from mitochondrial membranes we decided to use the crude microsomal P100 fraction for isolating ER membrane vesicles. By doing so, we discard a significant fraction of ER membrane with the P12 fraction (Fig. 1B). We cannot formally rule out the possibility that the discarded ER membrane vesicles in the P12 fraction have a different composition than the rest of the ER, which we subsequently isolate. The ER-luminal chaperone Kar2 was found both in the supernatant and the pellet after centrifugation at 100,000 × $g$ (S100). This suggests that a significant portion of ER-luminal content is released during cell disruption.

To ensure that our choice of P100 is optimal and to uncover the extent of loss of ER-luminal proteins, we followed the entire procedure from cell disruption to the elution of the isolated vesicles in a control experiment using fluorescence microscopy. To this end, we used cells expressing both an ER bait protein, but also an ER-targeted, superfolder-GFP variant equipped with a HDEL sequence for ER retrieval (Fig. 1C) (Lajoie et al, 2012). By following the fluorescent ER-luminal marker, we realized that the crude microsomal P100 fraction contains large clumps of GFP-positive and GFP-negative vesicles (Fig. 1C; P100, white arrowhead). Due to the loss of ER-luminal proteins, the GFP-negative vesicles could be derived from the ER but may also be from other organelles. We decided to reduce the size of the microsomes by sonication, which also separated larger clumps of aggregated vesicles (Fig. 1A,C).

Sonication transiently disrupts lipid bilayers and can theoretically induce lipid mixing or a transient fusion of adjacent lipid bilayers. Because this would obscure our measurement of the ER membrane composition, we performed control experiments to rule this out. We utilized small unilamellar vesicles of ~100 nm containing POPC, NBD-PE, and Rho-PE at a ratio of 98:1:1. The

two fluorescent lipid analogs form a Förster resonance energy transfer (FRET) pair (Appendix Fig. S1B). We sonicated these synthetic liposomes in the presence of a ~15.4-fold excess of microsomal membranes (P100) based on membrane phospholipid content. Because fusion between the synthetic liposomes and microsomal membranes would "dilute" the fluorescent lipid analogs, a decrease of the relative FRET efficiency would be expected upon membrane mixing or upon the exchange of individual fluorescent lipid molecules. However, the 10 cycles of sonication as used during MemPrep procedure for dissociating vesicle aggregates do not lead to a significant change of the FRET efficiency. Lower FRET efficiencies indicative for lipid exchange or membrane fusion, was only observed after 100 cycles of sonication, which also leads to sample warming, and upon incubation for 30 min at RT with either 18 mM methyl-β-cyclodextrin, which facilitates lipid exchange, or with 40% w/v PEG 8000, which supports membrane fusion (Appendix Fig. S1B) (Lentz, 1994; Cheng et al, 2009). Expectedly, we observed a dramatic drop of the relative FRET efficiency upon the addition of SDS, which dissolves both the liposomal and microsomal membranes (Appendix Fig. S1B). These data suggest that the sonication as used in the MemPrep procedure does not cause a significant degree of membrane mixing from fusion and/or lipid exchange.

After having optimized sample homogenization, we turned our attention to the immunoisolation procedure. We decided on magnetic dynabeads coated with Protein G and sparsely decorated with anti-FLAG antibodies. Notably, the low density of antibodies is required to lower avidity effects, which would impede the elution of membrane vesicles from the matrix. The capturing of GFP-positive, ER-derived vesicles to the affinity matrix was validated by fluorescence microscopy (Fig. 1C, bind). After extensive washing with 0.6 M urea-containing buffers, the isolated vesicles were eluted by cleaving the bait tag (Fig. 1C; eluate) as validated by immunoblotting using anti-myc antibodies (Fig. 1D; eluate). The isolated membrane material was harvested and concentrated by ultracentrifugation (264,360 × $g$, 2 h, 4 °C) (Fig. 1A). Immunoblotting demonstrated the co-purification of the bait (Rtn1) with other ER membrane proteins (Dpm1, Sec61) (Fig. 1D), while most of the

ER-luminal chaperone Kar2 is lost during the isolation. Remarkably, all markers for other organelles including the mitochondrial marker Por1 remained undetected in the final isolate (Fig. 1D). Even the light microsomal marker 40 kDa protein (40 kDa), whose subcellular localization is not fully established, is lost during the preparation (Zinser et al, 1991). Hence, MemPrep yields highly enriched ER membrane preparations. While previous approaches for isolating organelles such as the Lyso-IP were optimized for speed (Abu-Remaileh et al, 2017; Ray et al, 2020; Fasimoye et al, 2023), we established a technique that also allows for the elution of the isolated membrane vesicles. This approach does not only increase the purity of our preparation, it also provides a significant advantage for coupling the isolation to analytical platforms such as quantitative lipidomics, proteomics, or fluorescence spectroscopy. Overall, the MemPrep procedure provides high purity of organelle-derived membranes at the expense of low yields (83 mg protein in the cell lysate yields ~30 µg of protein in the isolate via the Rtn1-bait). Assuming the ER accounts for 20% of the total cell protein (Zinser and Daum, 1995), we estimate that >99.8% of ER protein is lost during the isolation.

## Quantitative proteomics validates the performance of MemPrep with distinct bait proteins

Demonstrating the loss of several specific organelle markers by immunoblotting is often used as a "gold standard" for validating organelle isolations, however, this validation is limited by the detection sensitivity of the antibody as well as the specific abundance of the organelle marker utilized. Hence, for a more rigorous validation, we measured the level of cleanliness of our preparations by TMT-multiplexed, untargeted protein mass spectrometry (Fig. 1E). We also compared the enrichment of the ER over the cell lysate with that of other organelles using a total of 1670 proteins uniquely annotated for cellular compartments with gene ontology terms (GO terms) (Fig. 1E, Source Data file: Fig. 1E_annotations). The mean enrichment of 213 ER-specific proteins was 27.7-fold in the immuno-isolate over the cell lysate, which is also consistent with semi-quantitative, immunoblotting data (Appendix Fig. S1C). This quantitative approach revealed an efficient depletion of cytosolic and nucleoplasmic proteins and a strong enrichment over mitochondrial proteins, which represented a major contaminant in microsome preparations in the past (Schneiter et al, 1999; Reglinski et al, 2020). A moderate enrichment of markers from other organelles of the endomembrane system (Golgi apparatus, vacuole, etc.) was found as expected since they pass through the ER on their route to their subcellular destination and because the efficient removal of soluble proteins alone causes an enrichment of organelle membrane markers. In line with the procedure that is intended to enrich for the membrane fraction, ER membrane proteins were substantially more enriched than ER-luminal, soluble proteins or proteins from other organelles (Appendix Fig. S1D). Only a few proteins annotated to other organelles are enriched >20-fold over the lysate (Appendix Fig. S1D) and for most of these there is evidence that they in fact localize to the ER or the nuclear envelope, which is continuous with the ER membrane. Hence, Osm1, Yur1, Ist2, Ygr026w, Pex30, Pex29, Slc1, Uip6, Brr6, and She10 were falsely annotated as non-ER proteins (Appendix Fig. S1D). A dual localization including the ER and another organelle has been reported for

Osm1, Yur1, Pex31, Slc1, Cst26, Svp26, Ept1, and Cbr1. Likewise, there is evidence for an ER localization for the non-annotated proteins Ybr096w, Gta1, Msc1, and Hlj1. This suggests that MemPrep and quantitative proteomics can even predict ER membrane localization.

The ER forms an extended membrane network composed of sheets and tubules. We assayed whether MemPrep is suitable to isolate ER subdomains by an appropriate choice of the bait or if ER membranes isolated by this approach are representative for the "entire" ER membrane? To address this question, we compared the isolates using the Rtn1-bait localizing to the tubular ER with those using the Elo3-bait, which is enriched in the nuclear ER membrane (Fig. EV1A). Quantitative proteomics reveals that fusing the bait tag to either Rtn1 or Elo3 has no impact on the overall cellular proteome (Fig. EV1B) and that ER proteins can be enriched by MemPrep using either of the two baits (Fig. EV1C,D) with an estimated yield of 0.19% and 0.08% of the input material for the Rtn1- and Elo3-bait, respectively. We included an additional carbonate wash of the P100 microsomes prior to the immunoisolation procedure to further decrease contaminations from soluble proteins and to potentially increase the coverage of membrane proteins in the subsequent proteomics experiment. A direct comparison of the proteomes from the two isolates identified stunningly few differences: While 3013 proteins were detected and analyzed, only 12 proteins showed a >twofold difference between Rtn1-bait and Elo3-bait-derived ER isolates (Fig. EV1E). This suggests that MemPrep yields preparations, which rather represent the "entire" ER membrane than a specific ER subdomain. Yet, we cannot exclude that the portion of the ER that is lost/discarded during the preparation may have a different composition. We speculate that the harsh mechanical disruption of the cell, which is required to break the cell wall, causes a fragmentation of the ER network that disrupts lateral specializations. While we know from our work with mammalian cells that ER subdomains can be isolated via MemPrep, we are convinced that preparations representative of the "entire" organelle membranes have practical advantages for studying inter-organelle transport processes by lowering the minimally required sample number for proteomics and lipidomics experiments as indicated by first applications using the MemPrep technology and variations thereof (Reinhard et al, 2023; preprint: Koch et al, 2023).

## The lipid composition of the ER

Previous attempts to establish the ER membrane lipid composition in yeast were hampered by mitochondrial contaminations (Schneiter et al, 1999; Reglinski et al, 2020). Having established the isolation of ER-derived membranes via the Rtn1-bait, we were interested in determining their lipid composition using quantitative lipidomics (Fig. 2A–C). Compared to whole-cell lysates, the ER membrane features (1) substantially lower levels of neutral storage lipids (ergosterol esters (EEs) and triacylglycerols (TAGs)), (2) significantly more diacylglycerol (DAG), PC and PE, but (3) less phosphatidylinositol (PI) lipids. Notably, the same lipid composition was observed for ER membranes isolated via the Elo3-bait (Fig. EV2A–C). Hence, the ER maintains a characteristic lipid composition even though it readily exchanges membrane material with other organelles (Wong et al, 2019). Remarkably, the level of ergosterol in the ER (9.7 mol%) is barely distinct from the level in

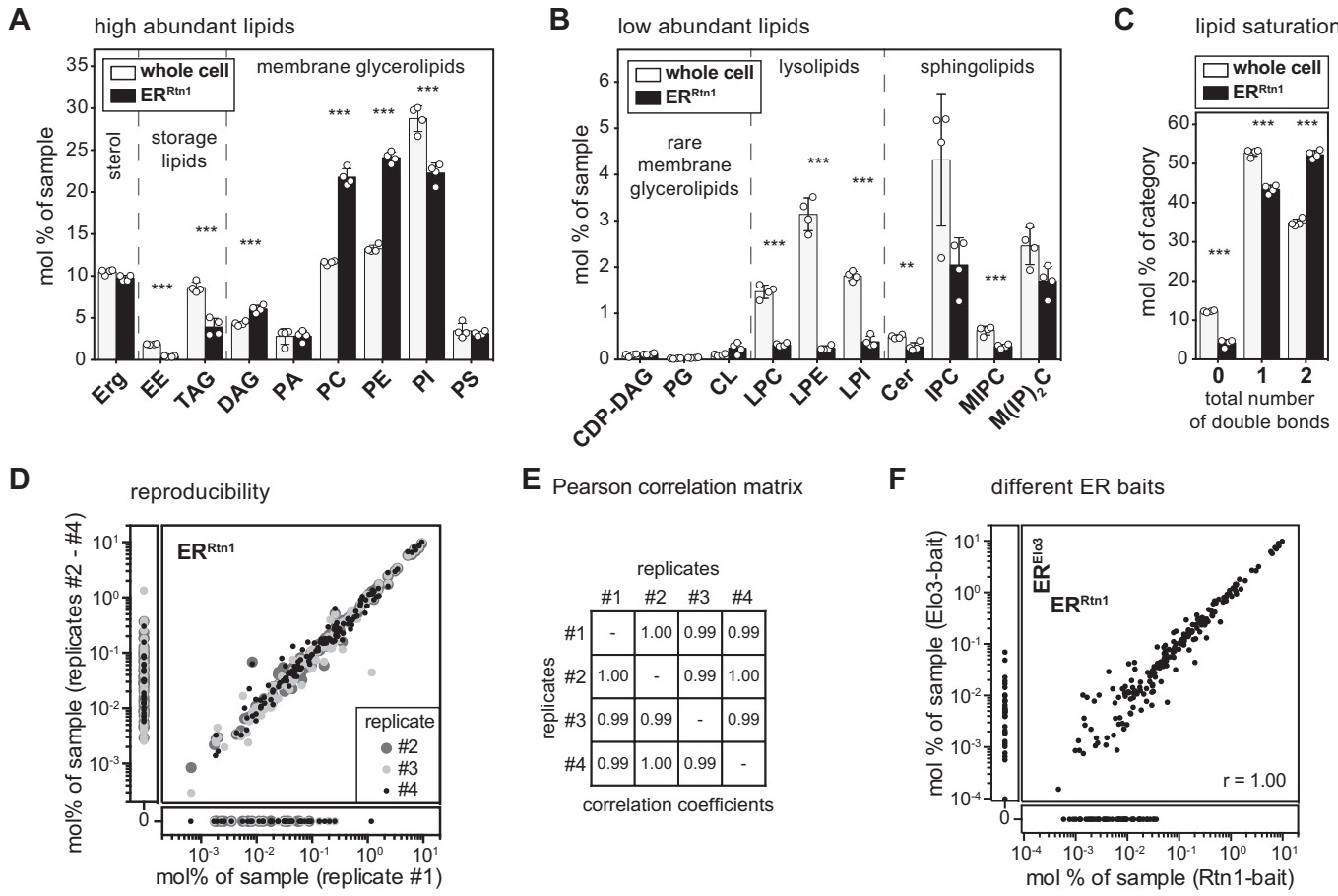

**Figure 2. Lipid composition of the ER membrane of *S. cerevisiae* determined by MemPrep via Rtn1-bait.**

SCD$_{complete}$ medium was inoculated to an OD$_{600}$ of 0.1 using stationary overnight cultures of cells expressing the bait protein. Cells were cultivated to an OD$_{600}$ of 1.0, harvested, frozen, stored, thawed, and then subjected to the MemPrep procedure. (A) Quantitative lipidomics reveals the lipid class composition given as mol% of all identified lipids in the sample. Classes are categorized into sterol (Erg ergosterol), storage lipids (EE ergosteryl ester, TAG triacylglycerol), membrane glycerolipids (DAG diacylglycerol, PA phosphatidic acid, PC phosphatidylcholine, PE phosphatidylethanolamine, PI phosphatidylinositol, PS phosphatidylserine) (*n* = 4 biological replicates). (B) Lipid class composition of rare membrane glycerolipids (CDP-DAG cytidine diphosphate diacylglycerol, PG phosphatidylglycerol, CL cardiolipin), lysolipids (LPC lyso-phosphatidylcholine, LPE lyso-phosphatidylethanolamine, LPI lyso-phosphatidylinositol), and sphingolipids (Cer ceramide, IPC inositolphosphorylceramide, MIPC mannosyl-IPC, M(IP)$_2$C mannosyl-di-IPC) given as mol% of all lipids in the sample (*n* = 4 biological replicates). (C) The total number of double bonds in membrane glycerolipids except for CL (i.e. CDP-DAG, DAG, PA, PC, PE, PG, PI, PS) as mol% of this category (*n* = 4 biological replicates). % (D) Reproducibility of immuno-isolated ER lipidome data shown as the correlation of mol% of sample values of all detected lipid species between replicate 1 and replicates 2–4. (E) Pearson correlation coefficients of lipidomics data for all combinations of replicate samples. (F) Correlation of mol% of sample values of all detected lipid species from Rtn1-bait and Elo3-bait derived ER membranes. Data information: Data from *n* = 4 biological replicates in (A–C) are presented as individual data points and as mean ± SD. **$P ≤ 0.01$, ***$P ≤ 0.001$ (multiple *t* tests, corrected for multiple comparisons using the method of Benjamini, Krieger, and Yekutieli, with *Q* = 1%, without assuming consistent SD). Nonsignificant comparisons are not highlighted. Source data for this figure are available online.

whole cells (10.5 mol%) (Fig. 2A) or in the *trans*-Golgi network/endosome (TGN/E) system (9.8 mol%) (Klemm et al, 2009), but much lower than in the plasma membrane (>44 mol%) (Surma et al, 2011). Notably, the absence of a steep sterol gradient in the early secretory pathway is not in conflict with previous studies (Zinser et al, 1991, 1993) and has important implications for the sorting of transmembrane proteins based on hydrophobic thickness (Bretscher and Munro, 1993; Ridsdale et al, 2006; Herzig et al, 2012). Complex sphingolipids such as inositolphosphorylceramide (IPC), mannosyl-IPC (MIPC), and mannosyl-di-(IP)C (M(IP)$_2$C) are found, as expected, at a significantly lower level in the ER (Figs. 2B and EV2B). While we cannot rule out that a minor fraction of these lipids may originate from contaminating

organelles, these data suggest a significant retrograde transport of complex sphingolipids from the Golgi apparatus to the ER, likely via COP-I vesicles together with ER-resident proteins bound to the HDEL receptor (Aguilera-Romero et al, 2008). A closer look at the fatty acyl chain composition of ER lipids reveals a particularly low level (<5 mol%) of tightly packing, saturated lipids and a significant enrichment of loosely packing, unsaturated lipids (Figs. 2C and EV2C). Loose lipid packing and high membrane compressibility are likely contributing to the remarkable ability of the ER to accept and fold the entire diversity of transmembrane proteins differing substantially in shape and hydrophobic thicknesses (Sharpe et al, 2010; Quiroga et al, 2013; Lorent et al, 2020; Renne and Ernst, 2023). Future work will be dedicated to quantifying also

phosphoinositides such as phosphatidylinositol-4,5-bisphosphate (PIP2), or phosphatidylinositol-3,4,5-triphosphate (PIP3).

In summary, our molecular analysis of the ER membrane reveals surprising insights, which are nevertheless consistent with previous findings and our current understanding of the properties and functions of the ER. The robustness and reproducibility of our MemPrep approach coupled to lipidomic platforms is demonstrated by the remarkable correlation of lipid abundances reported in four independent experiments with the Rtn1-bait (Fig. 2D,E) or between the isolates using the Rtn1- and the Elo3-bait (Fig. 2F).

## Stable lipid compositions after cell lysis contrasts ER lipid remodeling in living cells

While our isolation process is shorter than many previously employed methods for organelle purification, it still takes 8 h from cell lysis to finish. Hence, we wanted to exclude that ongoing lipid metabolism during the isolation procedure distorts the measured lipid composition. Consequently, we performed a control experiment in which we split a crude microsome preparation (P100) into two equal samples. The first sample was directly snap-frozen in liquid $N_2$ while the second one was frozen only after incubation at 4 °C for 8 h. A comparison of the two samples revealed remarkably similar lipid compositions (Appendix Fig. S2A–C). Only the low abundant lyso-PC, lyso-PE, and lyso-PI lipids showed some differences (Appendix Fig. S2B), suggesting a loss of lysolipids over time, which is consistent with their role as intermediates of lipid degradation (Harayama and Riezman, 2018). Hence, ongoing lipid metabolism has only a minor impact on the cellular lipidome.

## A molecular fingerprint of lipid bilayer stress during inositol depletion

Having successfully coupled the MemPrep technology to quantitative lipidomics, we turned our attention to the stressed ER. Lipid bilayer stress is a collective term for aberrant ER membrane compositions activating the UPR (Surma et al, 2013; Ho et al, 2018; Radanović and Ernst, 2021). It is expected that an acute depletion of inositol from the medium causes UPR activation without triggering a substantial accumulation of misfolded proteins (Cox et al, 1997; Promlek et al, 2011; Lajoie et al, 2012). We scored UPR activity in stressed and unstressed cells by determining the relative abundance of the spliced *HAC1* mRNA (Fig. EV3A) and the mRNA abundance of the UPR target genes *PDI1* and *KAR2* by RT-qPCR (Fig. EV3B,C). Depletion of inositol from the medium, but not supplementation with choline, triggered a robust UPR in both the Rtn1-bait strain and the respective wild-type yeast (BY4741) (Fig. EV3A–C). Hence, the bait tag attached to Rtn1 does not interfere with the ability of the cell to respond to lipid bilayer stress. We then isolated ER membranes via the Rtn1-bait and performed quantitative proteomics experiments. The ER membrane proteome remains remarkably unperturbed by inositol depletion: only 12 out of 2655 robustly detected proteins showed increased abundances in ER isolates upon inositol depletion (Fig. 3A). Among these, we found the myo-inositol transporter Itr1 (Nikawa et al, 1993), the glycerophosphoinositol permease Git1 (Patton-Vogt and Henry, 1998), and the soluble Inositol-3-phosphate synthase Ino1 (Hirsch and Henry, 1986), known to be transcriptionally upregulated in response to inositol depletion (Jesch et al, 2006).

Our main goal was establishing a molecular fingerprint of lipid bilayer stress in the ER (Fig. 3). Immuno-isolated ER membranes revealed, somewhat unsurprisingly, that inositol depletion causes a substantial drop of inositol-containing PI lipids in the ER (Fig. 3B) accompanied by a drastic accumulation of CDP-DAG lipids, which serve as direct precursors for PI synthesis via Pis1 (Fig. 3C,D) (Henry et al, 2012). Even the penultimate precursor of PI synthesis, PA, accumulates in the ER upon inositol depletion (Fig. 3B) (Henry et al, 2012). Inositol-containing sphingolipids, however, are not depleted under this condition (Fig. 3C). This implies distinct rates of PI and sphingolipid metabolism under this condition. Overall, the molecular lipid fingerprint of the lipid bilayer stress caused by inositol depletion is characterized by substantial changes in the abundance of anionic lipids, PI in particular (Fig. 3B,C).

We further dissected the compositional changes of the ER membrane lipidome upon inositol depletion at the level of the lipid acyl chains and observed a minor, nonsignificant trend toward more saturated glycerophospholipids (Fig. 3E). While these changes are likely to fine-tune the physicochemical properties of the ER membrane, it is unlikely that they alone are sufficient to trigger the UPR by activating Ire1 (Halbleib et al, 2017). Hence, we speculate that the overall reduction of anionic lipids might contribute to lipid bilayer stress. Our data provide a quantitative basis for studying the contribution of anionic lipids and collective membrane properties to chronic ER stress in vitro after reconstituting UPR transducers in native-like membrane environments.

## An increased PC-to-PE ratio does not cause lipid bilayer stress in yeast

Increased cellular PC-to-PE ratios have been associated with chronic ER stress in mammalian cells (Fu et al, 2011), while a decreased production of PC from PE causes lipid bilayer stress in yeast (Thibault et al, 2012; Ho et al, 2020; Ishiwata-Kimata et al, 2022). Because cells cultivated in a synthetic medium and challenged with 2 mM choline do not activate the UPR (Fig. EV3), we were interested in the ER lipidome under this condition. We isolated ER membranes via the Rtn1-bait and determined the resulting lipid composition (Appendix Fig. S3A–C). Choline can be activated to CDP-choline and then transferred onto diacylglycerol (DAG) to yield PC (Appendix Fig. S3D) (Kennedy and Weiss, 1956). Expectedly, the ER of choline-challenged cells features substantially higher levels of PC lipids at the expense of PE and causes an increase of the PC-to-PE ratio from ~1.1 to ~2.4 (Appendix Fig. S3A). This increased PC-to-PE ratio in the ER of choline-challenged cells is neither associated with changes in lipid saturation (Appendix Fig. S3C) nor with UPR activation (Fig. EV3A–C). Lipid metabolism and the PC-to-PE ratio may have been more affected if different concentrations of choline and inositol had been used (Hirsch and Henry, 1986; Gaspar et al, 2006). Nevertheless, our data show that even a grossly increased PC-to-PE ratio, which likely affects the lateral pressure profile and ER membrane fluidity (van den Brink-van der Laan et al, 2004; Marsh, 2007; Dawaliby et al, 2016) does not cause lipid bilayer stress in yeast.

## Lipid bilayer stress caused by proteotoxic agents Dithiothreitol (DTT) and Tunicamycin (TM)

Instances of acute proteotoxic stress disrupt protein folding in the ER and activate the UPR without causing substantial alterations to

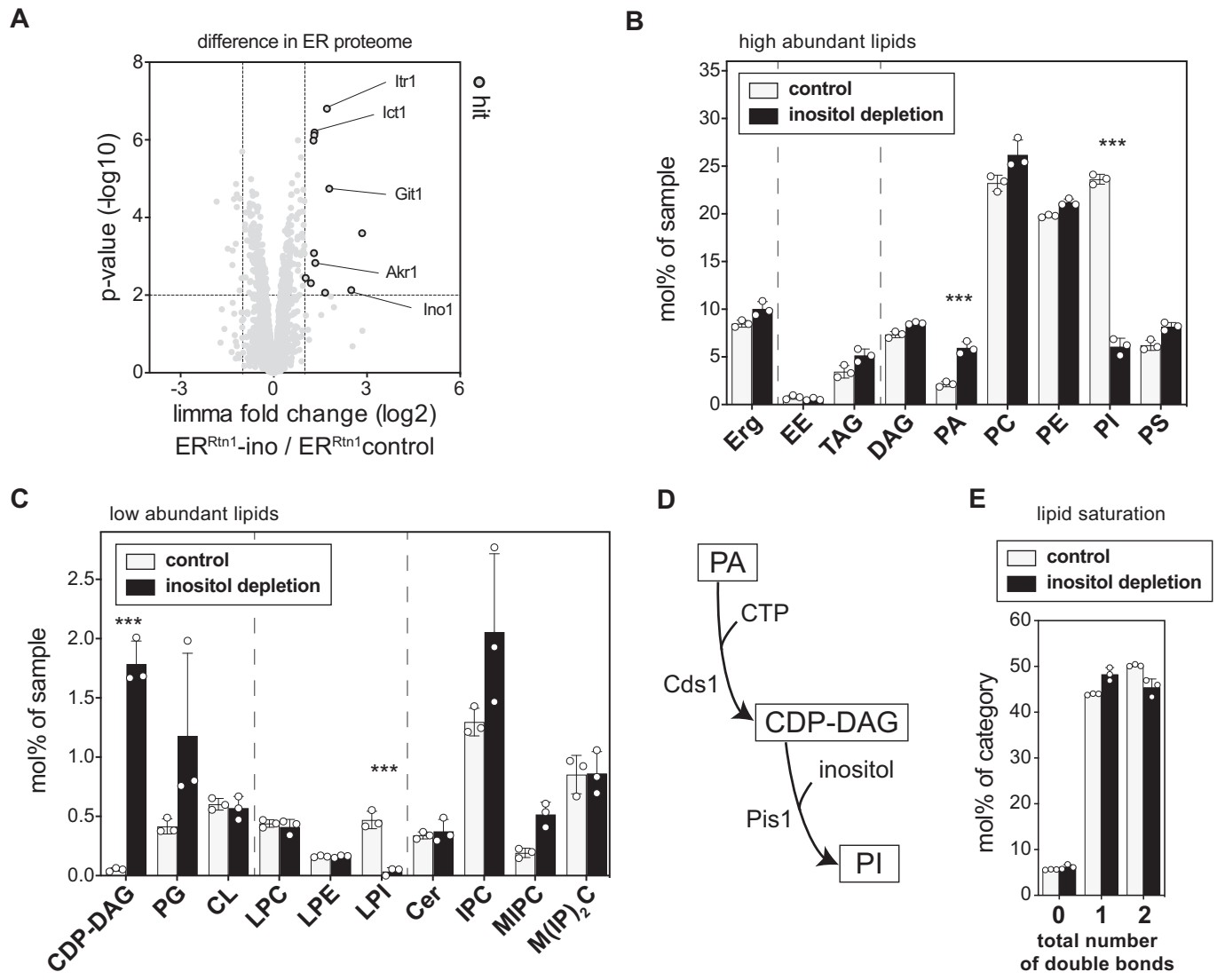

**Figure 3. Molecular fingerprints of lipid bilayer stress.**

SCD$_{complete}$ medium was inoculated with Rtn1-bait cells to an OD$_{600}$ of 0.003 from an overnight pre-culture and grown to an OD$_{600}$ of 1.2. Cells were washed with inositol-free medium and then cultivated for an additional 2 h in either inositol-free (inositol depletion) or SCD$_{complete}$ medium (control) starting with an OD$_{600}$ of 0.6. Note that the culturing conditions for the unstressed control was different from that used to determine the steady-state ER lipid composition in Fig. 2. ER-derived membranes were purified by differential centrifugation and immunoisolation and subsequently analyzed by TMT-labeling proteomics or quantitative shotgun lipidomics. (A) To increase the proteomics coverage for membrane proteins, P100 membranes were carbonate-washed before performing immunoisolation. Limma analysis of TMT-labeling proteomics. Highlighted are proteins that are more abundant in ER samples upon lipid bilayer stress by inositol depletion (ER$^{Rtn1}$-ino). (B) Lipid class composition given as mol% of all lipids in the sample. Erg ergosterol, EE ergosteryl ester, TAG triacylglycerol, DAG diacylglycerol, PA phosphatidic acid, PC phosphatidylcholine, PE phosphatidylethanolamine, PI phosphatidylinositol, PS phosphatidylserine ($n = 3$ biological replicates). (C) Class distribution of low abundant lipids CDP-DAG cytidine diphosphate diacylglycerol, PG phosphatidylglycerol, CL cardiolipin, LPC lyso-phosphatidylcholine, LPE lyso-phosphatidylethanolamine, LPI lyso-phosphatidylinositol, Cer ceramide, IPC inositolphosphorylceramide, MIPC mannosyl-IPC, M(IP)$_2$C mannosyl-di-IPC ($n = 3$ biological replicates). (D) Lipid metabolic pathway of PI biogenesis. (E) The total number of double bonds in membrane glycerolipids (except CL which has four acyl chains) as mol% of this category ($n = 3$ biological replicates). Data information: Data from $n = 3$ biological replicates in (A) are presented as the mean. We used the moderated t-test limma to test for differential enrichment. P values were corrected for multiple testing with the method from Benjamini and Hochberg. The data from three biological replicates are presented as individual points in (B, C, E) and as the mean ± SD. ***$P \leq 0.001$ (multiple t tests, corrected for multiple comparisons using the method of Benjamini, Krieger, and Yekutieli, with $Q = 1\%$, without assuming consistent SD). Nonsignificant comparisons are not highlighted. Source data are available online for this figure.

cellular lipidomes (Reinhard et al, 2020). However, prolonged proteotoxic stress triggers the UPR through a membrane-based mechanism (Promlek et al, 2011; Väth et al, 2021), with the underlying molecular basis remaining largely unexplored. Even prolonged yeast cultivation without external stressors transiently triggers the UPR around the time of the diauxic shift (Tran et al, 2019a). To replicate these observations in our experimental context, we investigated the impact of DTT, TM, and extended cultivation on UPR activity (Fig. EV4). Cells exposed to 2 mM DTT or 1.5 µg/ml TM for 4 h exhibited potent UPR activity, as evidenced by

increased mRNA abundance of spliced *HAC1*, *PDI1*, and *KAR2* compared to cells harvested before exposure to the proteotoxic agents (Fig. EV4A–C; pre-stress). As expected, TM failed to induce a UPR in *ire1Δ* cells (Fig. EV4A–C). Prolonged cultivation in the absence of proteotoxic agents triggered a mild UPR in both wild-type yeast and the Rtn1-bait strain (Fig. EV4A–C).

We sought to understand how prolonged proteotoxic stress and extended cultivation without external stressors affected the lipid composition of wild-type and *ire1Δ* cells (Fig. 4A). Comprehensive analyses revealed that DTT and TM have a remarkably similar impact on the cellular lipid composition (Fig. 4A, see Source Data for full dataset). Stressed cells, compared to exponentially growing cells before the proteotoxic insult, showed elevated levels of PC, DAG, ergosterol esters (EEs), and TAGs, but reduced levels of PE and PI (Fig. 4A). If the increased levels of storage lipids reflect an increased abundance in the ER membrane or a more intimate interaction of the ER with lipid droplets, which accumulate under ER stress (Stordeur et al, 2014; Garcia et al, 2021), remains to be investigated.

Prolonged cultivation in the absence of supplemented proteotoxic agents also resulted in significant remodeling of the cellular lipidome and much lower EE and PI levels compared to DTT- or TM-treated cells (Fig. 4A; untreated versus TM or DTT). Because standard SCD medium contains only 11 µM inositol and because the BY4741 is particularly dependent on inositol for normal growth (Hanscho et al, 2012) it is possible that prolonged cultivation of this strain leads to a "natural" inositol depletion. The increased abundances of precursors of PI biosynthesis, PA and CDP-DAG, also point in this direction. While it is likely that the mildly reduced levels of PI contribute to the membrane-based ER stress upon prolonged DTT or TM treatments, our data suggest a more complex remodeling of the cellular lipidome, which is distinct from the effects of inositol depletion and prolonged cultivation. This conclusion is corroborated by a principal component analysis of whole-cell lipidomic data showing distinct clustering of the data derived from pre-stressed, untreated, inositol-depleted, and the DTT- or TM-stressed cells (Fig. 4B). Notably, *ire1Δ* cells exhibited nearly identical lipidomic changes as their wild-type counterparts regardless of the treatment, and the respective data from wild-type and *ire1Δ* cells co-clustered in principal component analyses (Fig. 4B). Thus, lipidome remodeling cannot be attributed to UPR signaling in this context but may be involved in UPR induction (Fig. EV4A–C). Instead, lipidome remodeling may be related to the strong growth defect that is induced by both drugs (Reinhard et al, 2020).

Given our interest in the molecular basis of UPR activation by lipid bilayer stress, we focused on the impact of prolonged DTT and TM treatments on ER membrane composition. These treatments trigger a robust UPR by a membrane-based mechanism (Fig. EV4A–C) (Promlek et al, 2011; Väth et al, 2021) and induce severe remodeling of the cellular lipidome (Fig. 4B). We aimed to provide a quantitative description of the ER membrane composition from stressed cells, hoping it would facilitate a direct comparison of different types of lipid bilayer stress and reveal both commonalities and differences. Even though DTT and TM have different modes of action, they have a remarkably similar impact on ER membrane composition (Fig. 4D,E). The stressed ER features a lower content of unsaturated membrane lipids (Fig. 4C) and exhibits higher levels of storage lipids (EEs and TAGs)

(Fig. 4D), possibly due to reduced growth rates (Reinhard et al, 2020) and increased fatty acid flux into storage lipids, or a gradual depletion of lipid metabolites such as inositol from the medium (Listenberger et al, 2003; Vevea et al, 2015; Henne et al, 2018; Reinhard et al, 2020). The elevated levels of lipid metabolic intermediates CDP-DAG and DAG in the stressed ER support both possibilities (Fig. 4D,E). In fact, inositol depletion is a possible contributor to the membrane-based activation of the UPR under this condition. However, the most notable change was observed in the abundant lipid classes PC and PE. In the unstressed ER, the PC-to-PE ratio was 1.0, while the DTT- and TM-stressed ER featured PC-to-PE ratios of 2.8 and 3.1, respectively (Fig. 4D). The UPR transducer Ire1 is unlikely to be directly activated by increased PC-to-PE ratios, as artificially increasing it by choline supplementation to ~2.4 does not activate the UPR (Appendix Fig. S3A; Fig. EV3A–C), whereas inositol depletion triggers the UPR without perturbing the PC-to-PE ratio (Figs. 3B and EV3) (Fu et al, 2011; Gao et al, 2015; Ho et al, 2020; Ishiwata-Kimata et al, 2022). Instead, the ER membrane of DTT- and TM-stressed cells features lower levels of negatively charged, inositol-containing lipids (PI, LPI, IPC, MIPC), which is only partially compensated by increased levels of PA and CDP-DAG (Fig. 4D,E). The molecular fingerprints of lipid bilayer stress provide a crucial framework for dissecting the modulatory role of anionic lipids in UPR activation in the future.

## Characterizing complex ER-like lipid mixtures in vitro and in silico

Based on lipidomic data on isolated ER membranes and using twelve commercially available lipids, we established ER-like lipid compositions mimicking the stressed and unstressed ER (Appendix Fig. S4A). For each condition, these mixtures match the lipid class composition, the overall degree of lipid saturation, and the acyl chain composition of each lipid class. We assessed the molecular lipid packing density of these ER-like lipid mixtures in large unilamellar vesicles (LUVs) using the solvatochromic probe C-laurdan (Appendix Fig. S4B), which reports on water penetration into the membrane (Kim et al, 2007). Liposomes mimicking the ER upon inositol depletion featured higher generalized polarization ($GP_S$) values than those mimicking the unstressed ER and the ER exposed to prolonged proteotoxic stresses. Hence, there is no correlation between lipid packing in the water-membrane interface as measured by C-laurdan (Appendix Fig. S4B) and UPR activity (Figs. EV3 and EV4). This also means that lipid packing in this region of the bilayer is not the dominant modulator of UPR activity, which is consistent with the proposed hydrophobic mismatch-based mechanism of Ire1 that predicts a contribution of lipid packing across the entire lipid bilayer (Halbleib et al, 2017; Covino et al, 2018). All-atom molecular dynamics (MD) simulations revealed substantial differences between different ER-like lipid bilayers and those composed solely of PC lipids in terms of membrane thickness (Appendix Fig. S4D), lipid packing defects (Appendix Fig. S4E), and free volume profile (Appendix Fig. S4F). This highlights that PC-dominated membrane mixtures are not an accurate mimic for the ER membrane. Compared to a lipid composition mimicking the unstressed ER, the stressed ER (induced by DTT/TM or inositol depletion) was significantly thicker (Appendix Fig. S4D), aligning with the notion that

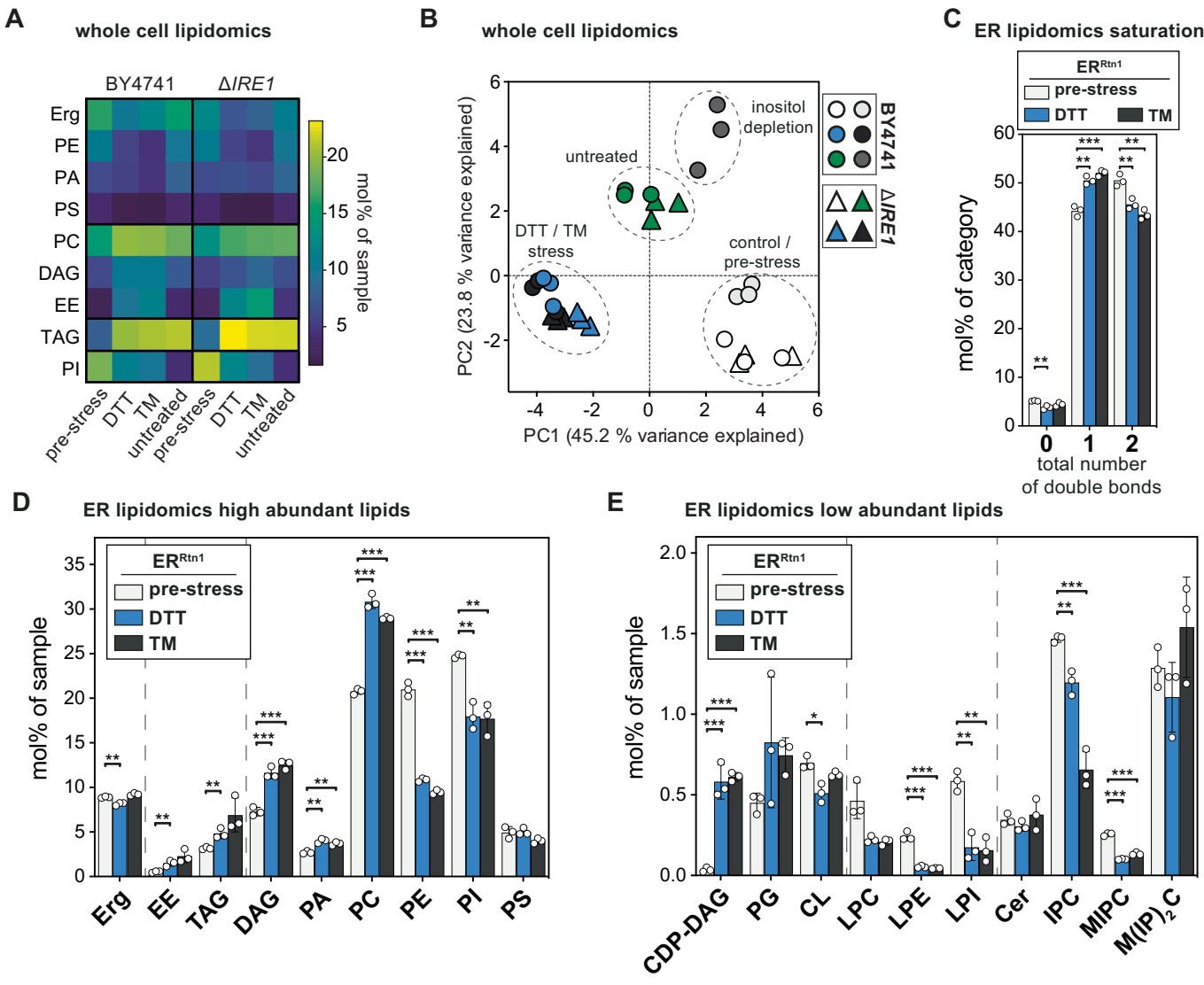

**Figure 4. ER stress induced by DTT and TM manifests in a distinct lipid fingerprint on the whole cell and ER level.**

(A) SCD$_{complete}$ medium was inoculated with BY4741 wild-type or $\Delta IRE1$ cells to an OD$_{600}$ of 0.1 from an overnight pre-culture. Cells were grown to an OD$_{600}$ of 0.8 and then stressed by the addition of either 2 mM DTT or 1.5 µg/ml TM or left untreated for 4 h. The lipidome of whole cells was determined by quantitative shotgun mass spectrometry. Mean abundance from three biologically independent replicates is shown as mol% of all lipid classes identified in the sample. Only classes with significant changes are shown and clustered by their abundance pattern. Erg, PE, PA, PS are decreased in DTT- and TM-stressed cells. PC, DAG, EE are increased in stressed cells. TAG increases in all three conditions (DTT/TM stress, untreated). PI is slightly decreased in DTT- and TM-stressed cells and strongly decreased in untreated cells. (B) Principal component analysis (PCA) of whole-cell lipidomics data from wild-type (BY4741, circles) and $\Delta IRE1$ (triangles) cells. Cells were subjected to prolonged proteotoxic stress by DTT or TM or left untreated. PCA includes whole-cell lipidomes of direct lipid bilayer stress by inositol depletion. Lipidomes of DTT and TM stress cluster together, indicating a high degree of similarity. Lipidomes of untreated cells form a distinct cluster different from pre-stressed and DTT- or TM-stressed conditions. Lipidomes of inositol depletion form a distinct cluster, the respective control condition is close to the pre-stress cluster. Cells for inositol depletion were grown as described in Fig. 3. Interestingly, lipidomes of $\Delta IRE1$ cells cluster with their respective wild-type counterparts, indicating little influence of UPR activity on the cellular lipidome under these conditions. (C) Rtn1-bait cells were grown as described for (A). ER-derived membranes were purified by MemPrep and subsequently analyzed by quantitative shotgun lipid mass spectrometry. Total number of double bonds in membrane glycerolipids (without CL) given as mol% of this category ($n = 3$ biological replicates). (D) Lipid class distribution of sterol, storage lipids and abundant membrane glycerolipids in ER-derived vesicles from cells that were either challenged with 2 mM dithiothreitol (DTT) or 1.5 µg/ml TM for 4 h. The ER lipidome undergoes significant remodeling upon ER stress ($n = 3$ biological replicates). (E) Lipid class distribution of rare membrane glycerolipids, lysolipids, and sphingolipids ($n = 3$ biological replicates). Data information: Data from $n = 3$ biological replicates in (A) are shown as the mean. Data from $n = 3$ biological replicates in (C–E) are presented as individual data points and as the mean ± SD. *$P \le 0.05$, **$P \le 0.01$, ***$P \le 0.001$ (multiple $t$ tests, corrected for multiple comparisons using the method of Benjamini, Krieger, and Yekutieli, with $Q = 1\%$, without assuming consistent SD). Nonsignificant comparisons are not highlighted. Source data are available online for this figure.

Ire1 senses lipid bilayer stress through a hydrophobic mismatch-based mechanism (Halbleib et al, 2017; Covino et al, 2018). A particularly intriguing difference between stressed and unstressed ER-like mixtures was the distinct distribution of positive and negative charges in the water-membrane interface (Appendix Fig. S4G) reflecting the different abundance of anionic lipids in the stressed ER (Fig. EV4D; Appendix Fig. S4C). These observations suggest that membrane thickness and membrane compressibility regulate the activity of the UPR and identify anionic lipids as potential modulators. The underlying mechanisms, however, remain to be dissected by in vitro experiments. Beyond that, our lipidomic data on the stressed and unstressed ER do not only establish lipid fingerprints of the stressed ER, but also provide a resource for studying the structure, folding, and function of ER membrane proteins in more realistic membrane environments.

## DTT and TM have similar yet distinct impact on the ER proteome

Next, we were interested in the repercussions of prolonged proteotoxic stress on the ER proteome. To this end, we isolated ER membranes from stressed and unstressed cells for a quantitative analysis via untargeted proteomics. Prior to subjecting microsomal membranes to the immunoisolation procedure, we washed the microsomes with sodium carbonate to remove loosely attached peripheral proteins and contaminating cytosolic proteins even more efficiently. Membrane contact sites are remodeled during ER stress in both yeast and mammalian cells (Vevea et al, 2015; Liu et al, 2017; Kwak et al, 2020; Liao et al, 2022). To exclude that any of the observed proteomic changes are due to increased contaminations with other organelles, we first assessed the impact of DTT and TM on the quality of our ER preparations by immunoblotting. Expectedly, the GPI-anchored cell wall protein Gas1 accumulates as a ~105 kDa precursor in the ER of TM-stressed cells, but not in the ER or DTT-stressed cells (Fig. EV5A) (Fankhauser and Conzelmann, 1991; Wang et al, 2024). The purity of the ER isolations was assessed by determining the enrichment of marker proteins for the ER, the vacuole, mitochondria, and endosomes (Fig. EV5A–C). While the Rtn1-bait and the ER membrane protein Dpm1 were several-fold enriched in the immunoisolation step starting from microsomes, the markers for the vacuole (Vph1), mitochondria (Por1), and endosomes (Pep12) were depleted relative to the microsomal P100 fraction (Fig. EV5A–C). Hence, MemPrep allows for the isolation of both the unstressed and the stressed ER.

Using untargeted, quantitative proteomics on ER isolates a total of 2952 proteins were robustly detected in three biological replicates of both the stressed and unstressed ER. Prolonged proteotoxic stresses cause a major remodeling of the ER proteome that involves hundreds of proteins (Fig. 5A,B). Notably, this is in stark contrast to inositol depletion, which affects the abundance of only a few proteins in the ER (Fig. 3A).

Globally, the ER proteomes of DTT- and TM-stressed cells are similar to each other (Pearson correlation coefficient $r = 0.82$, Fig. EV5D). Canonical UPR targets, including ER-luminal (co-)chaperones Kar2, Sil1, and Lhs1, proteins involved in disulfide bridge formation Eug1 and Ero1, and the lipid metabolic enzyme Ino1 were upregulated in the stressed ER (Fig. 5A,B) (Travers et al,

2000; Jesch et al, 2006). The increased abundance of various lipid metabolic enzymes such as Plb1, Plb3, and Cld1 for acyl chain remodeling may contribute to the lipidomic changes observed for the stressed ER and reflect, at least in part, homeostatic responses to maintain ER membrane function upon stress (Renne et al, 2015). Furthermore, major reorganizations of the secretory pathway in response to both DTT- and TM treatments can be inferred from the ER accumulation of the HDEL receptor Erd2 and crucial components of the COP-I (Emp46, and Sly1) and COP-II (Ret2) machinery.

To functionally annotate the complex proteomic changes, we determined the enrichment of gene ontology terms (GO terms) in all upregulated proteins (Fig. 5C). While DTT seems to act more prominently on vesicular transport and autophagic processes (regulation of macroautophagy) (Fig. 5C), TM affects more selectively hydrolytic enzymes and carbohydrate-related metabolic processes thereby leading to an aberrant ER accumulation of vacuolar proteins and cell wall proteins. This can be expected because TM is crucial for the maturation of N-glycosylated proteins and GPI-anchored proteins. To further investigate the differences of DTT- and TM-induced changes of ER proteomes, we performed K-means clustering of the proteomic data (Fig. EV5D). The analysis of GO term enrichments for the individual clusters revealed a small group of proteins that were accumulated in the DTT-stressed ER but depleted in the ER from TM-stressed cells (Fig. EV5D,E, cluster 2). These proteins are involved in copper and iron transport (Fre7, Ctr1, and Fre1), which is interesting because iron affects the clustering propensity of Ire1 and the amplitude of UPR signaling (Cohen et al, 2017).

Taken together, our proteomics data suggest that DTT and TM treatments induce globally similar, yet qualitatively distinct forms of stress, which are reflected in distinct ER proteomes. Both forms of ER stress cause an accumulation of non-ER proteins in the ER, whose contribution to UPR activation remains to be systematically investigated.

## Different forms of lipid bilayer stress leave different marks in the lipid acyl chain region

Prolonged proteotoxic stress and inositol depletion exert discrete effects on the ER proteome (Figs. 3A and 5A,B) and the lipid class composition in the ER (Figs. 3A,B and 4C–E). These discrete effects also extend to the lipid acyl chains (Appendix Fig. S5). Upon inositol depletion, a nuanced shift toward shorter and more saturated acyl chains is observed across major glycerophospholipid classes (Appendix Fig. S5A). This contrasts with the impact of an increased PC-to-PE ratio enforced by choline supplementation, which barely leaves any marks in the lipid acyl chain composition (Appendix Fig. S5B). Moreover, prolonged proteotoxic stress induced by DTT or TM elicits distinctive impacts on the composition of lipid acyl chains (Appendix Fig. S5C,D). In this context, the acyl chains demonstrate a tendency throughout most lipid classes to become slightly longer and more saturated. Collectively, these observations underscore the notion that different forms of lipid bilayer stress are based on unique molecular signatures despite having common denominators. Consequently, targeted perturbations in lipid metabolism designed to ameliorate specific types of lipid bilayer stress may prove detrimental in other metabolic contexts.

                                    

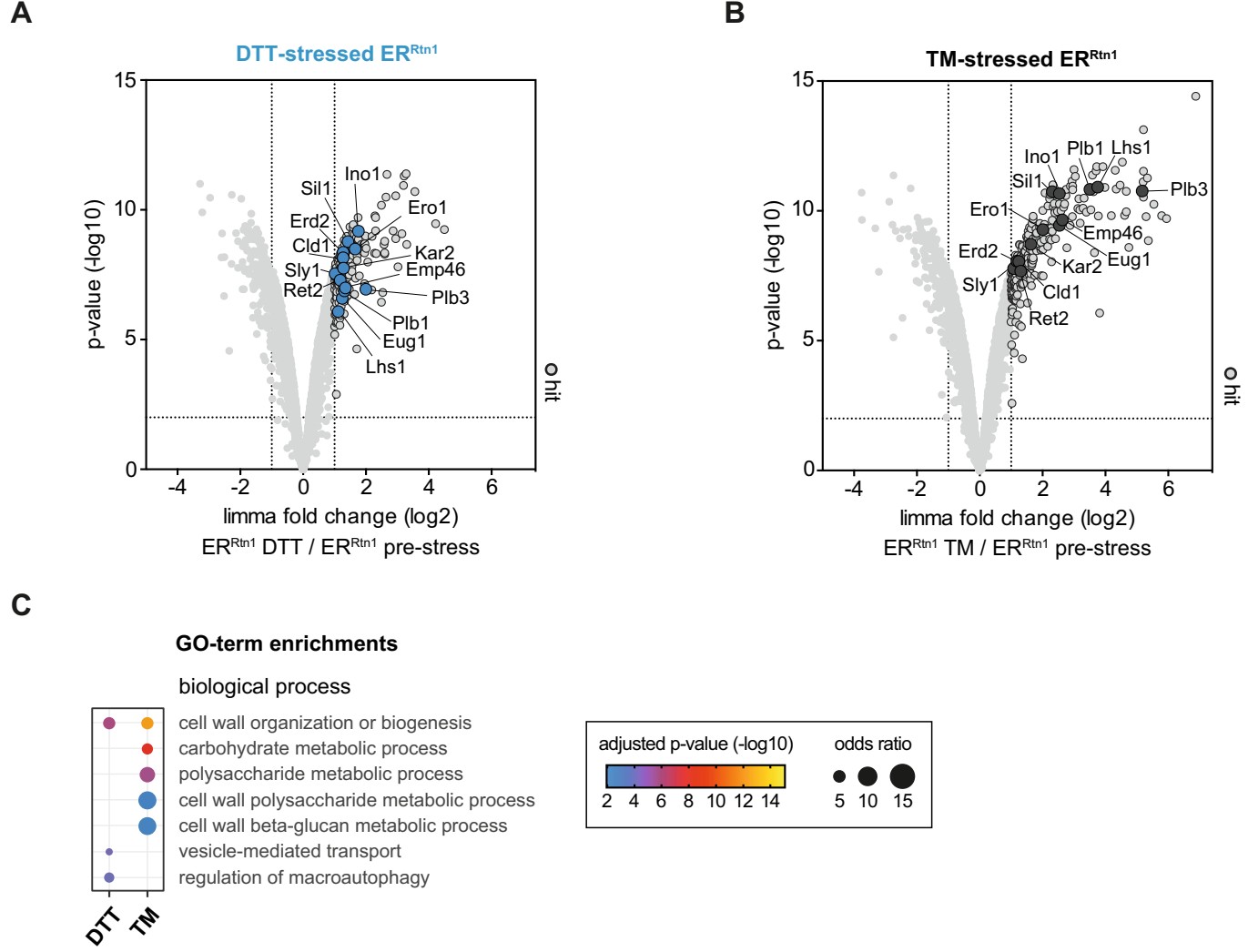

**Figure 5.  The proteome of the ER under conditions of prolonged proteotoxic stress.**

ER-derived vesicles were isolated by MemPrep and subsequently analyzed by untargeted proteomics. An additional sodium carbonate wash step was performed on P100 to remove soluble proteins from the membrane preparation. (**A**) Limma analysis identified proteins that are accumulating in ER preparations after prolonged DTT-induced stress (top right quadrant of volcano plot). Proteins that are discussed in the text are indicated. The enrichment of proteins in preparations of the stressed ER membrane was considered significantly when they were at least twofold enriched compared to their abundance in pre-stress conditions with a *P* value < 0.01 (*n* = 3 biological replicates). (**B**) Limma analysis showing proteins that are accumulating in the ER upon prolonged TM-induced ER stress (top right quadrant of volcano plot). All proteins discussed in the text are labeled. The enrichment of proteins in the preparations of the stressed ER was considered significant when they were enriched at least twofold compared to their abundance in pre-stress conditions with a *P* value < 0.01 (*n* = 3 biological replicates). (**C**) Enriched gene ontology terms (GO terms) in the list of enriched proteins. GO terms are grouped by categories, FDR < 1% (*n* = 3 biological replicates). Data information: Data in (**A**, **B**) are presented as the mean from three biological replicates. We used the moderated t-test limma to test for differential enrichment. *P* values were corrected for multiple testing with the method from Benjamini and Hochberg. Data in (**C**) presented as mean from three biological replicates. *P* values were derived from a Fisher-test and corrected for multiple testing with the method of Benjamini and Hochberg. Source data are available online for this figure.

## Demonstrating the broad applicability of MemPrep on vacuolar membranes

While MemPrep was initially designed and optimized for the isolation of ER membranes, our objective was to develop its applicability for isolating membranes from various organelles. This versatility was validated by successfully isolating vacuolar membranes (Fig. 6A,B). Given that the vacuole receives membrane material via the secretory pathway, endocytosis, macroautophagy, lipophagy, and direct lipid transfer, it was unclear what the lipid composition of the vacuole would be even though its lipid composition has been partially addressed before (Schneiter et al, 1999; González Montoro et al, 2018).

To investigate the lipid composition of the vacuole, we utilized a bait-tagged variant of Vph1, a subunit of the abundant ATP-driven proton pump in the vacuole. Employing the same procedures as for the ER isolation but with increased starting material, we conducted the subcellular fractionation (Appendix Fig. S6A) and immunoisolation (Fig. 6A). Immunoblot analysis of the final isolate confirmed the presence of two vacuolar membrane proteins (the

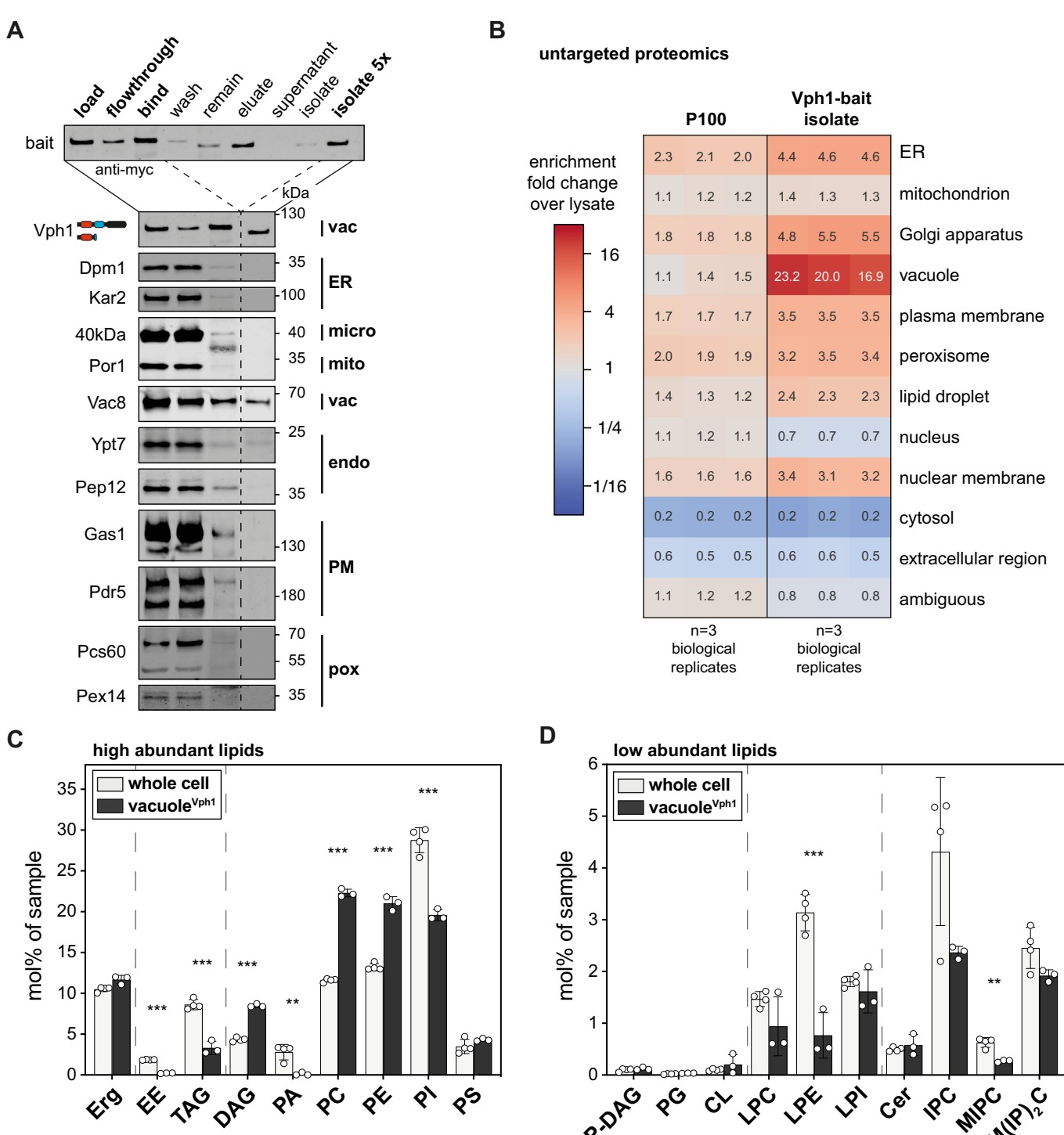

Vph1-bait and the palmitoylated Vac8), while markers for other organelles remained undetectable, underscoring the broad applicability of MemPrep for organelle membrane isolation (Fig. 6A).

Untargeted proteomics robustly detected 3264 proteins in all three biological replicates and revealed a >20-fold enrichment of many annotated vacuolar proteins (both soluble and membrane proteins) in the final isolate (Fig. 6B). Only a few proteins

annotated to other organelles were enriched >20-fold in the final isolate (Appendix Fig. S6B). Apart from Sys1, which is well-characterized as membrane protein in the Golgi apparatus, all other proteins (Adp1, the uncharacterized Ypr003c, Syg1, Fet5, and Tul1) have previously been observed to localize to the vacuole, affirming the accuracy of our isolation method (Breker et al, 2013, 2014; Yofe et al, 2016; Weill et al, 2018). More specifically, the iron oxidase

**Figure 6.   Lipid composition after MemPrep of the vacuolar membrane.**

(A) Immunoblot analysis of fractions after immunoisolation via a vacuolar bait protein (Vph1-bait). Common organellar markers are shown: ER, endoplasmic reticulum (Dpm1 and Kar2); micro, microsomal fraction (40 kDa); mito, mitochondria (Por1); vac, vacuole (Vac8); endo, endosomal system (Ypt7 and Pep12); PM, plasma membrane (Gas1 and Pdr5); pox, peroxisomes (Pcs60 and Pex14). 0.2% of each fraction loaded per lane. (B) Untargeted protein mass spectrometry analysis showing enrichment of P100 and isolate fractions over whole-cell lysate. The enrichment of proteins over the cell lysate (fold change) is based on uniquely annotated subcellular locations and provided for each of $n = 3$ biological replicates. The illustrated numbers represent the median enrichment for each biological replicate. (C) Lipid class composition given as mol% of all lipids in the sample. Classes are categorized into sterol (Erg ergosterol), storage lipids (EE ergosteryl ester, TAG triacylglycerol), membrane glycerolipids (DAG diacylglycerol, PA phosphatidic acid, PC phosphatidylcholine, PE phosphatidylethanolamine, PI phosphatidylinositol, PS phosphatidylserine). Whole-cell lipid data are identical with the data presented in Fig. 2A,B ($n = 4$ biological replicates for whole cell; $n = 3$ biological replicates for vacuole$^{Vph1}$). (D) Continuation of lipid class composition given as mol% of all lipids in the sample. Classes are categorized into rare membrane glycerolipids (CDP-DAG cytidine diphosphate diacylglycerol, PG phosphatidylglycerol, CL cardiolipin), lysolipids (LPC lyso-phosphatidylcholine, LPE lyso-phosphatidylethanolamine, LPI lyso-phosphatidylinositol) and sphingolipids (Cer ceramide, IPC inositolphosphorylceramide, MIPC mannosyl-IPC, M(IP)$_2$C mannosyl-di-IPC). Whole-cell lipid data are identical with the data presented in Fig. 2B. Data information: Data in (B) are presented as the median enrichment of uniquely annotated genes for $n = 3$ biological replicates. Data in (C, D) are presented as individual data points and the mean ± SD. **$P \le 0.01$, ***$P \le 0.001$ (multiple $t$ tests, corrected for multiple comparisons using the method of Benjamini, Krieger, and Yekutieli, with $Q = 1\%$, without assuming consistent SD). Nonsignificant comparisons are not highlighted. Source data are available online for this figure.

Fet5 was demonstrated to reside in the vacuole (Urbanowski and Piper, 1999), while Tul1 is part of the yeast Dsc E3 ubiquitin ligase complex that localizes, depending on its exact composition, to the Golgi apparatus, endosomes, or the vacuole (Yang et al, 2018). These observations collectively support the conclusion that the MemPrep procedure can yield highly pure vacuole membranes.

Further analysis through shotgun lipidomics unveiled substantial differences in lipid composition between vacuolar membranes and whole-cell lysate (Fig. 6C,D), as well as the plasma membrane (Surma et al, 2011). Noteworthy similarities with the ER, such as the abundance of PC, PI, and complex sphingolipids, were observed (Appendix Fig. S6C,D). However, distinct features included significantly higher levels of ergosterol and DAGs in the vacuole membrane, as well as an almost complete absence of PA lipids (Appendix Fig. S6C). These findings align with previous studies using alternative protocols for vacuole isolation (González Montoro et al, 2018; Kim and Budin, 2024) and our own work utilizing Mam3 as an alternative bait protein (Reinhard et al, 2023).

In line with its function of the vacuole as a lipid-degrading organelle, elevated levels of lysolipids (LPC, LPE, and LPI) were detected in the vacuole compared to the ER membrane (Appendix Fig. S6D). In addition, the lipid fatty acyl chains are more saturated in the vacuole compared to the ER membrane (Appendix Fig. S6E). These findings underscore the remarkable versatility of the MemPrep immunoisolation procedure and its suitability for organellar lipidomics.

## Discussion

Understanding the homeostasis and adaptation of organellar membranes to metabolic perturbation and cellular stress is one of the key challenges in membrane biology. We developed MemPrep for the isolation of organellar membranes and a quantitative characterization of their composition. The versatility of this approach is demonstrated by the immunoisolation of membrane vesicles from two organelles in yeast: the ER and the vacuole. Using state-of-the-art lipidomics we provide a comprehensive, molecular description of their membrane composition and establish a baseline for dissecting the role of lipids in transmembrane protein folding, trafficking, and function. Atomistic molecular dynamics (MD) simulations highlight the difference between ER-mimetic

membranes and PC-based lipid bilayers with respect to membrane thickness, lipid packing, the free volume profile, and surface charge distribution (Appendix Fig. S4C–G). The biochemical reconstitution of ER proteins in more realistic membrane environments is now feasible and will become particularly relevant for the characterization of membrane property sensors and the machineries that insert and extract membrane proteins into and out of the ER, respectively (Covino et al, 2018; Wu and Rapoport, 2021).

MemPrep overcomes the challenges associated with extensive membrane contact sites for purifying organelle membranes. In contrast to recent strategies optimized for a rapid precipitation of organelles from yeast and mammalian cells (Liao et al, 2018; Melero et al, 2018; Ray et al, 2020; Higuchi-Sanabria et al, 2020), MemPrep maximizes purity at the expense of low yields and provides full access to the eluted membrane vesicles for a straightforward coupling to quantitative, analytical platforms. Using the Rtn1-bait, MemPrep facilitates a mean enrichment of 27.7 for over 213 tested ER-resident proteins. This is remarkable, because even a sevenfold enrichment over the cell lysate has been considered as sufficient or even optimal in the past (Zinser and Daum, 1995). While we cannot rule out the possibility that certain lipids redistribute during cell disruption and organelle isolation, our proteomic and lipidomic data identify organelle-specific compositions that exclude a global mixing of organelle membranes or a broad equilibration of lipids during these procedures. Nevertheless, lateral specializations of the ER membrane such as sheets and tubules collapse during the preparation, as isolates via the Rtn1-bait and the Elo3-bait feature almost identical proteomes (Fig. EV1) and lipidomes (Fig. EV2). Based on these observations, we think that MemPrep reports rather on the "global" ER membrane composition, whereas ER subcompartments in vivo may establish local, specialized compositions.

Quantitative lipidomics demonstrates that the ER membrane has a remarkably high content of unsaturated fatty acyl chains (~74 mol%) and a low level of ergosterol (9.7 mol%). The resulting low degree of lipid packing is a crucial determinant of ER identity and renders it highly compressible (Bigay and Antonny, 2012; Holthuis and Menon, 2014; Renne and Ernst, 2023). Lipid packing in the ER membrane is continuously monitored by lipid saturation sensors (Covino et al, 2016; Ballweg et al, 2020) and actively maintained by the OLE pathway that regulates the production of unsaturated fatty acids and sterols (Hoppe et al, 2000; Rice et al,

2010). Maintaining ER membrane compressibility, on the other hand, is important to facilitate the insertion and folding of membrane proteins that differ substantially in their hydrophobic thicknesses and surface roughness depending on their subcellular distribution along the secretory pathway (Sharpe et al, 2010; Quiroga et al, 2013; Lorent et al, 2017, 2020). The machineries that insert and remove membrane proteins into and from the ER, respectively, induce a local membrane thinning, presumably to lower the energy barrier for insertion and extraction (Wu and Rapoport, 2021). Consequently, aberrantly high sterol levels or increased lipid saturation inhibit the insertion of transmembrane proteins in model membranes, the mammalian ER, and bacterial membranes (Nilsson et al, 2001; Brambillasca et al, 2005; Kamel et al, 2022). Therefore, it comes as no surprise that the compressibility and thickness of the ER membrane is continuously monitored by the UPR for regulating the relative rate of protein and lipid biosynthesis (Halbleib et al, 2017; Schuck et al, 2009; Covino et al, 2018).

Our lipidomic data are fully consistent with a gradual increase of lipid saturation along the secretory pathway (Van Meer et al, 2008; Bigay and Antonny, 2012), thereby complementing previous work on the composition of the trans-Golgi network/endosomal (TGN/E) system, secretory vesicles, and the plasma membrane in yeast (Klemm et al, 2009; Surma et al, 2011). Likewise, the ergosterol level of 9.7 mol % in the yeast ER membrane is consistent with previous estimations (Zinser and Daum, 1995; Schneiter et al, 1999; Van Meer et al, 2008) and parallels findings in mammalian cells, for which a resting cholesterol level in the ER between 7 and 8 mol% has been reported along with a switch-like regulation of sterol response element binding protein (SREBP) processing, when the level of cholesterol drops below 5 mol% in the ER (Radhakrishnan et al, 2008; Sokoya et al, 2022). From a cell biological viewpoint, our data suggest that the sterol gradient along the secretory pathway is rather flat from the ER (9.7 mol%) to the TGN/E system (9.8 mol%) (Klemm et al, 2009). If this is indeed the case, our findings support the view that active transport of sterols by lipid transfer proteins (Mesmin et al, 2013) aids lipid and protein sorting at the level of the TGN (Klemm et al, 2009). A flat sterol gradient in the early secretory pathway and a step-wise increase at the level of the TGN has important implications for the sorting of transmembrane proteins (Sharpe et al, 2010; Herzig et al, 2012; Quiroga et al, 2013; Lorent et al, 2017) and is consistent with recent models that favor sterol-enriched vesicular carriers (Borgese, 2016) or sterol-based, selective diffusion barriers for membrane proteins in the early secretory pathway (Weigel et al, 2021). This would be reminiscent of ceramide-based diffusion barriers for membrane proteins in the ER of dividing cells, which limits the access for 'old', potentially damaged membrane proteins to the daughter cell (Clay et al, 2014; Megyeri et al, 2019). Significant levels of complex sphingolipids in the ER, on the other hand, have previously been observed and are not surprising, even though the de novo biosynthesis of these lipids occurs in the Golgi apparatus (Hechtberger et al, 1994; Schneiter et al, 1999) (Fig. 2B). In fact, complex sphingolipids can be transported to the ER at substantial rates, in part for their degradation to ceramide by the phospholipase Isc1 in the context of a sphingolipid salvage pathway (Matmati and Hannun, 2008).

The lipid composition of the vacuole is vastly distinct from that of the plasma membrane (Surma et al, 2011) despite a substantial intake of membrane material via the endocytic route (Wendland

et al, 1998). The vacuolar membrane is also distinct from the ER membrane both in terms of lipid saturation (71 mol% unsaturated lipid acyl chains) and sterol content (11.7 mol%). It is possible that a tighter lipid packing in the vacuole membrane is required to reduce membrane permeability and to support vacuolar acidification. Most strikingly, however, is the near complete absence of PA lipids in the vacuole membrane (Fig. 6C), which is consistent with recent observations using alternative means of organelle isolation (González Montoro et al, 2018; Kim and Budin, 2024). Intriguingly, PA lipids are important signaling molecules in the ER membrane that regulate lipid biosynthesis by sensing the cytosolic pH, which in turn is crucially regulated by the vacuolar proton pump (Young et al, 2010; Hofbauer et al, 2018). The higher level of lysolipids observed in vacuolar versus ER membranes is consistent with the role of the vacuole as a lipid-degrading organelle (Henry et al, 2012). Notably, due to the large head-to-tail volume ratio, lysolipids exhibit large positive intrinsic curvature that would favor the formation of membrane defects (Ting et al, 2018). Hence, the high levels of the negatively curved DAG may be required to counterbalance undesired effects from lysolipids on membrane stability.

We have established and employed MemPrep to identify molecular fingerprints of lipid bilayer stress. While lipid metabolic changes of the ER membrane have been firmly associated with chronic ER stress (Hotamisligil, 2010), the underlying molecular changes remain understudied. We show that distinct conditions of lipid bilayer stress, namely inositol depletion and prolonged proteotoxic stresses, are associated with significant changes of the ER composition (Figs. 3 and 4) and identify an increased membrane thickness as a common denominator of the membrane-based UPR activation (Appendix Fig. S4D). This is in line with the prevailing model that Ire1 senses lipid bilayer stress via a hydrophobic mismatch-based mechanism (Halbleib et al, 2017). Our unbiased approach provides the surprising insight that a reduced abundance of anionic lipids in the ER membrane correlates with UPR activation. A general role of negatively charged lipids as attenuators of the UPR would have important physiological implications, because the level of PI and other anionic lipids change substantially in different growth stages (Casanovas et al, 2015). In fact, the availability of inositol is limiting for optimal growth of the commonly used strain BY4741 (Hancsho et al, 2012) and its prolonged cultivation in synthetic medium containing only 11 μM causes UPR activation even in the absence of exogenous stressors (Fig. EV4) when the cellular level of anionic PI lipids is low (Fig. 4A). Integrating information on the membrane composition and properties, either directly or indirectly, is crucial for Ire1 to orchestrate membrane biogenesis by balancing the production of proteins and lipids in different stages of growth (Covino et al, 2018). Now, it is possible to study the role of anionic lipids on Ire1 oligomerization by systematic reconstitution experiments using realistic membrane environments.

MemPrep also facilitates a direct comparison of different forms of lipid bilayer stress on the level of the proteome. Inositol depletion affects the abundance of only a handful of proteins, which are involved in handling either inositol or inositol-containing lipids (Fig. 3A). In stark contrast, DTT and TM cause a broader and more complex remodeling of the ER proteome (Figs. 5 and EV5D,E). This is not surprising as these potent proteotoxic agents can rapidly trigger the UPR and maintain UPR activity upon prolonged treatments via a membrane-based

mechanism (Promlek et al, 2011; Väth et al, 2021). Many proteins accumulating in the ER of DTT- or TM-stressed cells are known UPR targets, including ER chaperones, oxidoreductases, and lipid metabolic enzymes (Figs. 5 and EV5) (Travers et al, 2000; Jesch et al, 2006). While this accumulation likely reflects the duration of UPR activity, our proteomic analyses of the stressed ER are in line with the proposition that proteotoxic and lipid bilayer stress are associated with distinct transcriptional programs (Ho et al, 2020). Nevertheless, it is remarkable that DTT and TM, which establish somewhat similar but clearly distinct ER proteomes (Figs. 5 and EV5D,E), have virtually identical ER lipid fingerprints even down to the level of lipid acyl chains (Fig. 4C–E; Appendix Fig. S5).

Our quantitative data address an open question on the role of the PC-to-PE ratio in the ER as a potential driver of the UPR. PC and PE lipids are key determinants of the lateral pressure profile and curvature stress in cellular membranes, thereby affecting the conformational dynamics of membrane proteins (Marsh, 2007; van den Brink-van der Laan et al, 2004; Phillips et al, 2009). Aberrantly increased PC-to-PE ratios in the ER were suggested to cause chronic ER stress in obese mice (Fu et al, 2011), but the general validity of this interpretation has been controversially discussed (Gao et al, 2015). We employed MemPrep, quantitative lipidomics, and sensitive UPR assays to investigate this point in yeast. Prolonged proteotoxic stress is associated with a dramatically increased PC-to-PE ratio in the ER, which goes well beyond the range of physiological variation observed in cells at different growth stages (Klose et al, 2012; Janssen et al, 2000; Casanovas et al, 2015; Tran et al, 2019a). In contrast, artificially increasing the PC-to-PE ratio in the ER by supplementing choline to the medium is not sufficient to trigger the UPR (Fig. EV3). Inositol depletion, on the other hand, triggers a robust UPR without significantly perturbing the PC-to-PE ratio (Fig. EV3). Hence, an increased PC-to-PE ratio is not sufficient to activate the UPR in yeast. Instead, we favor the idea that a reduced PC-to-PE ratio and the accumulation of lipotoxic intermediates trigger the UPR by activating Ire1 either directly or indirectly (Ho et al, 2020; Ishiwata-Kimata et al, 2022). In summary, our quantitative analysis of the ER membrane composition isolated from stressed and metabolically challenged cells provides important insights to tackle mechanistic questions related to the chronic activation of the UPR.

Combining MemPrep with quantitative proteomics and lipidomics unlocks a toolbox to study the inter-organelle transport of both proteins and lipids. It is now feasible to investigate membrane protein targeting and sorting comprehensively and with organellar resolution. A variation of the MemPrep approach has been employed to establish a crucial role of the endoplasmic reticulum-mitochondria encounter structure (ERMES) for the delivery of mitochondrial preproteins via the ER surface to the mitochondrial import machinery (preprint: Koch et al, 2023). There are also fascinating examples of inter-organelle communication between the ER, lipid droplets, mitochondria, peroxisomes, and the vacuole in dealing with ER stress and lipotoxicity (Listenberger et al, 2003; Piccolis et al, 2019; Garcia et al, 2021). Here, the exchange of lipids between organelles provides a means to adapt to cellular stress and metabolic cues (Scorrano et al, 2019). While changes in lipid abundance can be readily identified by whole-cell lipidomics, it is not possible to study their redistribution in cells from one organelle to another unless individual organelle membranes can be isolated and analyzed. This is now becoming feasible. Naturally, it also possible in this context to combine gene deletions with organelle-specific bait strains for studying the role of a specific gene on the lipid composition of an individual organelle.

There are, however, also limitations to the MemPrep approach, which has been optimized for highest purity. The yields of the preparation are low, which can make the isolation of membranes from low abundant organelles less feasible. While MemPrep provides a comprehensive snapshot of a specific organelle membrane, the associated preparative efforts are significant, and the benefit of extensive time-course experiments should be carefully evaluated. Because lateral specializations of the ER membrane and membrane contact sites are disrupted during the preparation, the MemPrep approach is not suitable to isolate membrane contact sites or other lateral membrane specializations such as the outer or inner nuclear membrane. For these purposes, proximity labeling approaches and a selective solubilization of membrane proteins are promising developments (Kwak et al, 2020; van 't Klooster et al, 2020).

Elegant approaches have provided a first glimpse at the rate of lipid exchange between individual certain organelles (John Peter et al, 2022), but there is a great need for new preparative and analytical tools to obtain a comprehensive map of lipids at a given time. The combination of biosensors providing high spatial and temporal resolution with MemPrep, which provides quantitative and comprehensive snapshots of a given organelle at a certain time, surfaces as a promising approach to study membrane adaptation in a holistic fashion. We make MemPrep accessible to the community and have generated a collection of strains that facilitates the isolation of any organellar membrane of interest, as demonstrated for the ER and the vacuolar membrane.

## Methods

### Generation of MemPrep library

The C-terminus SWAp Tag (SWAT) library from yeast was used to generate a library with a C-terminal myc-HRV3C-3xFLAG tag as previously published (Meurer et al, 2018). In short, a SWAT donor strain (yMS2085) was transformed with a donor plasmid (pMS1134) containing the myc-HRV3C-3xFLAG cassette and then SWATted as described. The final library genotype is his3Δ1 leu2Δ0 met15Δ0 ura3Δ0, can1Δ::GAL1pr-SceI-NLS::STE2pr-SpHIS5, lyp1Δ::STE3pr-LEU2, XXX-L3-MYC-HRV3C-3xFLAG-ADH1ter-TEFpr-KANMXR-TEFter-L4)]. Once generated, to check proper integration of the cassette into the genome, random proteins were tested by polymerase chain reaction (PCR) and sodium dodecyl sulfate-polyacrylamide gel electrophoresis (SDS-PAGE), confirming their in-frame tagging.

### Manual generation of bait-tag strains

DNA sequences of HRV3C site-GSG and myc-GSG were introduced via primers in two consecutive steps between 6xGLY and 3xFLAG encoding regions on plasmid pFA6a-6xGLY-3xFLAG-kanMX6 (Funakoshi and Hochstrasser, 2009) (Addgene plasmid #20754) using Gibson assembly to yield pRE866. The DNA sequence of the

**Reagents and tools table**

| Reagent/resource | Reference or source | Identifier or catalog number |
| --- | --- | --- |
| **Experimental models** | | |
| BY4741 (*S. cerevisiae*) | Euroscarf | MATa *his3Δ1 leu2Δ0 met15Δ0 ura3Δ0* |
| Δ*IRE1* (*S. cerevisiae*) | Euroscarf | MATa *his3Δ1 leu2Δ0 met15Δ0 ura3Δ0 IRE1*::KANMX4 |
| Rtn1-bait, RESC000796 (*S. cerevisiae*) | This study | MATa *his3Δ1 leu2Δ0 met15Δ0 ura3Δ0 RTN1*-MYC-3C-3xFLAG-*KANMX6* + pRE512 |
| Elo3-bait, RESC000799 (*S. cerevisiae*) | This study | MATa *his3Δ1 leu2Δ0 met15Δ0 ura3Δ0 ELO3*-MYC-3C-3xFLAG-*KANMX6* |
| Vph1-bait, RESC000798 (*S. cerevisiae*) | This study | MATa *his3Δ1 leu2Δ0 met15Δ0 ura3Δ0 VPH1*-MYC-3C-3xFLAG-*KANMX6* |
| Rtn1-bait + ER-sfGFP-HDEL, RESC000797 (*S. cerevisiae*) | This study | MATa *his3Δ1 leu2Δ0 met15Δ0 ura3Δ0 RTN1*-MYC-3C-3xFLAG-*KANMX6* + pRE512 |
| **Recombinant DNA** | | |
| *Indicate species for genes and proteins when appropriate* | | |
| pRE866 | This study | pFA6a-KANMX6 containing the MYC-3C-3xFLAG tag for bait tagging |
| pRE512 | Eric Snapp | pRS415-ER-sfGFP-HDEL (ER-luminal marker) |
| **Antibodies** | | |
| Anti-FLAG, mouse monoclonal M2 | Sigma-Aldrich | F1804 |
| Anti-myc, mouse monoclonal 9E10 (1:1000) | Sigma-Aldrich | M4439 |
| Anti-Dpm1, mouse monoclonal 5C5A7 (1:2000) | Fisher Scientific | A6429 |
| Anti-Sec61, rabbit antiserum (1:1000) | Karin Römisch | |
| Anti-Kar2, rabbit polyclonal y-115 (1:1000) | Santa Cruz | sc-33630 |
| Anti-40kDa, rabbit antiserum (1:10,000) | Sepp Kohlwein | |
| Anti-Por1, rabbit antiserum (1:10,000) | Sepp Kohlwein | |
| Anti-Tlg2, rabbit antiserum (1:1000) | Susan Ferro-Novick | |
| Anti-Vac8, rabbit antiserum (1:10,000) | Christian Ungermann | |
| Anti-Ypt7, rabbit antiserum (1:1000) | Christian Ungermann | |
| Anti-Pep12, mouse monoclonal 2C3 (1:1000) | Thermo Fisher Scientific | |
| Anti-Gas1, rabbit antiserum (1:2000) | Howard Riezman | |
| Anti-Pdr5, rabbit antiserum (1:10,000) | Karl Kuchler | |
| Anti-Pcs60, rabbit antiserum (1:5000) | Ralf Erdmann | |
| Anti-Pex14, rabbit antiserum (1:2000) | Ralf Erdmann | |
| Anti-Vph1, mouse monoclonal (1:2000) | Abcam | ab113683 |
| Anti-rabbit, goat IRDye 680LT (1:15,000) | LI-COR | 926-68021 |

| Reagent/resource | Reference or source | Identifier or catalog number |
|---|---|---|
| Anti-rabbit, goat IRDye 800CW (1:15,000) | LI-COR | 926-32211 |
| Anti-mouse, goat IRDye 680LT (1:2000) | LI-COR | 926-68020 |
| Anti-mouse, goat IRDye 800CW (1:15,000) | LI-COR | 926-32210 |
| **Oligonucleotides and sequence-based reagents** | | |
| RE1012 (tag Rtn1 with bait from pRE866 fwd) | This study | GAAAAGTACAAAAAACTTGCAAATGAATTGGAAAAAAACAACGCTggggggaggcgggggtgga |
| RE1013 (tag Rtn1 with bait from pRE866 rev) | This study | CAAAAGTTAGCTATTCTTGTTTGAAATGAAAAAAAAAAAGCACTCAgaattcgagctcgtttaaac |
| RE1014 (tag Vph1 with bait from pRE866 fwd) | This study | GGAAGTCGCTGTTGCTAGTGCAAGCTCTTCCGCTTCAAGCggggggaggcgggggtgga |
| RE1015 (tag Vph1 with bait from pRE866 rev) | This study | GAAGTACTTAAATGTTTCGCTTTTTTTAAAAGTCCTCAAAATTTAgaattcgagctcgtttaaac |
| RE1018 (tag Elo3 with bait from pRE866 fwd) | This study | CTGGTGTCAAGACCTCTAACACCAAGGTCTCTTCCAGGAAAGCTggggggaggcgggggtgga |
| RE1019 (tag Elo3 with bait from pRE866 rev) | This study | CATTTAATTTTTTTCTTTTTCATTCGCTGTCAAAAATTCTCGCTTCCTATTTAgaattcgagctcgtttaaac |
| **Chemicals, enzymes, and other reagents** | | |
| NBD-PE, 1,2-dioleoyl-sn-glycero-3-phosphoethanolamine-N-(7-nitro-2-1,3-benzoxadiazol-4-yl) | Avanti Polar Lipids | 810145 |
| Rho-PE, 1,2-dioleoyl-sn-glycero-3-phosphoethanolamine-N-(lissamine rhodamine B sulfonyl) | Avanti Polar Lipids | 810150 |
| DPDG, 1,2-dipalmitoyl-sn-glycerol | Avanti Polar Lipids | 800816 |
| PODG, 1-palmitoyl-2-oleoyl-sn-glycerol | Avanti Polar Lipids | 800815 |
| DODG, 1–2-dioleoyl-sn-glycerol | Avanti Polar Lipids | 800811 |
| Erg, Ergosterol | Sigma-Aldrich | PHR1512 |
| 16:1-PE, 1,2-dipalmitoleoyl-sn-glycero-3-phosphoethanolamine | Avanti Polar Lipids | 850706 |
| POPE, 1-palmitoyl-2-oleoyl-sn-glycero-3-phosphoethanolamine | Avanti Polar Lipids | 850757 |
| 16:1-PC, 1,2-dipalmitoleoyl-sn-glycero-3-phosphocholine | Avanti Polar Lipids | 850358 |
| POPC, 1-palmitoyl-2-oleoyl-glycero-3-phosphocholine | Avanti Polar Lipids | 850457 |
| POPI, 1-palmitoyl-2-oleoyl-sn-glycero-3-phosphoinositol | Avanti Polar Lipids | 850142 |
| POPA, 1-palmitoyl-2-oleoyl-sn-glycero-3-phosphate | Avanti Polar Lipids | 840857 |

| Reagent/resource | Reference or source | Identifier or catalog number |
|---|---|---|
| DOPA, 1,2-dioleoyl-sn-glycero-3-phosphate | Avanti Polar Lipids | 840875 |
| POPS, 1-palmitoyl-2-oleoyl-sn-glycero-3-phospho-L-serine | Avanti Polar Lipids | 840034 |
| DOPS, 1,2-dioleoyl-sn-glycero-3-phospho-L-serine | Avanti Polar Lipids | 840035 |

bait-tag cassette (Appendix Supplementary Methods) was amplified by PCR from plasmid pRE866 using primers RE1012/RE1013, RE1018/RE1019, RE1014/RE1015 for tagging Rtn1, Elo3, and Vph1, respectively. Primers contain homologous regions for C-terminal tagging at the endogenous loci (*RTN1*, *ELO3*, *VPH1*). PCR products were used to transform wild-type yeast strain BY4741 by the lithium-acetate method (Ito et al, 1983).

## Fluorescence microscopy

In total, 3 µl of a yeast cell suspension ($OD_{600} = 50$), crude cell lysates or a fraction from the isolation procedure were placed on a thin agarose pad (1% agarose prepared in SCD) and then covered with a coverslip. Images were acquired on an Axio Observer Z1 equipped with a Rolera em-c2 camera (QImaging) and a Colibri 7 (Zeiss) light source for fluorescence excitation. Using either an EC Plan-Neofluar $100 \times /1.3$ or an EC Plan-Apochromat $63 \times /1.4$ objective in combination with a $1.6 \times$ tube lens (Zeiss), GFP fluorescence was excited using a 475-nm LED module and a 38 HE filter (Zeiss). Differential interference contrast (DIC) images were acquired using a translight LED light source. Image contrasts were adjusted linearly using Fiji (Schindelin et al, 2012).

## Immunofluorescence

Yeast cells for immunofluorescence were grown to $OD_{600} = 1.0$. 20 $OD_{600}$ units were fixed in 1 ml PBS containing 4% paraformaldehyde at RT for 2 h. Fixed cells were then washed with SP buffer (25 mM HEPES pH 7.4, 1 mM EDTA, 1.2 M sorbitol) and subsequently incubated in SP buffer containing 2% (v/v) 2-mercaptoethanol and 0.1 mg/ml Zymolyase 20 T (amsbio, 120491-1) at RT for 15 min to generate spheroblasts. 10 µl of washed spheroblast solution was placed on a round 10 mm coverslip, previously coated with poly-L-lysine (Sigma-Aldrich, P8920), and incubated at RT for 30 min. Cells were permeabilized by incubation in ice-cold methanol for 5 min and then acetone for 30 s. After brief air-drying, coverslips were immediately placed in 12-well plates and incubated with 1 ml of PBS-B (PBS, 1% BSA) for 30 min. Samples were then incubated with primary anti-FLAG (1:500) in PBS-BT (PBS, 1% BSA, 0.1% Tween-20) at RT for 1 h. After washing with PBS-B, samples were incubated with secondary anti-mouse IRDye 680LT (1:2000) in PBS-BT at RT for 1 h. After washing with PBS-B, PBS and rinsing in 95% ethanol, samples were mounted with 4 µl Fluoromount-G (Thermo Fisher Scientific) on glass slides. Images were acquired on an Axio Observer Z1 equipped with a Rolera em-c2 camera (QImaging) and a Colibri 7 (Zeiss) light source for fluorescence excitation. Using an EC Plan-Neofluar $100 \times /1.3$ objective, GFP and IRdye fluorescence were excited using a 475 nm and 630 nm LED module with filters 38 HE and 90 HE (Zeiss), respectively. Image contrasts were adjusted linearly using Fiji (Schindelin et al, 2012).

## Cell cultivation

Cells were cultivated at 30 °C in $SCD_{complete}$ medium (0.79 g/l complete supplement mixture (Formedium, batch no: FM0418/8363, FM0920/10437), 1.9 g/l yeast nitrogen base without amino acids and without ammonium sulfate (YNB) (Formedium, batch no: FM0A616/006763, FM0718/8627), 5 g/l ammonium sulfate (Carl Roth) and 20 g/l glucose (ACS, anhydrous, Carl Roth)) and constantly agitated by shaking the cultures at 220 rpm. Unless stated otherwise, overnight cultures (21 h) were used to inoculate a main culture to an $OD_{600}$ of 0.1. Cells were harvested by centrifugation ($3000 \times g$, 5 min, room temperature (RT)) at an $OD_{600}$ of 1.0, washed with 25 ml ice-cold phosphate-buffered saline (PBS), snap-frozen in liquid nitrogen, and stored at $-80$ °C until further use. In each case ER and vacuolar membranes were isolated from a total of 2000 and 4000 $OD_{600}$ units, respectively. This general procedure for cell cultivation and harvesting was also employed for stressed cells, with minor adaptations. For isolating the ER from cells before and after the induction of prolonged proteotoxic stresses, the cells were cultivated in the absence of stress to an $OD_{600}$ of 0.8 and a "pre-stress" sample was harvested. The residual culture was supplemented with either 2 mM dithiothreitol (DTT) or 1.5 µg/ml tunicamycin (TM) and cultivated for another 4 h prior to harvesting the cells. For isolating the ER from inositol-depleted cells, a first culture was inoculated to an $OD_{600}$ of 0.003 and cultivated overnight to an $OD_{600}$ of 1.2. Cells from this culture were pelleted ($3000 \times g$, 5 min, RT), washed twice with 100 ml pre-warmed inositol-free medium ($SCD_{complete-ino}$ prepared using yeast nitrogen base lacking inositol (YNB-ino) [Formedium batch no: FM0619/9431]). The washed cells were then resuspended in either inositol-containing $SCD_{complete}$ (control) or in $SCD_{complete-ino}$ (inositol depletion) medium to an $OD_{600}$ of 0.6 and cultivated for another 2 h prior to harvesting the cells.

## Cell lysis and differential centrifugation

Frozen cell pellets of 1000 $OD_{600}$ units were thawed on ice, resuspended in microsome preparation (MP) buffer (25 mM HEPES pH 7.0, 600 mM mannitol, 1 mM EDTA, 0.03 mg/ml protease inhibitor cocktail containing equal weights of pepstatin, antipain, chymotrypsin and 12.5 units/ml benzonase nuclease (Sigma-Aldrich)) and mechanically disrupted in 15 ml reaction tubes previously loaded with 13 g zirconia/silica beads (0.5 mm diameter, Carl Roth) using a FastPrep-24 bead beater (5 m/s, 10

cycles of 15 s beating and 45 s of cooling in an ice bath). Cell lysates were centrifuged twice ($3234 \times g$, 5 min, 4 °C) in a swinging bucket rotor to remove unbroken cells, cell debris and nuclei. The resulting post nuclear supernatant (PNS) was centrifuged ($12,000 \times g$, 20 min, 4 °C) in a Beckman type 70 Ti rotor to remove large organelle fragments. Using the same rotor, the resulting supernatant (S12) was centrifuged ($100,000 \times g$, 1 h, 4 °C) to obtain microsomes in the pellet. Microsomes were resuspended in 1 ml MP buffer per 1000 $OD_{600}$ units of original cell mass, snap-frozen in liquid nitrogen, and stored at −80 °C until further use. For subsequent proteomics analyses of samples from pre-stressed and stressed cells, microsomes were additionally resuspended in MP buffer containing 200 mM sodium carbonate (pH 10.6) and incubated rotating overhead at 3 rpm and 4 °C for 1 h to remove soluble and membrane-associated proteins. Carbonate-washed microsomes were neutralized by the addition of concentrated HCl, sedimented by ultracentrifugation ($100,000 \times g$, 1 h, 4 °C), resuspended in 1 ml MP buffer per 1000 $OD_{600}$ units original cell mass, snap-frozen in liquid nitrogen, and stored at −80 °C until further use.

## Immunoisolation

The entire isolation procedure was performed on ice or at 4 °C. Microsomes were thawed in 1.5-ml reaction tubes and then dissociated using a sonotrode (MS72) on a Bandelin Sonopuls HD 2070 with 50% power and 10 pulses of each 0.7 s (duty cycle 0.7). While sonication may affect the sidedness of ER-derived vesicles, this step was crucial for maximizing the purity of the preparation. After sonication, the microsomes were centrifuged ($3000 \times g$, 3 min, 4 °C). In total, 700 μl of the resulting supernatant (corresponding to 700 $OD_{600}$ units) were mixed with 700 μl immunoprecipitation (IP) buffer (25 mM HEPES pH 7.0, 150 mM NaCl, 1 mM EDTA) and loaded onto magnetic beads (dynabeads, protein G, 2.8 μm diameter, Invitrogen), which were previously decorated with sub-saturating quantities of a monoclonal anti-FLAG antibody (M2, F1804, Sigma-Aldrich). Specifically, the affinity matrix was prepared by using 800 μl of magnetic bead slurry per 700 $OD_{600}$ units of cells, which were incubated overnight at 4 °C with 3.2 μg of the anti-FLAG antibody using an overhead rotor at 20 rpm. Subsequently, microsomes were loaded on the antibody-decorated magnetic beads and allowed to bind for 2 h at 4 °C using an overhead rotor at 3 rpm.

The bound membrane vesicles were washed two times with 1.4 ml of wash buffer (25 mM HEPES pH 7.0, 75 mM NaCl, 600 mM urea, 1 mM EDTA) and twice with 1.4 ml of IP buffer. Specific elution was performed by resuspension of the affinity matrix in 700 μl elution buffer (PBS pH 7.4, 0.5 mM EDTA, 1 mM DTT, and 0.04 mg/ml affinity-purified GST-HRV3C protease) per 700 $OD_{600}$ units of original cell mass followed by incubation for 2 h at 4 °C on an overhead rotor at 3 rpm. The eluate was centrifuged ($264,360 \times g$, 2 h, 4 °C) in a Beckman TLA 100.3 rotor to harvest the ER- or vacuole-derived vesicles. The membrane pellet was resuspended in 200 μl PBS per 1400 $OD_{600}$ units of original cell mass (isolate), snap-frozen, and stored at −80 °C until lipid extraction and lipidomics analysis. For proteomics experiments, the membrane pellet was resuspended in 40 μl PBS-SDS (1%) per 1400 $OD_{600}$ units.

## Protein concentration determination, SDS-PAGE, and immunoblotting

Protein concentrations of all fractions from the immunoisolation were determined using the microBCA protein assay (Thermo Fisher Scientific #23235) following the manufacturer's recommendations. For SDS-PAGE, 1 volume membrane sample buffer (8 M urea, 0.1 M Tris-HCl, pH 6.8, 5 mM EDTA, 3.2% (w/v) SDS, 0.05% (w/v) bromphenol blue, 4% [v/v] glycerol, and 4% (v/v) β-mercaptoethanol) was mixed with two volumes of the immunoisolation fraction, incubated at 60 °C for 10 min, and loaded onto 4–15% mini-PROTEAN TGX precast protein gels (Bio-Rad). Proteins were separated at 185 V for 35 min. After separation, proteins were transferred by semi-dry Western blotting onto nitrocellulose membranes (Amersham Protran Premium 0.45 μm). The proteins of interest were detected using specific primary antibodies and fluorescent secondary antibodies (see "Reagents and tools table") on a fluorescence imager (LI-COR, Odyssey DLx). Signal intensities on immunoblots were quantified using ImageStudioLite.

## Liposome integrity and fusion assay

POPC-based, multilamellar liposomes containing 1 mol% 1,2-dioleoyl-sn-glycero-3-phosphoethanolamine-N-(7-nitro-2-1,3-benzoxadiazol-4-yl) (NBD-PE) and 1 mol% 1,2-dioleoyl-sn-glycero-3-phosphoethanolamine-N-(lissamine rhodamine B sulfonyl) (Rho-PE) as a FRET pair (Weber et al, 1998) were prepared by rehydrating dried lipids in MP buffer without protease inhibitors and benzonase (25 mM HEPES pH 7.0, 600 mM mannitol, 1 mM EDTA). Multilamellar liposomes were extruded through 0.4 μm and then 0.1-μm filters with each 21 strokes to yield large unilamellar vesicles (LUVs). LUVs were mixed with a 15.4 ± 1.3-fold excess of unlabeled P100 microsomes as quantified by a total phosphate determination after lipid extraction and acid hydrolysis based on classical protocols (Chen et al, 1956; Bligh and Dyer, 1959; Rouser et al, 1970). In brief, lipids from 100 μl of LUV or microsome solution were mixed with 100 μl MP buffer without protease inhibitors and benzonase (25 mM HEPES pH 7.0, 600 mM mannitol, 1 mM EDTA) and extracted in two steps by addition of first 750 μl chloroform:methanol (1:2) and then 750 μl chloroform and 250 μl ABC buffer (155 mM ammonium bicarbonate) at RT. The organic phase was recovered, and the solvent was evaporated using a centrifugal evaporator. Dried lipids were resuspended in 100 μl chloroform:methanol (1:2). 25 μl of this lipid extract was transferred to a pyrex glass tube, and the solvent was evaporated. The resulting lipid cake was treated with 300 μl 70% perchloric acid at 180 °C. Subsequently, 1 ml Milli-Q water, 0.4 ml of 5% (w/v) ascorbic acid, and 0.4 ml of 1.25% (w/v) ammonium molybdate were added and the resulting sample was boiled for 15 min. Absorbance at 797 nm was measured at RT after the sample cooled down. To test if sonication might potentially induce lipid exchange between liposomes and microsomes or even membrane fusion, the suspension was either sonicated for 10 s (Sonotrode MS72; Bandelin Sonopuls HD 2070 with 50% volume as 70% pulses) to match the MemPrep procedure or for 100 s to validate the impact of a much harsher treatment. As a positive control for lipid exchange, the suspension was mixed with 18 mM methyl-β-

cyclodextrin and incubated for 30 min at RT. To induce membrane fusion between the liposomes and microsomes as a control, we adjusted the mix to 40% (w/v) PEG 8000 and incubated for 30 min at RT. Full solubilization of the liposomes and the microsomes was achieved by adjusting the suspension to 1% SDS. Fluorescence emission spectra were recorded between 500 and 700 nm (step size of 2 nm) upon excitation at 467 nm using an Infinite 200 Pro (Tecan) plate reader. Bandwidths for the excitation and emission were set to 10 nm. Acceptor fluorescence upon donor excitation in the absence of a donor fluorophore (crosstalk) was determined using LUVs containing only the acceptor fluorophore (99 mol% POPC and 1 mol% Rho-PE) and subtracted from raw spectra of samples containing the FRET pair to yield background-corrected spectra. To aid spectral unmixing, we recorded fluorescence emission spectra of LUVs containing only the donor fluorophore (99 mol% POPC and 1 mol% NBD-PE) excited at 467 nm and of LUVs containing only the acceptor fluorophore (99 mol% POPC and 1 mol% Rho-PE) excited at 520 nm. Both, the background-corrected spectrum of the FRET pair and the two reference spectra were normalized by setting the area under the curve to 1 and decomposed using the software a|e by FluorTools to determine the relative contribution of the donor (NBD-PE) and the acceptor (Rho-PE) to the overall spectrum. Relative FRET efficiencies were calculated as $I_{Rho-PE}/(I_{NBD-PE}+I_{Rho-PE})$ with $I_{Rho-PE}$ being the acceptor and $I_{NBD-PE}$ the donor component after decomposition of the spectrum recorded from 500 to 700 nm.

## C-laurdan fluorescence spectroscopy

Lipid films were generated by drying lipid mixtures in chloroform under a stream of nitrogen at 50 °C and constant agitation at 400 rpm. Subsequently, residual chloroform was removed in an evacuated desiccator at RT for 1 h. Multilamellar liposomes were generated by the swelling method in liposome buffer (25 mM HEPES pH 7.0, 150 mM NaCl) at 1200 rpm shaking, 60 °C for 1 h. After sonication in a sonication bath for 20 min, the resulting multilamellar vesicles were snap-frozen in liquid nitrogen and stored at –80 °C. After thawing, the multilamellar vesicles were subjected to sonication using the sonotrode MS72 on a Bandelin Sonopuls HD 2070 with 50% power for 10 pulses of each 0.7 s. Liposomes were diluted (0.1 mM lipid) in liposome buffer and equilibrated in black 96-well plates in an Infinite 200 Pro (Tecan) plate reader at 30 °C. Background fluorescence emission spectra were recorded at 400–530 nm in 2 nm steps with 40 μs integration time, excitation at 375 nm and both bandwidths set to 10 nm. In total, 0.2 μM C-laurdan was added and incubated for 15 min before recording a fluorescence spectrum with the same settings. Generalized polarization was calculated as GP = $(I1 − I2) / (I1 + I2)$ with $I1$ being the background-corrected signal sum between 400 and 460 nm and $I2$ the background-corrected signal sum between 470 and 530 nm.

## Lipid extraction for mass spectrometry lipidomics

Mass spectrometry-based lipid analysis was performed by Lipotype GmbH (Dresden, Germany) as described (Ejsing et al, 2009; Klose et al, 2012). Lipids were extracted using a two-step chloroform/methanol procedure (Ejsing et al, 2009). Samples were spiked with internal lipid standard mixture containing: CDP-DAG 17:0/18:1, ceramide 18:1;2/17:0

(Cer), diacylglycerol 17:0/17:0 (DAG), lyso-phosphatidate 17:0 (LPA), lyso-phosphatidylcholine 12:0 (LPC), lyso-phosphatidylethanolamine 17:1 (LPE), lyso-phosphatidylinositol 17:1 (LPI), lyso-phosphatidylserine 17:1 (LPS), phosphatidate 17:0/14:1 (PA), phospha-tidylcholine 17:0/14:1 (PC), phosphatidylethanolamine 17:0/14:1 (PE), phosphatidylglycerol 17:0/14:1 (PG), phosphatidylinositol 17:0/14:1 (PI), phosphatidylserine 17:0/14:1 (PS), ergosterol ester 13:0 (EE), triacylgly-cerol 17:0/17:0/17:0 (TAG), stigmastatrienol, inositolphosphorylceramide 44:0;2 (IPC), mannosyl-inositolphosphorylceramide 44:0;2 (MIPC) and mannosyl-di-(inositolphosphoryl)ceramide 44:0;2 (M(IP)2C). After extraction, the organic phase was transferred to an infusion plate and dried in a speed vacuum concentrator. 1st step dry extract was resuspended in 7.5 mM ammonium acetate in chloroform/methanol/propanol (1:2:4, V:V:V) and 2nd step dry extract in 33% ethanol solution of methylamine in chloroform/methanol (0.003:5:1; V:V:V). All liquid handling steps were performed using Hamilton Robotics STARlet robotic platform with the Anti Droplet Control feature for organic solvents pipetting.

## Mass spectrometry (MS) data acquisition for lipidomics

Samples were analyzed by direct infusion on a QExactive mass spectrometer (Thermo Scientific) equipped with a TriVersa NanoMate ion source (Advion Biosciences). Samples were analyzed in both positive and negative ion modes with a resolution of $R_{m/z=200} = 280,000$ for MS and $R_{m/z=200} = 17,500$ for MSMS experiments, in a single acquisition. MS/MS was triggered by an inclusion list encompassing corresponding MS mass ranges scanned in 1 Da increments (Surma et al, 2015). Both MS and MSMS data were combined to monitor EE, DAG, and TAG ions as ammonium adducts; PC as an acetate adduct; and CL, PA, PE, PG, PI, and PS as deprotonated anions. MS only was used to monitor LPA, LPE, LPI, LPS, IPC, MIPC, M(IP)2C as deprotonated anions; Cer and LPC as acetate adducts and ergosterol as protonated ion of an acetylated derivative (Liebisch et al, 2006).

## Data analysis and post-processing for lipidomics

Data were analyzed with in-house developed lipid identification software based on LipidXplorer (Herzog et al, 2011; Herzog et al, 2012). Data post-processing and normalization were performed using an in-house developed data management system. Only lipid identifications with a signal-to-noise ratio >5, and a signal intensity fivefold higher than in corresponding blank samples were considered for further data analysis.

## Sample preparation for proteomics via liquid chromatography (LC)-MS/MS

Lysates were adjusted to 1% SDS and a final concentration 1 mg/ml. 5 μg of cell lysates and 10 μg of ER membrane were subjected to an in-solution tryptic digest using a modified version of the Single-Pot Solid-Phase-enhanced Sample Preparation (SP3) protocol (Hughes et al, 2014; Moggridge et al, 2018). In total, three biological replicates were prepared including control, wild-type and mutant-derived lysates ($n = 3$). Lysates were added to Sera-Mag Beads (Thermo Scientific, #4515-2105-050250, 6515-2105-050250) in 10 μl 15% formic acid and 30 μl of ethanol. Binding of proteins was achieved by shaking for 15 min at room temperature. SDS was removed by 4 subsequent washes with 200 μl of 70% ethanol. Proteins were

digested overnight at room temperature with 0.4 µg of sequencing grade modified trypsin (Promega, #V5111) in 40 µl HEPES/NaOH, pH 8.4 in the presence of 1.25 mM TCEP and 5 mM chloroacetamide (Sigma-Aldrich, #C0267). Beads were separated, washed with 10 µl of an aqueous solution of 2% DMSO and the combined eluates were dried down. Peptides of ER membranes were reconstituted in 10 µl of H₂O and reacted for 1 h at room temperature with 80 µg of TMT10plex (Thermo Scientific, #90111) (Werner et al, 2014) label reagent dissolved in 4 µl of acetonitrile. Peptides of cell lysates were reconstituted in 10 µl of H₂O and reacted for 1 h at room temperature with 50 µg of TMT16pro™ label reagent (Thermo Scientific, #A44521) dissolved in 4 µl of acetonitrile. Excess TMT reagent was quenched by the addition of 4 µl of an aqueous 5% hydroxylamine solution (Sigma, 438227). Peptides were reconstituted in 0.1% formic acid, mixed to achieve a 1:1 ratio across all TMT-channels and purified by a reverse phase clean-up step (OASIS HLB 96-well µElution Plate, Waters #186001828BA). Peptides were subjected to an off-line fractionation under high pH conditions (Hughes et al, 2014). The resulting 12 fractions were then analyzed by LC-MS/MS on an Orbitrap Fusion Lumos mass spectrometer.

## LC-MS/MS analysis of ER membranes and cell lysates

Peptides were separated using an Ultimate 3000 nano RSLC system (Dionex) equipped with a trapping cartridge (Precolumn C18 PepMap100, 5 mm, 300 µm i.d., 5 µm, 100 Å) and an analytical column (Acclaim PepMap100. 75 × 50 cm C18, 3 mm, 100 Å) connected to a nanospray-Flex ion source. The peptides were loaded onto the trap column at 30 µl per min using solvent A (0.1% formic acid) and, for analysis of ER membranes, eluted using a gradient from 2 to 38% Solvent B (0.1% formic acid in acetonitrile) over 52 min and then to 80% at 0.3 µl per min (all solvents were of LC-MS grade). Peptides for analysis of cell lysates were eluted using a gradient from 2 to 80% Solvent B over 2 h at 0.3 µl per min. The Orbitrap Fusion Lumos was operated in positive ion mode with a spray voltage of 2.4 kV and capillary temperature of 275 °C. Full-scan MS spectra with a mass range of 375–1500 $m/z$ were acquired in profile mode using a resolution of 60,000 or 120,000 for analysis of ER membranes or cell lysates respectively, with a maximum injection time of 50 ms, AGC operated in standard mode and a RF lens setting of 30%.

Fragmentation was triggered for 3 s cycle time for peptide-like features with charge states of 2–7 on the MS scan (data-dependent acquisition). Precursors were isolated using the quadrupole with a window of 0.7 $m/z$ and fragmented with a normalized collision energy of 36% for analysis of ER membranes (34% for analysis of cell lysates). Fragment mass spectra were acquired in profile mode and a resolution of 30,000 in profile mode. Maximum injection time was set to 94 ms or an AGC target of 200%. The dynamic exclusion was set to 60 s.

## Analysis of proteomics data

Acquired data were analyzed using IsobarQuant (Franken et al, 2015) and Mascot V2.4 (Matrix Science) using a reverse UniProt FASTA *Saccharomyces cerevisiae* database (UP000002311), including common contaminants and the following Rtn1-myc-3C-3xFLAG-tagged (bait) protein employed for the enrichment of subcellular membranes:

sp|P1707_RE | P1707_RE

MSASAQHSQAQQQQQQKSCNCDLLLWRNPVQTGKYFGG SLLALLILKKVNLITFFLKVAYTILFTTGSIEFVSKLFLGQGLIT-KYGPKECPNIAGFIKPHIDEALKQLPVFQAHIRKTVFAQVP KHTFKTAVALFLLHKFFSWFSIWTIVFVADIFTFTLPVIYHSY-KHEIDATVAQGVEISKQKTQEFSQMACEKTKPYLDKVESKLG-PISNLVKSKTAPVSSTAGPQTASTSKLAADVPLEPESKAYTSSA QVMPEVPQHEPSTTQEFNVDELSNELKKSTKNLQNELEKNNA GGGGGGEQKLISEEDLGSGLEVLFQGPGSGDYKDHDGDYKDH DIDYKDDDDK

The following modifications were considered: Carbamidomethyl (C, fixed), TMT10plex (K, fixed), Acetyl (N-term, variable), Oxidation (M, variable) and TMT10plex (N-term, variable). TMT16plex labeled samples The TMT16plex (K, fixed) and TMT16plex (N-term, variable) labels were considered as modifications. The mass error tolerance for full-scan MS spectra was set to 10 ppm and for MS/MS spectra to 0.02 Da. A maximum of two missed cleavages were allowed. A minimum of two unique peptides with a peptide length of at least seven amino acids and a false discovery rate below 0.01 were required on the peptide and protein level (Savitski et al, 2015).

## Enrichment calculation based on untargeted proteomics and clustering

For enrichment calculations, IsobarQuant output data were analyzed on a gene symbol level in R (https://www.R-project.org) using in-house data analysis pipelines. In brief, data were filtered to remove contaminants and proteins with less than two unique quantified peptide matches. Subsequently, log2 transformed protein reporter ion intensities ("signal sum" columns) were first cleaned for batch effects using the "removeBatchEffects" function from the limma package (Ritchie et al, 2015) and further normalized within the TMT set using the vsn package (Huber et al, 2002). UniProt annotations of subcellular location were retrieved from UniProt (accessed 27.01.2023).

For clustering analyses, proteins were tested for differential expression using the limma package. The replicate information was added as a factor in the design matrix given as an argument to the "lmFit" function of limma. A protein was annotated as a hit with a false discovery rate (fdr) smaller 5% and a fold change of at least 100% and as a candidate with a fdr below 20% and a fold change of at least 50%. Proteins classified as hits and candidates as well as a positive fold changes of the "pre" condition were clustered into nine clusters (method k-means) based on the Euclidean distance between normalized TMT reporter ion intensities divided by the median of the "pre-stress" condition.

## Molecular dynamics (MD) simulations

All-atom MD simulations were set up and carried out using the GROMACS software (Páll et al, 2020). Lipid topologies and structures were taken from the CHARMM-GUI web server (Jo et al, 2009). Bilayers were then generated using MemGen (Knight and Hub, 2015). Three different ER compositions were used (Appendix Fig. S4A) as well as a reference membrane composed of 50% POPC and 50% DOPC. Each system contained 100 lipids per leaflet and 60 water molecules per lipid. Na⁺ and Cl⁻ ions were added to reach an ionic concentration of 0.15 M. Taken together each system contained approximately 60000 atoms. Simulations

were carried out using the CHARMM36m forcefield (Huang et al, 2017) and the CHARMM-modified TIP3P water model (Jorgensen et al, 1983). The system was kept at 303 K using velocity recording (Bussi et al, 2007). Semi-isotropic pressure coupling at 1.0 bar was applied using the Berendsen barostat (Berendsen et al, 1984) for equilibration and the Parrinello-Rahman barostat using a coupling time constant of $\tau = 2\,ps$ (Parrinello and Rahman, 1981) during production runs. Electrostatic interactions were calculated using the particle-mesh Ewald method (Essmann et al, 1995). A cutoff of 1.2 nm according to CHARMM36 specifications (Best et al, 2012) was used for the non-bonded interactions, while the Lennard-Jones interactions were gradually switched off between 1.0 and 1.2 nm. Bond constraints involving hydrogen atoms were implemented using LINCS (Hess, 2008), thus a 2 fs time step was chosen. Each system was initially equilibrated for 50 ns followed by three independent production runs of at least 700 ns.

For the analysis, the first 100 ns of each production run was discarded. Mass density profiles were calculated along the $z$ axis (membrane normal) using the Gromacs module gmx density. The membrane thickness was extracted from the density profiles using a threshold of $500\,kg/m^3$. Errors were obtained by averaging and calculating the SEM over independent simulations segments. An in-house modified version of the Gromacs module freevolume was used to calculate the free volume profile as a function of $z$. The module estimates the accessible free volume by inserting probe spheres of radius $R$ at random positions, while testing the overlap with the Van der Waals radii of all simulated atoms. Here, we used a probe radius or $R = 0$, thereby obtaining the total free volume. Errors for each individual simulation were obtained by multiple independent insertion rounds carried out by gmx freevolume, the overall error for the averaged curves was calculated using standard error propagation. Surface packing defects were calculated using the program PackMem (Gautier et al, 2018). The program uses a grid-based approach to identify surface defects and distinguishes between deep and shallow defects based on the distance to the mean glycerol position (everything below a threshold of 1 Å was considered a deep defect). By fitting a single exponential to the obtained defect area distribution, the defect size constant was determined for each defect type. To achieve converged results, trajectory snapshots taken every 100 ps were used. Error estimation was conducted by block averaging dividing each simulation into three blocks of equal size, and calculating the SEM over all blocks.

## RNA preparation, cDNA synthesis, and RT-qPCR analysis

UPR activation was measured by determining the mRNA levels of spliced *HAC1*, *PDI*, and *KAR2*. For each experimental condition total RNA was prepared from five $OD_{600}$ units of cells using the RNeasy Plus RNA Isolation Kit (Qiagen). Synthesis of cDNA was performed using 500 ng of prepared RNA, Oligo(dT) primers, and the Superscript II RT protocol (Invitrogen). RT-qPCR was performed using the ORA qPCR Green ROX L Mix (HighQu) and a Mic qPCR cycler (Bio Molecular Systems) using a reaction volume of 20 µl. Primers were used at a final concentration of 400 nM to determine the Ct values of the housekeeping gene *TAF10* and the genes of interest: spliced *HAC1* forward: 5'-TACCTGCCGTAGA-CAACAAC-3'; spliced *HAC1* reverse: 5'-ACTGCGCTTCTGGATTAC-3'; *PDI* forward: 5'-TTCCCTCTATTTGCCATCCAC-3'; *PDI* reverse: 5'-GCCTTAGACTCCAACACGATC-3'; *KAR2* forward: 5'-TGCTGTCG TTACTGTTCCTG-3'; *KAR2* reverse: 5'-GATTTATCCAAACCG-TAGGCAATG-3'; *TAF10* forward: 5'-ATATTCCAGGATCAGGT CTTCCGTAGC-3'; *TAF10* reverse: 5'-GTAGTCTTCTCATTCTGTT-GATGTTGTTGTTG-3'.

The qPCR program consisted of the following steps: (1) 95 °C, 15 min; (2) 95 °C, 20 s; (3) 62 °C, 20 s; (4) 72 °C, 30 s; and (5) 72 °C, 5 min; steps 2–4 were repeated 40 times. We used the comparative $\Delta\Delta Ct$ method with normalization to the Ct values recorded for *TAF10* to assess the relative levels of the spliced *HAC1*, *PDI*, and *KAR2* mRNAs.

## Data availability

The mass spectrometry proteomics data from this publication have been deposited to the ProteomeXchange Consortium via the PRIDE [http://www.ebi.ac.uk/pride] partner repository with the dataset identifier PXD048553 (Perez-Riverol et al, 2022). Other datasets are included in the source data file.

## Peer review information

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

## Acknowledgements

The authors wish to thank Michael Knop for generously providing the C′ SWAT library. We would like to thank Sepp Kohlwein, Karin Römisch, Christian Ungermann, Howard Riezman, Karl Kuchler, and Ralf Erdmann for providing antibodies and Sarah L. Keller, Georg Pabst, Sebastian Schuck as well as Chris Stefan for fruitful discussions and helpful comments. We thank Cynthia Alsayyah for an informative control experiment. This work was funded by the VW foundation (Life?, #93089, #93092, #93090) to RE, MS, and JS, by the Deutsche Forschungsgemeinschaft in the framework of the SFB894 to RE and the SFB1027 to both JH and RE, and by the European Research Council under the European Union's Horizon 2020 research and innovation program (grant agreement no. 866011) to RE. MS is an incumbent of the Dr. Gilbert Omenn and Martha Darling Professorial Chair in Molecular Genetics.

## Author contributions

**John Reinhard**: Resources; Data curation; Formal analysis; Validation; Investigation; Visualization; Methodology; Writing—original draft; Writing—review and editing. **Leonhard Starke**: Data curation; Software; Formal analysis; Validation; Visualization; Methodology; Writing—review and editing. **Christian Klose**: Data curation; Formal analysis; Validation; Methodology. **Per Haberkant**: Resources; Data curation; Software; Validation; Visualization. **Henrik Hammarén**: Resources; Data curation; Software; Validation; Visualization. **Frank Stein**: Resources; Data curation; Software; Validation; Visualization. **Ofir Klein**: Resources; Investigation; Methodology. **Charlotte Berhorst**: Investigation; Methodology. **Heike Stumpf**: Investigation; Methodology. **James P Sáenz**: Funding acquisition; Investigation. **Jochen Hub**: Resources; Data curation; Software; Formal analysis; Supervision; Validation; Visualization; Writing—review and editing. **Maya Schuldiner**: Resources; Data curation; Investigation; Methodology; Writing—review and editing. **Robert Ernst**: Conceptualization; Data curation; Supervision; Funding acquisition; Validation; Visualization; Methodology; Writing—original draft; Project administration; Writing—review and editing.

## Funding

## Disclosure and competing interests statement

CK declares competing financial interests related to the publication of this study, including being chief technology officer at Lipotype GmbH, Dresden. The remaining authors declare no competing interests.

# Expanded View Figures

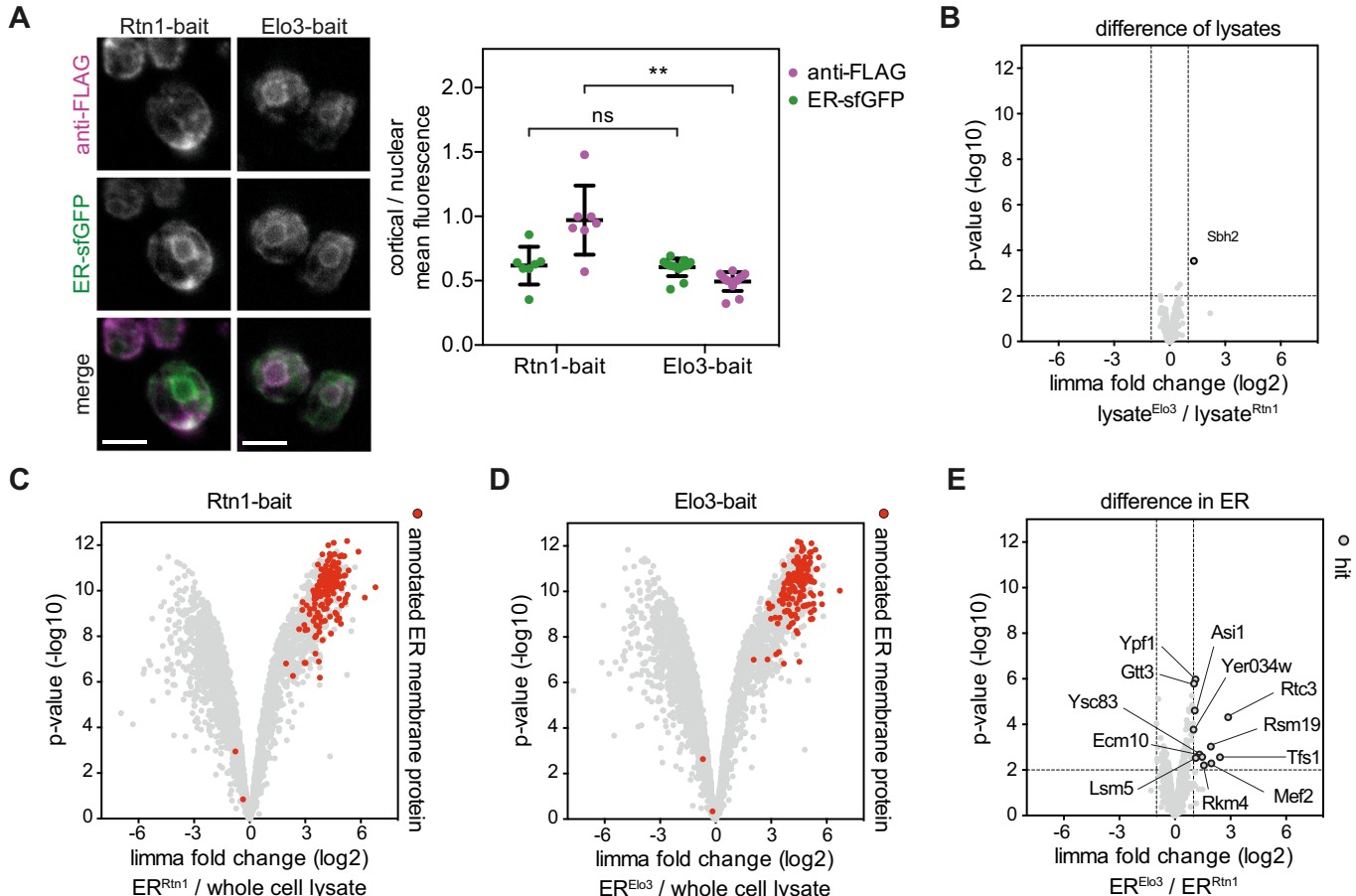

**Figure EV1. ER MemPrep via two different bait proteins.**

(A) Immunofluorescence showing the localization of two different ER membrane bait proteins (Rtn1-bait, Elo3-bait) relative to the ER-luminal marker ER-sfGFP-HDEL. Scale bar indicates 5 μm. Quantification of fluorescence distribution. Cell and nuclear areas were chosen manually. Cortical area was defined as total cellular area minus nuclear area. The ER-luminal marker ER-sfGFP-HDEL shows the same cortical-to-nuclear distribution in both bait strains. Rtn1-bait has a stronger preference for the cortical ER, compared to Elo3-bait. $n = 7$ cells for Rtn1-bait, $n = 14$ cells for Elo3-bait Data from individual cells are represented as data points yielding the average ± SD. $^{ns}P > 0.05$, $^{**}P ≤ 0.01$ (unpaired parametric $t$ test with Welch's correction). (B) Limma analysis of TMT-labeling proteomics reveals that the proteome of Rtn1-bait and Elo3-bait whole-cell lysates is identical except for a single outlier (Sbh2) ($n = 3$ biological replicates). (C) To increase the proteomics coverage for membrane proteins, P100 membranes were carbonate-washed before performing immunoisolation. MemPrep via Rtn1-bait enriches ER membrane proteins in the isolate (ER$^{Rtn1}$) ($n = 3$ biological replicates). (D) ER membrane proteins are enriched to the same extent by MemPrep via the bait protein Elo3-bait ($n = 3$ biological replicates). (E) MemPrep via Rtn1-bait and Elo3-bait yields almost identical sample composition with only 12 proteins that are enriched in the Elo3-bait derived ER. Data information: Data in (B–E) are presented as the mean from $n = 3$ biological replicates. A moderated t-test limma to test for differential enrichment was used. $P$ values were corrected for multiple testing with the method from Benjamini and Hochberg. Source data are available online for this figure.

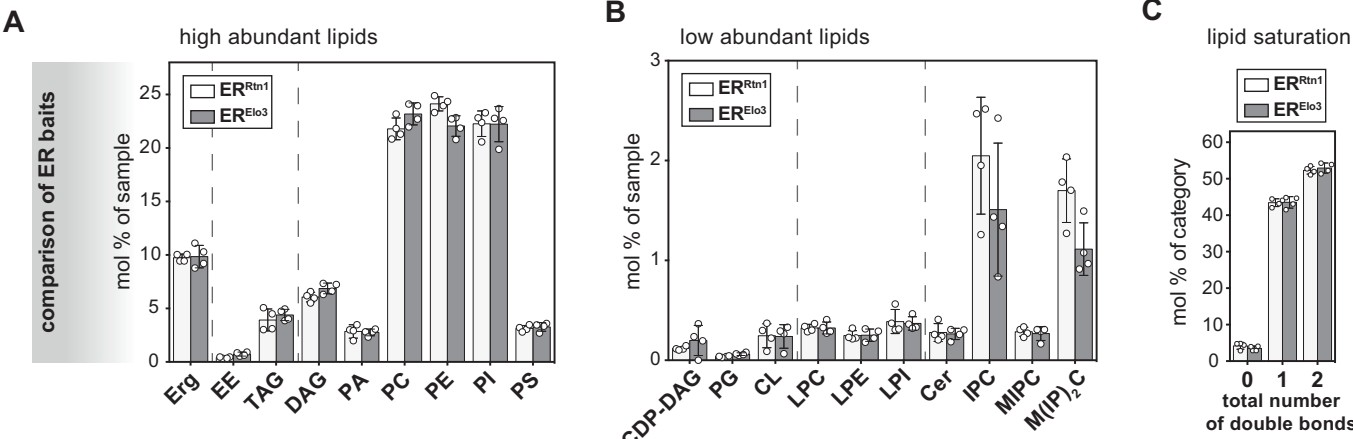

**Figure EV2.  The lipidome of Rtn1-bait and Elo3-bait derived ER membranes is identical.**

Quantitative lipidomics of ER membranes derived via two different bait proteins. (**A**) Distribution of lipid classes with high abundance. Erg ergosterol, EE ergosteryl ester, TAG triacylglycerol, DAG diacylglycerol, PA phosphatidic acid, PC phosphatidylcholine, PE phosphatidylethanolamine, PI phosphatidylinositol, PS phosphatidylserine (*n* = 4 biological replicates). (**B**) Distribution of lipid classes with low abundance. CDP-DAG cytidine diphosphate diacylglycerol, PG phosphatidylglycerol, CL cardiolipin, LPC lyso-phosphatidylcholine, LPE lyso-phosphatidylethanolamine, LPI lyso-phosphatidylinositol, Cer ceramide, IPC inositolphosphorylceramide, MIPC mannosyl-IPC, M(IP)$_2$C mannosyl-di-IPC (*n* = 4 biological replicates). (**C**) Total number of double bonds in membrane glycerolipids, except for CL, (i.e., CDP-DAG, DAG, PA, PC, PE, PG, PI, PS) as mol% of this category. Lipid data of Rtn1-bait derived membranes are identical with the data presented in Fig. 2A–C. Data information: In (**A–C**), data from *n* = 4 biological replicates are presented as individual data points and as mean ± SD. All differences of ER$^{Rtn1}$ versus ER$^{Elo3}$ were nonsignificant with *P* > 0.05 (multiple *t* tests, corrected for multiple comparisons using the method of Benjamini, Krieger and Yekutieli, with Q = 1%, without assuming consistent SD). Nonsignificant comparisons are not highlighted. Source data are available online for this figure.

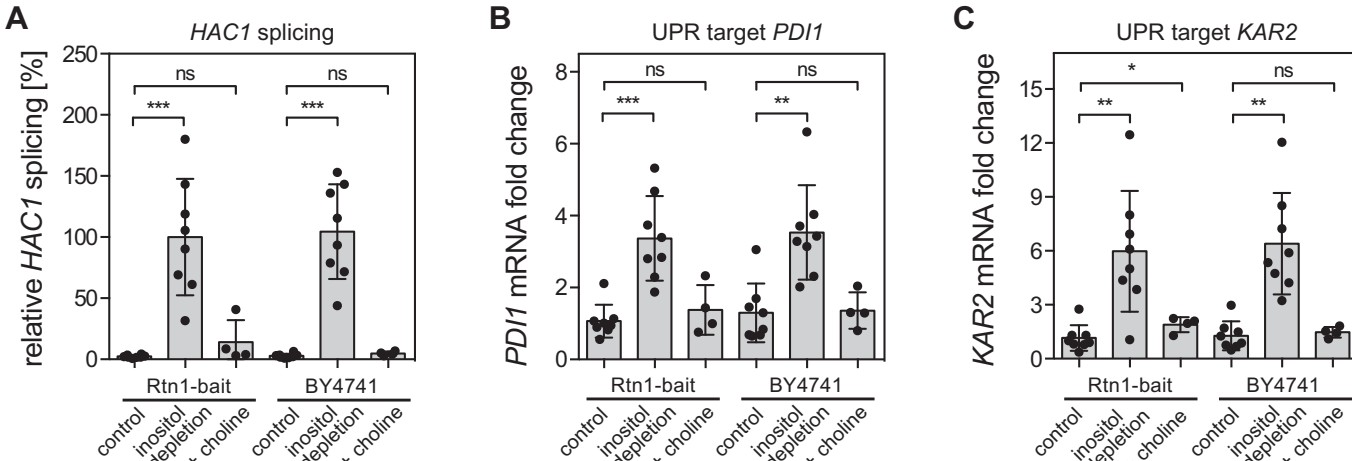

**Figure EV3. Activation of the UPR by lipid bilayer stress.**

SCD$_{complete}$ medium was inoculated with Rtn1-bait cells to an OD$_{600}$ of 0.003 from an overnight pre-culture and grown to an OD$_{600}$ of 1.2. Cells were washed with inositol-free medium and then cultivated for an additional 2 h in either inositol-free (inositol depletion) or SCD$_{complete}$ medium (control) starting with an OD$_{600}$ of 0.6. Another perturbation of lipid metabolism was achieved by addition of choline. For '+choline' conditions, SCD$_{complete}$ medium was inoculated to an OD$_{600}$ of 0.1 using stationary overnight cultures. Cells were then cultivated to an OD$_{600}$ of 1.0 in the presence of 2 mM choline. (**A**) UPR activation was measured by determining the levels of spliced *HAC1* mRNA. Data for relative *HAC1* splicing was normalized to the inositol depletion Rtn1-bait condition ($n = 8$ biological replicates based on two technical replicates for Rtn1-bait control, Rtn1-bait inositol depletion, BY4741 control and BY4741 inositol depletion, but $n = 4$ biological replicates based on two technical replicates for Rtn1-bait + choline and BY4741 + choline). (**B**) mRNA upregulation of the downstream UPR target gene *PDI*. *PDI* mRNA fold change was calculated as $2^{-\Delta\Delta Ct}$ and normalized to Rtn1-bait control condition ($n = 8$ biological replicates based on two technical replicates for the Rtn1-bait control, Rtn1-bait inositol depletion, BY4741 control, and BY4741 inositol depletion, but $n = 4$ biological replicates based on two technical replicates for Rtn1-bait + choline and BY4741 + choline). (**C**) Upregulation of mRNA of the downstream UPR target gene *KAR2* calculated as $2^{-\Delta\Delta Ct}$ and normalized to Rtn1-bait control condition ($n = 8$ biological replicates based each on two technical replicates for Rtn1-bait control, Rtn1-bait inositol depletion, BY4741 control, and BY4741 inositol depletion, but $n = 4$ biological replicates based on two technical replicates for Rtn1-bait + choline and BY4741 + choline). Data information: All data from biological replicates are presented in (**A–C**) as individual data points with the mean ± SD. $^{ns}P > 0.05$, $^*P \le 0.05$, $^{**}P \le 0.01$, $^{***}P \le 0.001$ (unpaired parametric $t$ test with Welch's correction). Nonsignificant comparisons are not highlighted. Source data are available online for this figure.

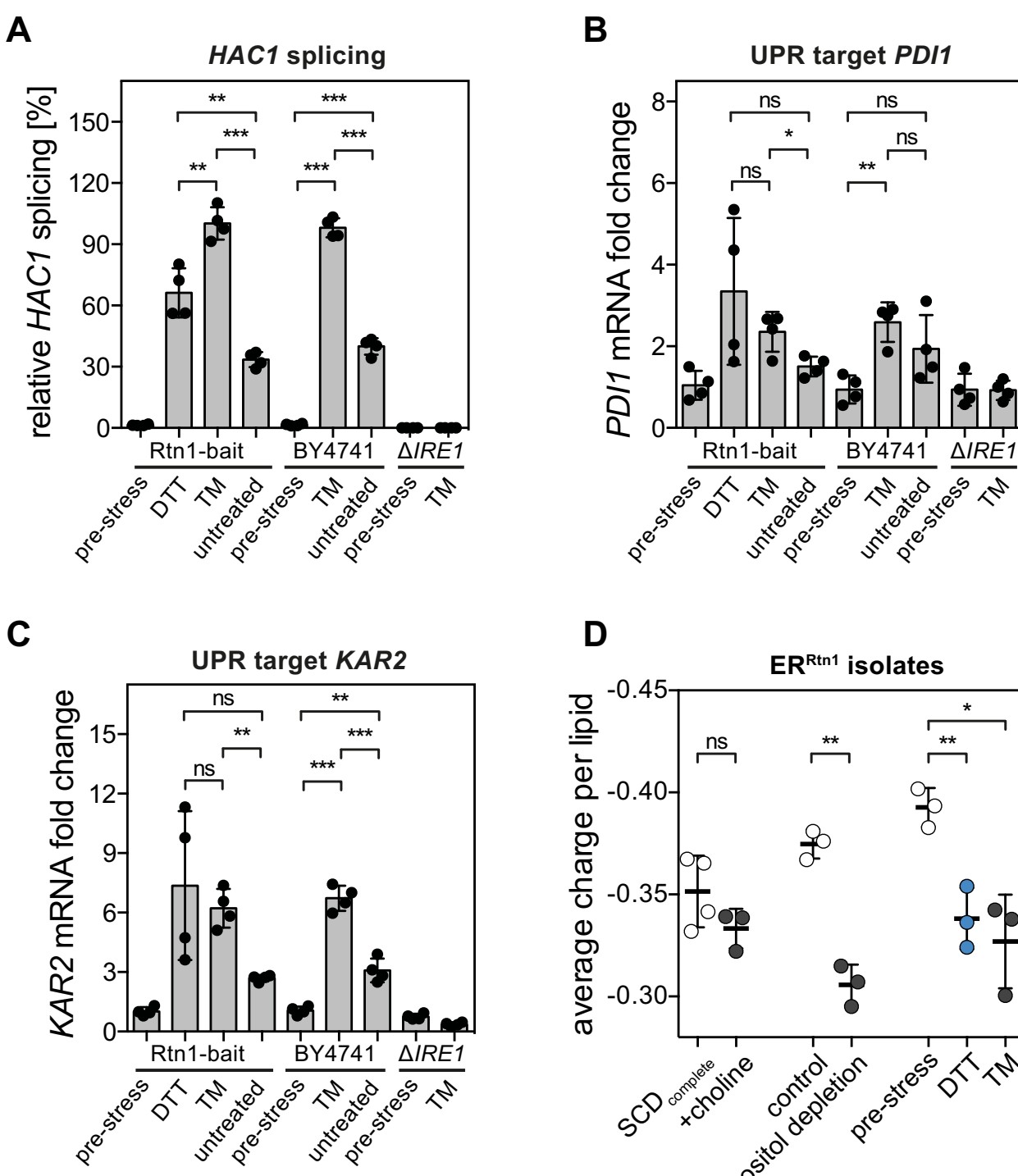

**Figure EV4. Activation of the UPR upon prolonged proteotoxic stress.**

Cells were grown as described above. UPR activation was measured by determining the levels of (A) spliced *HAC1* mRNA and the mRNA of the downstream UPR target gene ($n = 4$ biological replicates based on two technical replicates) (B) *PDI* ($n = 4$ biological replicates based on two technical replicates) and (C) *KAR2* before and after 4 h of DTT or TM treatment ($n = 4$ biological replicates based on two technical replicates). Data for relative *HAC1* splicing was normalized to the TM-treated Rtn1-bait condition. *PDI* and *KAR2* mRNA fold changes were calculated as $2^{-\Delta\Delta Ct}$ and normalized to Rtn1-bait pre-stress. (D) Calculation of the average charge per lipid from ER lipidomics data shown in Appendix Fig. S3A,B (SCD$_{complete}$, +choline), Fig. 3B,C (control, inositol depletion), and Fig. 4D,E (pre-stress, DTT, TM). Conditions with active UPR show reduced negative lipid charges compared to their respective controls. Net charges of the lipid classes were considered as follows: Erg 0, EE 0, TAG 0, DAG 0, PA -1, PC 0, PE 0, PI -1, PS -1, CDP-DAG -2, PG -1, CL -2, Cer 0, IPC -1, MIPC -1, M(IP)$_2$C-2 ($n = 4$ biological replicates for SCD$_{complete}$ and $n = 3$ biological replicates for all other conditions). Data information: All data are presented as individual data points and the mean ± SD. $^{ns}P > 0.05$, $^*P \leq 0.05$, $^{**}P \leq 0.01$, $^{***}P \leq 0.001$ (unpaired parametric $t$ test with Welch's correction). Source data are available online for this figure.

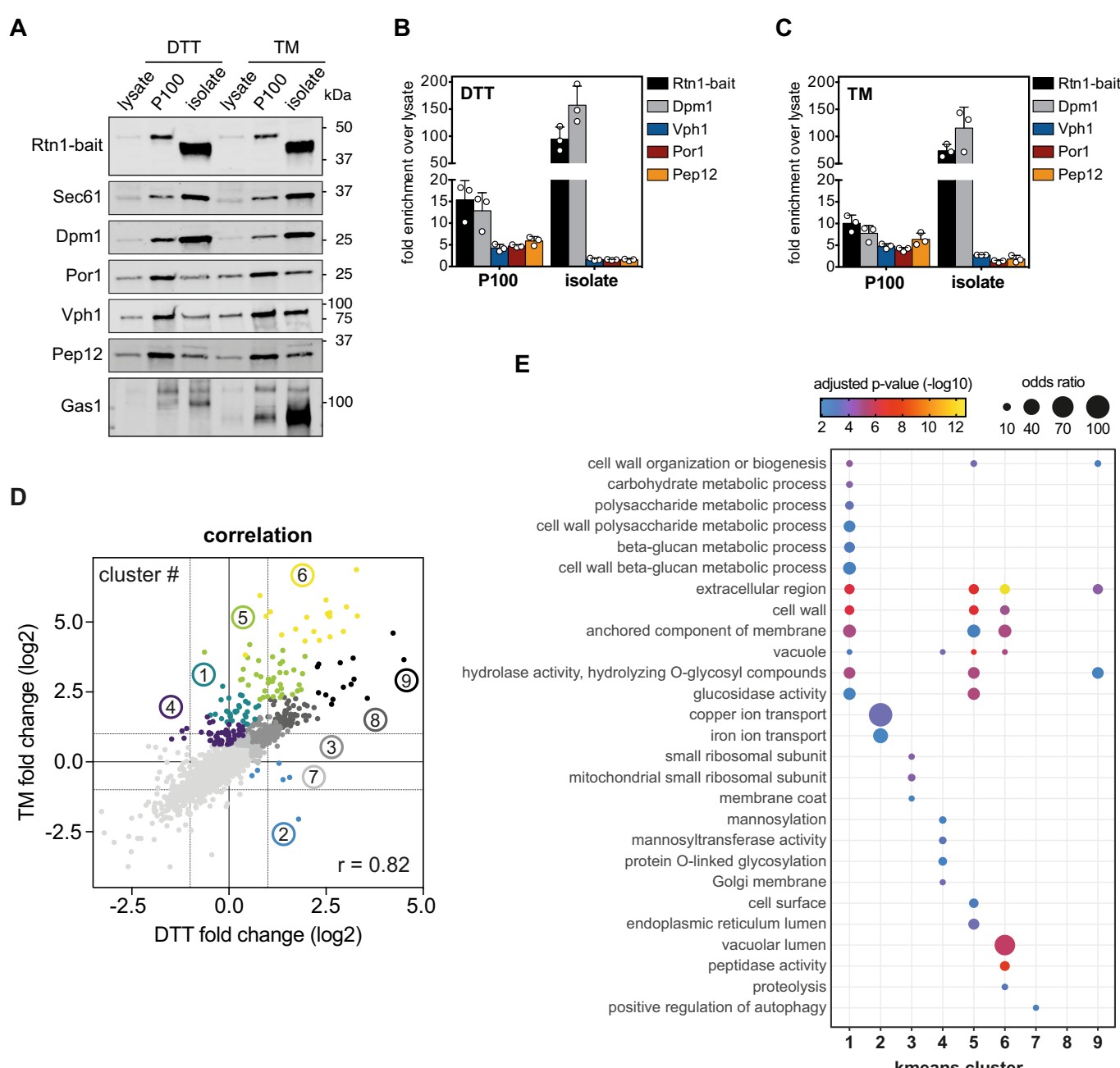

**Figure EV5. Enrichment of stressed ER membranes by MemPrep.**

(A) Immunoblot analysis of the indicated organellar markers in whole-cell lysates (lysate), crude membranes (P100), and MemPrep isolates (isolate). ER membranes were immuno-isolated via the Rtn1-bait protein. Sec61 and Dpm1 are prototypical ER membrane markers. Por1 is a marker for the outer mitochondrial membrane, Vph1 is a vacuolar marker. Pep12 marks endosomes and Gas1 serves as plasma membrane marker. 1 µg total protein loaded per lane. (B) Quantification of the organelle markers Dpm1, Vph1, Por1, Pep12 and the Rtn1-bait protein from three immunoblots of independent replicate ER MemPreps after prolonged proteotoxic stress induced by DTT ($n = 3$ biological replicates). Error bars indicate standard deviations ($n = 3$ biological replicates). (C) Quantification of three immunoblots from independent replicate ER MemPreps after prolonged proteotoxic stress induced by TM. Error bars indicate standard deviations ($n = 3$ biological replicates). (D) Correlation of DTT- and TM-induced fold changes, after Limma analysis, over pre-stress with a Pearson correlation coefficient $r = 0.82$. K-means clusters are indicated by colored groups and their respective cluster number ($n = 3$ biological replicates). (E) Gene ontology term enrichments in K-means clusters ($n = 3$ biological replicates). Data information: Data in (B, C), data from three biological replicates are presented as individual data points and as the mean ± SD. Data in (E) from $n = 3$ biological replicates are presented as the mean. $P$ values were derived from a Fisher-test and corrected for multiple testing with the method of Benjamini and Hochberg. Source data are available online for this figure.

