## [Peer Review File · The EMBO Journal]

MemPrep, a new technology for isolating organellar membranes provides fingerprints of lipid bilayer stress

John Reinhard, Leonhard Starke, Christian Klose, Per Haberkant, Henrik Hammarén, Frank Stein, Ofir Klein, Charlotte Berhorst, Heike Stumpf, James Sáenz, Jochen Hub, Maya Schuldiner, and Robert Ernst

Corresponding author(s): Robert Ernst (robert.ernst@uks.eu)

Review Timeline:

Submission Date:	14th Sep 22
Editorial Decision:	4th Nov 22
Appeal Received:	10th Nov 22
Editorial Decision:	24th Nov 22
Revision Received:	8th Jan 24
Editorial Decision:	9th Feb 24
Revision Received:	16th Feb 24
Accepted:	26th Feb 24

Editor: Hartmut Vodermaier

Transaction Report:

Prof. Robert Ernst
Saarland University
Medical Biochemistry and Molecular Biology
Kirrberger Str. 100
Building 61.4
Homburg, Saarland 66421
Germany

4th Nov 2022

Re: EMBOJ-2022-112605

A new technology for isolating organellar membranes provides fingerprints of lipid bilayer stress

Dear Robert,

Thank you again for submitting your manuscript on development and utilization of the MemPrep method as a resource article for The EMBO Journal. I apologize for the delay in getting back to you with a response, due to initial difficulties in receiving referee responses, and subsequently to somewhat equivocal feedback that necessitated further discussions within the editorial team as well as consultations with additional outside advisors of the journal. Based on the combined input and all these considerations, we unfortunately decided not to offer publication in The EMBO Journal.

As you will see from the three referee reports below, the reviewers acknowledge the applied state-of-the-art technology and the overall quality of the analyses, as well as the substantial amount of work and data presented in the manuscript. Nevertheless, they do raise several potential caveats, e.g. concerning the effects mediated by inositol, DTT or tunicamycin, but also regarding both the specific and the broader applicability of the method, and the purity/identity of the enriched ER fractions - issues that are particularly highlighted by the most critical reviewer 3. Still, during our post-review referee cross-consultation session, also referee 2 now concurred that despite the more efficient ER membrane protein enrichment compared to previous protocols, the general methodological advance is not sufficiently major to warrant publication in a broad general journal; agreeing in particular with the raised key concern regarding the fidelity of the derived membrane fragments and the mixed membrane vesicle populations likely derived from the employed cell disruption method. In light of these reservations, we decided to consult an additional expert advisor in the field of yeast organelle work, for an independent assessment of the methodological advance and potential applicability of MemPrep. It turned out that also this advisor independently concluded that membrane fractionation and affinity isolation of sonicated membrane fragments have been applied previously and cannot per se be considered a major novel development; and that the sonication step may either completely mix or separate the multiple functional domains of the ER, leaving it unclear whether the isolated fractions can be considered representative of the entire ER or constituting a subfraction. Having discussed all this input once more within our team, including my colleague William Teale, we unfortunately had to conclude that the present manuscript may not be a strong-enough candidate for an EMBO Journal resource article.

That said, given the improvements in ER enrichment and potential for further development/extension, as well as the field-specific value of the derived proteomics and lipidomics datasets, I realize that a revised manuscript clarifying some of the specific queries of the referees, and including more critical interpretations, should nevertheless be well-suited for publication in our open-access partner journal, Life Science Alliance (life-science-alliance.org). I already discussed the study and the reviews with the Executive Editor of Life Science Alliance, Dr. Eric Sawey, who would be interested in publication of this work. Revision for Life Science Alliance should incorporate responses to and discussions of the various specific questions raised in all three reports, as well as critical discussion of the methodological advances and of potential persisting caveats. At this stage, my Life Science Alliance colleagues would in essence need a careful (tentative) point-by-point response to the included referee reports, on the basis of which they would further discuss the revision process for Life Science Alliance with you.

If you are interested in this option, please follow the link below for transfer. Eric (e.sawey@life-science-alliance.org) will be happy to answer any questions you may have. Thank you once more for the opportunity to consider this work, and I am sorry that the reports did not allow me to be more positive for The EMBO Journal on this occasion.

With kind regards,

Hartmut

Referee #1:

In this manuscript submitted to EMBO Journal (EMBOJ-2022-1126050), Reinhard et al. describes a new, forward-thinking technology known as MemPrep, for isolating different organellar membranes in yeast. The authors thoroughly vetted the purification process and took careful consideration in obtaining a pure organellar fraction via use of quantitative proteomics and immunoblotting with specific organellar markers. The authors demonstrated that the isolated organellar membranes can be used in a variety of downstream applications such as lipidomics and proteomics in different stress conditions; all contributing to a comprehensive understanding of how cells adapt their lipid composition at the organellar level. Many studies are emerging which demonstrates how membrane protein function is influenced by the lipid environment and vice versa. Thus, this technology is significant and can be broadly appreciated in many fields including protein quality control, protein translocation, lipid homeostasis, and stress. Overall, this manuscript is well-written and presented in logical fashion and the quality of the data is sufficient to make arguments convincing. This paper presents a new technology, which will be an excellent addition to many fields. To improve the manuscript, I have included suggestions below, which the scientific community could benefit from, if they were using MemPrep.

Experimental concerns:

1. Because this is a technology paper, it would be beneficial for the community if you have a section discussing limitations of this approach, if any. For example, what is the criteria for choosing an ideal bait protein? Do they have to be abundant like the reticulon used in this study? Perhaps you have tested several bait proteins for different organelles already? Would be nice to state what they are (if not, no need to add). Moreover, many in the field are interested in separating ER vs nuclear membrane or smooth ER vs. rough ER. Will be useful, to discuss limitations for separating membranes that appear contiguous.
2. In regards to smooth ER vs. Golgi ER statement above, by pulling down Rtn1-baited ER (localized mainly in smooth ER) if you look at other ER proteins enriched in Fig. 1E, do you also see enrichment of smooth vs. rough ER membrane proteins? If so, would be invaluable to add this analysis to the manuscript. Of course, this isn't needed if you don't see a difference and can be included in your discussion within the "technical limitation" section.
3. For Figure 1D, was a nuclear membrane protein marker used as another control? Alternatively, in Figure 1E, it appears nuclear membrane proteins are slightly depleted in the purified ER fraction. I suggest stating the fold difference in the results section to make it clear to the readers.
4. It is very interesting that DTT and TM treatment leads to similar changes in lipid composition and yet has different impact on the ER proteome. Was DTT and tunicamycin treated at different times other than 4 hours? Would be interesting to see if 2 hour or longer time points led to distinct lipid composition in DTT-treated cells vs. tunicamycin treated cells. If this was done, can just state as data not shown.

Text/Figure concerns:

1. Line 51, should state Ire1 is from *S. cerevisiae* here as opposed to sentence below(line 54).
2. Line 371, add comma after "In the past"
3. For Figure 3D, I don't see difference in number of double bonds between control and inositol depletion conditions. Please clarify as this difference was stated in line 556.

Referee #2:

In this work, the authors describe a method to enrich ER-derived membranes and vacuolar membranes from microsomes prepared from cell extracts and differential centrifugation. The method depends on expression of a set of tagged ER membrane-associated proteins for which membrane fragments bearing the tagged protein can be enriched based on trapping the tagged protein. The isolated membranes were analyzed for protein and lipid contents by state-of-the-art proteomics and lipidomics methodologies. The data supported the conclusion that the ER membranes were enriched to a degree never achieved in previous reports making it possible to assign lipid composition of organelles in a more definitive manner. Moreover, the new method permitted analyses to examine the correlation between lipid composition and membrane protein insertion, folding, and sorting in response to stress and phospholipid precursor availability. The work is expertly performed and appropriately quantified. The "MemPrep" method described here, along with the data sets generated, will be of great value to the yeast community with interests in organelle isolation and the metabolic processes occurring therein. Minor criticisms of the work and manuscript are noted below.

Comments:

1. The inositol-mediated effects on lipid synthesis and the unfolded protein response pathways may be compromised by the fact

that the inositol concentration in the synthetic media is ~ 11 micromolar. Studies by the Henry laboratory in the 1980s and 1990s have shown that the regulatory effects of inositol occurs at concentrations > 50 micromolar. Perhaps the authors supplemented their growth media with inositol, but I didn't see this. In future studies, I suggest supplementing the inositol-free media with 50-75 micromolar inositol for comparing effects of this precursor. Additionally, the effects of choline on lipid synthesis are dependent on the higher concentration of inositol. This information is found in the primary literature cited in the review (Henry SA, Kohlwein SD & Carman GM (2012) Metabolism and regulation of glycerolipids in the yeast *Saccharomyces cerevisiae*. Genetics 190: 317-49) that is cited in this manuscript. If I'm correct about this issue, I don't request repeating the experiments, but instead, discuss this caveat.

Parenthetically, I'm disappointed that many do not read nor cite the primary literature.

2. The membrane enrichment protocol is dependent on strains expressing organelle-specific tagged proteins. While the authors are willing to make the strains freely available, depositing the strains in widely available repository is advisable for the long term.

3. Investigators will want to isolate highly enriched membrane fractions from cells with specific mutations. Have the authors introduced any mutations in the strains expressing organelle-specific tagged proteins. This might be mentioned in the text.

4. On lines 44 and 45, the authors cite preprints which have not been peer reviewed. I hope that this journal does not condone this practice.

5. On line 656, the authors use the much overused phrase "It is tempting to speculate", and then they go on to make the speculation. Just state "we speculate" that

Referee #3:

This manuscript reports an improved protocol "MemPrep" for purifying yeast ER that is based on the immune isolation reported by Klemm et al. (2009). The extensively optimized procedure yields an unprecedented enrichment of ER membranes as judged by immunoblot and untargeted protein mass spectrometry. The full lipidome of the purified ER fraction is presented. To identify ER compositions that trigger UPR, the effect of several conditions including prolonged proteotoxic stress, on the ER proteome and lipidome is analyzed. Based on the data, mixtures of synthetic lipids are proposed that mimic ER lipid composition under these conditions, to facilitate future mechanistic in vitro studies addressing UPR activation. To demonstrate broader application, MemPrep is used for establishing the lipidome of the yeast vacuole. The discussion is predominately speculative in line with the descriptive nature of the study.

The enormous effort and rigour invested in this work is impressive, as is the quality of the proteomics and lipidomics data presented. The data provide a useful resource for future research. However, this reviewer cannot help thinking that the authors got carried away by the wealth and quality of the data, because of the often biased interpretation of results.

Comments

1. The ER is generally considered a heterogeneous organelle comprising (specialized) subdomains such as SER, RER, tubular, sheet, perinuclear, cortical, PAM, MAM. Surprisingly, data are not discussed or interpreted in this context. Is the variation in ER marker enrichment factors (Fig S1C, S1D, Table S4) related to ER heterogeneity?

2. Is the purified ER fraction representative for "the ER"? Is the ER fraction that ends up in P12 (p.13) expected to be similar to the purified ER fraction? Although the prime goal was purity, the protein yield of the procedure remains an important parameter in organelle isolation. Is it possible to estimate the yield of pure ER from the cell lysate based on marker analysis?

3. Does the sonication step in the procedure affect the sidedness of ER vesicles? A FRET assay is used to exclude the occurrence of extensive organelle fusion or lipid exchange during sonication. The results (Fig S1B) do not convincingly support the conclusion. The assay conditions chosen may underestimate lipid exchange or fusion due to insufficient sensitivity. With PEG and Ca²⁺ present relative FRET efficiency is only decreased to 80%, indicative for a 1:1 dilution of the probes (Struck et al., 1981); with SDS added (concentration?) relative FRET is still 50%. Results suggest that the fold-excess of P100 over liposomal lipids is less than 8.2, which may originate from the peculiar scattering method used for quantifying liposomes and P100 that requires similar sizes of the vesicular structures to be compared. Quantitation of lipid content after extraction is more straight-forward and reliable.

4. The immuno-isolated highly purified ER prep is concluded to be largely devoid of cytosolic, inner nuclear membrane and mitochondrial markers (p.14). The significant enrichment of numerous markers of PM, vacuole, and Golgi in the ER fraction vs. P100 (Fig. 1E) and vs. cell lysate (Fig S1C, Table S4) goes unmentioned. Yet the contamination with these organelles most likely accounts for the unexpected finding of IPC, MIPC and M(IP)2C in the ER fraction. Accordingly, Fig 3B-control showing almost 2-fold less IPC and M(IP)2C than Fig 2B reflects the batch-dependence of the contamination level. The suggestion of substantial retrograde flux of complex sphingolipids from the Golgi complex to the ER (p.4, p.14-15, p.22) is far-fetched and lacks experimental support. Likewise, the small but significant amounts of cardiolipin (and PG) in pure ER fractions (0.2-0.3% CL in Fig. 2B; 0.5% in Fig. 3B; 0.6% in Fig. 4B) are in agreement with batch-dependent enrichment of mitochondrial markers in the ER fraction vs. cell lysate (Fig S1C). It is recommended to insert an additional Y-axis in Fig S1C (0 to 30-fold) for the non-ER ontology genes (and provide it with a legend that properly explains the presentation of the data). "The near perfect correlation of lipid abundances reported in four independent experiments" (Fig 2D p.15) may turn out less perfect if lipid species are summed

per class.

5. The interpretation of the inositol depletion data (Fig 3, p.16) ignores the existence of lipid turnover. PI is known to exhibit rapid turnover compared to other lipid classes including the sphingolipids. Therefore the statement "This implies a strict prioritization for sphingolipid biosynthesis over PI synthesis when inositol becomes limiting" is not justified. The inositol depletion-induced increase of PA confirms a vast body of literature data, since it is a key feature of the transcriptional regulation of phospholipid biosynthesis genes by Opi1-Ino2/4. Note that the increase of MIPC vs. M(IP)2C may reflect the reduced PI level (Jesch et al JBC 2010).

6. The induction of bilayer stress by prolonged exposure to DTT or TM reduces the growth rate (p.17). Please show the corresponding data. The response of the ER lipidome, i.e. decreases in PE and PI, increased acyl length and decreased desaturation, is reminiscent of the changes occurring in cell lysates of yeast cells entering stationary phase (co-author Klose et al PLOS One 2012; p.23), where UPR is not activated. To what extent do the molecular fingerprints of ER bilayer stress reflect slower growth or growth arrest? Did the authors consider UPR triggers other than specific, static ER lipid compositions, e.g. rates of changing lipid composition or desynchronization of protein and lipid synthesis? The proposed (un-)stressed ER-like lipid compositions may be preliminary.

7. To demonstrate the purity of the immune-purified vacuolar fraction in the absence of protein MS data, a blot analyzing a fixed amount of total protein per fraction (as in Fig 1B) is more revealing and informative than the blot showing 0.2% of each fraction (Fig 6A).

8. The methods section and legends need extensive proofreading to make sure that experimental details and sources of materials are correct and complete. Several methods are currently missing, including the construction of the tag, protein concentration assay, gel electrophoresis and immune-blotting, quantitation of blots, K-means clustering, as well as source and genotype of strains and plasmids used.

Other remarks

a. Provide legends for Tables S3, 4, 5 on a separate Excel sheet, detailing the experimental conditions (S3) and the parameters shown (S4, 5).

b. Does ER-stress affect purity of the ER fractions obtained (e.g. due to an increase of MCS)? TableS5 includes a.o. contaminating mitochondrial proteins, which may provide insight.

c. The clarity of the description of the untargeted protein MS procedures on p.8-9 could be improved by referring to the two datasets produced (Table S4 and S5).

d. Tables S4 and S5, and Fig S2D-F are not referred to in the Results section.

e. Explain the membrane topology of (tagged) Rtn1.

f. P.12 states "small organelle fragments are less likely to form contacts ..." What is this based on? One could argue that increasing the number of fragments promotes chances of interaction.

g. In Fig 1C, the eluate also seems to contain GFP-negative vesicles. What is the conclusion regarding loss of luminal proteins (Kar2 is also lost in Fig 1D)? What is the size of the scale bar? Does it apply to all 6 panels?

h. Lipid vesicles obtained by extrusion using 0.2um filters are denoted as large unilamellar vesicles (LUV) (p.13).

i. Include panels depicting average acyl chain length or length distribution in Figs 2, 3, S2, S3, S6.

j. Cki1 phosphorylates choline at the expense of ATP (Fig S3D).

k. Lipidome of whole cell lysate in Fig 6 is identical to that in Fig 2. Was 1 culture used for isolating both ER and vacuole? If so, please indicate.

*** As a service to authors, The EMBO Journal offers the possibility to directly transfer declined manuscripts to another EMBO Press title (EMBO Reports, EMBO Molecular Medicine, Molecular Systems Biology) or to the open access journal Life Science Alliance launched in partnership between EMBO Press, Rockefeller University Press and Cold Spring Harbor Laboratory Press. The full manuscript (including reviewer comments, where applicable and if chosen) will be automatically forwarded to the receiving journal, to allow for fast handling and a prompt decision on your manuscript. For more details of this service, and to transfer your manuscript to another EMBO title please follow this link:

Link Not Available

Response to reviewers
Manuscript EMBOJ-2022-112605

We thank all reviewers and the editor for their efforts in identifying strengths of our manuscript and points that can be improved/clarified. I am convinced that by addressing the reviewer's concerns and questions we will improve the manuscript significantly.

Referee #1:

In this manuscript submitted to EMBO Journal (EMBOJ-2022-1126050), Reinhard et al. describes a new, forward-thinking technology known as MemPrep, for isolating different organellar membranes in yeast. The authors thoroughly vetted the purification process and took careful consideration in obtaining a pure organellar fraction via use of quantitative proteomics and immunoblotting with specific organellar markers. The authors demonstrated that the isolated organellar membranes can be used in a variety of downstream applications such as lipidomics and proteomics in different stress conditions; all contributing to a comprehensive understanding of how cells adapt their lipid composition at the organellar level. Many studies are emerging which demonstrates how membrane protein function is influenced by the lipid environment and vice versa. Thus, this technology is significant and can be broadly appreciated in many fields including protein quality control, protein translocation, lipid homeostasis, and stress. Overall, this manuscript is well-written and presented in logical fashion and the quality of the data is sufficient to make arguments convincing. This paper presents a new technology, which will be an excellent addition to many fields. To improve the manuscript, I have included suggestions below, which the scientific community could benefit from, if they were using MemPrep.

We thank the reviewer for highlighting the potential of this technique for a wide variety of downstream applications.

After sharing the MemPrep manuscript as a preprint on bioRxiv and presenting the isolation approach at the *EMBO workshop 'The endoplasmic reticulum: The master regulator of membrane trafficking'* we received many requests for collaborations from leading labs focusing on various aspects of protein quality control, the ERAD-system, organelle biology, lipid metabolism, and membrane biophysics.

For example, we got requests from **Pedro Carvalho** (Dunn School of Pathology, Oxford), **Chris Stefan** (LMCB, London), **Maria Moriel-Carretero** (Université de Montpellier), and **Alwin Köhler** (Max Perutz Laboratories, Vienna). We also have a long-established collaboration with the membrane biophysicist **Sarah Keller** (University of Washington, Seattle). We shared our protocols and data with the teams of **Christian Ungermann** (University of Osnabrück) and **Johannes Herrmann** (Technical University Kaiserslautern).

This documents a **wide interest in this technique from scientists working in different fields**. I feel this is important point to consider, for the discussion as to whether this manuscript should be published in a specialized journal or rather a general-interest journal such as The EMBO Journal.

Experimental concerns:

1. Because this is a technology paper, it would be beneficial for the community if you have a section discussing limitations of this approach, if any. For example, what is the criteria for choosing an ideal bait protein? Do they have to be abundant like the reticulon used in this study? Perhaps you have tested several bait proteins for different organelles already? Would be nice to state what they are (if not, no need to add). Moreover, many in the field are interested in separating ER vs nuclear membrane or smooth ER vs. rough ER. Will be useful, to discuss limitations for separating membranes that appear contiguous.

The reviewer raises important points, which were also (in part) mentioned by reviewer 3 (points 1 and 2). We will provide an extended discussion of the limitations of the MemPrep strategy in the revised manuscript. Sharing our insights from several years of experience with the technique will be beneficial for all readers especially those interested in using this technique. For the revised manuscript, we will provide guidelines on 'how to choose an appropriate bait'.

What is a good bait protein? Shortly, a good bait protein is a true transmembrane protein. Peripheral proteins and proteins anchored to the membrane only by a single palmitoylation do not work. It is important that the C-terminal tag is accessible to the cytosol so that it is accessible for the anti-Flag antibodies after cell lysis. Naturally, there may be adaptations necessary to the protocol when aiming for analyzing inner nuclear membranes or the inner mitochondrial membrane.

Have we used different bait proteins? Yes. We isolated the ER via the Rtn1- and the Elo3-bait (see below). We can also isolate vacuolar membranes using two different bait proteins (Vph1 in this manuscript and Mam3 in a different manuscript, which is available on bioRxiv: [/doi.org/10.1101/2022.10.11.511736](https://doi.org/10.1101/2022.10.11.511736) and currently in revision at the Biophysical Journal). Notably, we can isolate vacuolar membranes from logarithmically growing cells and from cells in the stationary phase. We are convinced that the MemPrep strategy can be readily modified for isolating other organellar membranes. Nevertheless, targeting a new organelle does require some optimization as for any other isolation technique. Our experience from initial experiments on mitochondrial membranes and peroxisomal membranes is that the early steps of the procedure i.e. cell lysis and differential centrifugation deserve some special attention for finding an optimal solution.

Can we isolate ER-subdomains (e.g. ER vs nuclear membrane or smooth ER vs. rough ER)? At the moment, the answer is no. We have performed such an experiment as suggested by the reviewer. We isolated ER membranes using two different baits (Rtn1 and Elo3). Rtn1 is enriched in the peripheral, tubular ER. Elo3 is enriched in ER sheets (nuclear envelope). Both quantitative lipidomics and proteomics reveal almost identical composition of the isolates using these two bait proteins localizing to different ER subdomains.

Initially, this was a surprise and somewhat disappointing. **Yet, the fact that the MemPrep strategy reflects rather the composition of several subdomains than a single one is not a weakness. Instead, it opens the opportunity for a range of applications.** When combining the MemPrep approach with quantitative proteomics to study for example the substrate spectrum of a particular membrane protein insertase (or e.g. a particularly ERAD pathway), it is much easier to analyze the proteome of a sample, which reflects several (potentially all) ER subdomains rather than only a single one. Hence, this limitation of the MemPrep procedure opens a wide range of applications, by minimizing the number of required immuno-isolations and proteomic experiments, which are costly and time intensive. We would be happy to explain this also in the revised manuscript.

Why do we think is it currently impossible to isolate ER subdomains? We speculate that the ER architecture is disrupted prior to the immuno-isolation. We think, this happens either upon freezing/unfreezing of the cells after cultivation (which we used as a natural break-point) or in the process of cell disruption. This might cause the mixing of some (or all) ER subdomains. We will discuss this issue in the revised manuscript as a limitation and present possible workaround schemes.

What can be done to potentially make the isolation of subdomains via MemPrep possible? Here, we base our answer on the experiences gained from applying the MemPrep strategy to mammalian cells. We used HEK293T cells stably expressing a tagged variant of Reep5-bait as a bait. Reep5 (or DP1) localizes to the tubular ER. In this case, the cells were not frozen before they were subjected to the MemPrep procedure, and the cell lysis was much milder (using a dounce homogenizer in hypotonic buffers). Our preliminary proteomic analyses of immuno-isolated ER membranes using an overexpressed Reep5-bait (a protein that is found enriched in the tubular ER), we find a substantial enrichment of other proteins with such curvature preference such as Atlastin-2, RTN3, RTN4, and the endogenous, untagged Reep5 over other ER proteins normally found in the rough ER (not shown). While still preliminary, these data convinced us that the isolation of ER subdomains using a variation of the current protocol will be possible.

In summary, we are happy to include the new data shown in the Figure R1 along with a discussion on the limitations (and the resulting possibilities) of the overall strategy.

2. In regards to smooth ER vs. Golgi ER statement above, by pulling down Rtn1-baited ER (localized mainly in smooth ER) if you look at other ER proteins enriched in Fig. 1E, do you also see enrichment of smooth vs. rough ER membrane proteins? If so, would be invaluable to add this analysis to the manuscript. Of course, this isn't needed if you don't see a difference and can be included in your discussion within the "technical limitation" section.

See also the answer to point 1. We performed an experiment as suggested by the reviewer (Figure R1). Using our current protocol, we do not see a specific enrichment of ER proteins in any of the isolations based either on the Rtn1- or the Elo3-bait. We will discuss this in the 'technical limitation' section and will also discuss possible workaround schemes, which might make it possible to isolate ER subdomain (see above).

3. For Figure 1D, was a nuclear membrane protein marker used as another control? Alternatively, in Figure 1E, it appears nuclear membrane proteins are slightly depleted in the purified ER fraction. I suggest stating the fold difference in the results section to make it clear to the readers.

For immuno-blotting experiments, we have not used an inner nuclear membrane marker. Yet, our mass spectrometry-based experiments uses the abundance of a total 21 nuclear membrane markers with a unique GO-term annotation 'nuclear membrane' including NUP100, NUP1, NUP85, NUP120, NUP84; NUR1. These markers are enrichment an average between 1.0 and 1.2-fold. See Supplementary Table S1 for a complete list.

As suggested by the reviewer, we will state the fold difference in the results section and provide an updated Figure 1E, in which the fold change is directly indicated. For a further discussion, see our answer to reviewer 3, point 4.

4. It is very interesting that DTT and TM treatment leads to similar changes in lipid composition and yet has different impact on the ER proteome. Was DTT and tunicamycin treated at different times other than 4 hours? Would be interesting to see if 2 hour or longer time points led to distinct lipid composition in DTT-treated cells vs. tunicamycin treated cells. If this was done, can just state as data not shown.

We agree with the reviewer that these are important considerations. DTT and TM treatments had the same duration. The concentrations of DTT and TM were chosen to yield the same degree of growth defect for both DTT- and TM- treated cells as described in our previous publication (PMID: 32850859). In this publication we also studied the lipidome of cells, which were stressed for 1 h with the two proteotoxic agents.

Because **we and others have found that UPR activity peaks 4 hrs after addition of the proteotoxic agents**, we have decided to investigate this time-point. We will describe this point in greater detail in the revised manuscript.

We also want to take the opportunity to clarify an important point, which may have been misleading in the original manuscript.

We are interested in uncovering the membrane lipid compositions, which serve as activating signals for the UPR. We do not aim to establish how UPR signaling or proteotoxic agents, such as DTT or TM affect the cellular lipid composition. We also do claim that it is the DTT or TM treatment alone, which causes the changes of the lipid composition observed in Figure 4. As explained in the original manuscript, we use DTT and TM treatments merely as a tool to reach a point where the UPR via a membrane-based mechanism (PMID: 21775630; PMID: 34196665).

Furthermore, we have performed several additional whole cell lipidomic experiments to better dissect and clarify what is happening to the lipidome in DTT- and TM- stressed cells and what could be the underlying mechanism (**Figure R2**). We are happy to explain the implications of these findings in the revised manuscript.

The bottom line is that DTT- and TM-treatments are unlikely the sole reason for the complex lipid metabolic changes, which lead to the distinct lipid compositions observed for pre-stressed and stressed cells.

Text/Figure concerns:

1. Line 51, should state Ire1 is from *S. cerevisiae* here as opposed to sentence below(line 54).

Thank you. We corrected this for the revised manuscript.

2. Line 371, add comma after "In the past"

Corrected.

3. For Figure 3D, I don't see difference in number of double bonds between control and inositol depletion conditions. Please clarify as this difference was stated in line 556.

We changed the text to 'We further dissected the compositional changes of the ER membrane lipidome upon inositol-depletion at the level of the lipid acyl chains and observed a seemingly consistent trend through all lipid classes toward more saturated (Figure 3D, E) and shorter fatty acyl chains (Figure 3E) in glycerophospholipids.'

Referee #2:

In this work, the authors describe a method to enrich ER-derived membranes and vacuolar membranes from microsomes prepared from cell extracts and differential centrifugation. The method depends on expression of a set of tagged ER membrane-associated proteins for which membrane fragments bearing the tagged protein can be enriched based on trapping the tagged protein. The isolated membranes were analyzed for protein and lipid contents by state-of-the-art proteomics and lipidomics methodologies. The data supported the conclusion that the ER membranes were enriched to a degree never achieved in previous reports making it possible to assign lipid composition of organelles in a more definitive manner. Moreover, the new method permitted analyses to examine the correlation between lipid composition and membrane protein insertion, folding, and sorting in response to stress and phospholipid precursor availability. The work is expertly performed and appropriately quantified. The "MemPrep" method described here, along with the data sets generated, will be of great value to the yeast community with interests in organelle isolation and the metabolic processes occurring therein. Minor criticisms of the work and manuscript are noted below.

Comments:

1. The inositol-mediated effects on lipid synthesis and the unfolded protein response pathways may be compromised by the fact that the inositol concentration in the synthetic media is ~ 11 micromolar. Studies by the Henry laboratory in the 1980s and 1990s have shown that the regulatory effects of inositol occurs at concentrations > 50 micromolar.

We agree with the note the UAS-INO genes are fully repressed only at concentrations >50 μM . However, already 11 μM can substantially lower the *INO1* mRNA level (PMID: 3025587) and it is also sufficient to rescue inositol-auxotrophy mutants at 30°C. Our assay conditions were optimized for observing a rapid, full activation of the UPR upon removing inositol from the medium. Our medium contains ~11 μM inositol and it is not supplemented to 50-75 μM inositol.

Perhaps the authors supplemented their growth media with inositol, but I didn't see this. In future studies, I suggest supplementing the inositol-free media with 50-75 micromolar inositol for comparing effects of this precursor.

We thank the reviewer for this suggestion. In fact, we were discussing back-and-forth which medium and which additives to use at what concentration for this study. We will explain the rationale of our choice in the revised manuscript.

We are aware of the vast body of literature on the UAS-INO regulatory system. When working with the BY4741 strain (as we do it), it is particularly important to have a careful eye on the inositol concentration and the activity of the UAS-INO regulatory system. For the current manuscript, we have chosen 11 μM inositol to match previous experiments in the UPR field. We wanted to use conditions of which we knew from previous data by others and us (PMID: 34196665; PMID: 21775630) when the UPR activity peaks after inositol-depletion. We do this because we are interested in the lipid composition of the ER in this very moment. We want to learn more about which membrane features trigger the UPR.

Additionally, the effects of choline on lipid synthesis are dependent on the higher concentration of inositol. This information is found in the primary literature cited in the review (Henry SA, Kohlwein SD & Carman GM (2012) Metabolism and regulation of glycerolipids in

the yeast *Saccharomyces cerevisiae*. Genetics 190: 317-49) that is cited in this manuscript. If I'm correct about this issue, I don't request repeating the experiments, but instead, discuss this caveat.

We do not fully understand this point. When we add choline to the medium, we do see quite substantial changes in the lipidome (Figure S3). Showing that the addition of a lipid metabolite can remodel the ER membrane was the only purpose of our choline-feeding experiment.

Parenthetically, I'm disappointed that many do not read nor cite the primary literature.

We will do our best to refer better to the primary literature in the revised manuscript.

2. The membrane enrichment protocol is dependent on strains expressing organelle-specific tagged proteins. While the authors are willing to make the strains freely available, depositing the strains in widely available repository is advisable for the long term.

This is a very good point. We will carefully consider this option.

3. Investigators will want to isolate highly enriched membrane fractions from cells with specific mutations. Have the authors introduced any mutations in the strains expressing organelle-specific tagged proteins. This might be mentioned in the text.

The reviewer indicates an important application of MemPrep technology, which we will discuss in the extended section on 'limitations and potentials'. We have already introduced mutations in strains expressing organelle-specific tagged proteins and we are in the process to isolate the first organellar membranes for a separate manuscript.

4. On lines 44 and 45, the authors cite preprints which have not been peer reviewed. I hope that this journal does not condone this practice.

Formal citations of preprints are encouraged by The EMBO Journal.

5. On line 656, the authors use the much overused phrase "It is tempting to speculate", and then they go on to make the speculation. Just state "we speculate" that

We removed the phrase as suggested by the reviewer.

Referee #3:

This manuscript reports an improved protocol "MemPrep" for purifying yeast ER that is based on the immune isolation reported by Klemm et al. (2009). The extensively optimized procedure yields an unprecedented enrichment of ER membranes as judged by immunoblot and untargeted protein mass spectrometry. The full lipidome of the purified ER fraction is presented. To identify ER compositions that trigger UPR, the effect of several conditions including prolonged proteotoxic stress, on the ER proteome and lipidome is analyzed. Based on the data, mixtures of synthetic lipids are proposed that mimic ER lipid composition under these conditions, to facilitate future mechanistic in vitro studies addressing UPR activation. To demonstrate broader application, MemPrep is used for establishing the lipidome of the yeast vacuole. The discussion is predominately speculative in line with the descriptive nature of the study. The enormous effort and rigor invested in this work is impressive, as is the quality of the proteomics and lipidomics data presented. The data provide a useful resource for future

research. However, this reviewer cannot help thinking that the authors got carried away by the wealth and quality of the data, because of the often biased interpretation of results.

Comments

1. The ER is generally considered a heterogeneous organelle comprising (specialized) subdomains such as SER, RER, tubular, sheet, perinuclear, cortical, PAM, MAM. Surprisingly, data are not discussed or interpreted in this context. Is the variation in ER marker enrichment factors (Fig S1C, S1D, Table S4) related to ER heterogeneity?

See our answer to point 1 by reviewer 1. When we isolate the ER using two different bait proteins localizing to two different regions of the ER, we obtain almost identical lipid and protein compositions (**Figure R1; see above**). We will discuss the different subdomains of the ER in the revised manuscript and add additional data.

We have no indication that the variation in ER marker enrichment is due to the isolation of different ER subdomains. However, this is exactly what we can observe in our preliminary preparations of ER membranes from HEK293T cells using Reep5 as a bait protein. Here, we clearly separate tubular ER markers from non-tubular ER markers.

2. Is the purified ER fraction representative for "the ER"?

We think that the purified ER fraction is representative for "the ER membrane" because the composition of the isolate is identical even when isolated by different bait proteins, which normally reside in separate subdomains of the ER.

Is the ER fraction that ends up in P12 (p.13) expected to be similar to the purified ER fraction?

We would expect that. However, nobody can answer this question at the moment beyond any doubt. Because the organelle fragments in P12 correspond to large ER fragments, they should still be dominated by the tubular ER and ER sheets. Our immuno-isolations with the Rtn1- and Elo3-bait suggest that our immuno-isolations are representative to "the ER membrane".

Although the prime goal was purity, the protein yield of the procedure remains an important parameter in organelle isolation. Is it possible to estimate the yield of pure ER from the cell lysate based on marker analysis?

We isolate the ER from 2L of cells with an $OD_{600} = 1$ (2000 ODunits). Typically, 2000 ODunits of cells yield a total protein concentration of the lysate of ~83000 μg .

For an isolation from 2000 OD units via the Rtn1-bait, we obtained on average 31 μg isolated ER membrane.

For an isolation from 2000 OD units via the Elo3-bait, we obtained on average 14 μg isolated ER membrane.

Assuming that the ER accounts for ~20% of the total cell protein (PMID: 7645343), this would correspond to a yield of 1.9% for the ER isolated via the Rtn1-bait, and 0.84% for the Elo3-bait. We are happy to include this information to the discussion of 'limitations and potential' of the MemPrep procedure.

3. Does the sonication step in the procedure affect the sidedness of ER vesicles?

We do not fully understand the relevance of this question. If the sidedness of the ER vesicles would flip, the bait tag would not be accessible for the antibodies for the immuno-isolation.

A FRET assay is used to exclude the occurrence of extensive organelle fusion or lipid exchange during sonication. The results (Fig S1B) do not convincingly support the conclusion. The assay conditions chosen may underestimate lipid exchange or fusion due to insufficient sensitivity. With PEG and Ca²⁺ present relative FRET efficiency is only decreased to 80%, indicative for a 1:1 dilution of the probes (Struck et al., 1981); with SDS added (concentration?) relative FRET is still 50%. Results suggest that the fold-excess of P100 over liposomal lipids is less than 8.2, which may originate from the peculiar scattering method used for quantifying liposomes and P100 that requires similar sizes of the vesicular structures to be compared. Quantitation of lipid content after extraction is more straightforward and reliable.

We used the FRET assay to gain a semi-quantitative insight into whether there is substantial exchange of membrane materials from vesicles fusion or lipid exchange during sonication. The assay was sufficient in our eyes to exclude this possibility.

We apologize for having caused a confusion. As stated in the methods section of the original manuscript (but not as indicated in the Fig S1B), we used Triton X-100 (and not SDS) at a concentration of 1% in 250 μ l to dissolve the membranes. We are happy to redo the FRET experiments using optimized conditions to rule out any lipid exchange between the ER and other organelles during the sonication step.

Importantly, our interpretation that sonication does not neither induce membrane mixing nor any other substantial lipid exchange is strongly supported by the lipidome data (Figure 2A-C and S2D-F). The degree of lipid saturation in isolated ER membranes (Figure SC and S2F) differs substantially from the degree of lipid saturation in whole cells and in P100 microsomes.

4. The immuno-isolated highly purified ER prep is concluded to be largely devoid of cytosolic, inner nuclear membrane and mitochondrial markers (p.14). The significant enrichment of numerous markers of PM, vacuole, and Golgi in the ER fraction vs. P100 (Fig. 1E) and vs. cell lysate (Fig S1C, Table S4) goes unmentioned. Yet the contamination with these organelles most likely accounts for the unexpected finding of IPC, MIPC and M(IP)2C in the ER fraction.

We respectfully disagree. The statement that a contamination with these organelles 'most likely accounts to the unexpected finding of IPC, MIPC, and M(IP)2C in the ER fraction' is not correct. This misinterpretation is probably because the abundance of the ER membrane in the lysate is not considered.

In the revised manuscript, we indicate the average fold change of 178 ER proteins for each replicate, and also indicate the average fold change observed for other organellar markers (see updated Figure 1E).

The reviewer seems to argue that enrichments of the plasma membrane and other organellar markers of 3.0 to 5.4 indicate a significant contamination with these organelles, which in turn 'likely' account for the substantial quantities of sphingolipids, which we report in the ER. This is not correct.

Firstly, the reviewer does not seem to account that the proteomic experiment is normalized to the protein concentration in the lysate and that it is not possible to 'transfer' this type of normalization when trying to assess the degree of lipid contaminations. A five-fold enrichment of PM proteins in the final isolate does not mean that PM lipids are also five-fold enriched in the preparation. When pelleting membranes and washing away soluble, cytosolic proteins membrane proteins from all organelles are enriched when using the protein concentration for normalizing the data. However, when normalizing with lipids, the same experiment that reported on an enrichment of e.g. plasma proteins would show no enrichment of plasma membrane lipids.

Secondly, and probably more important: The reviewer does not seem to account for the important fact that the ER is the most abundant organelle in glucose-grown cells already in the initial cell lysate.

Given that the ER is already a highly abundant organelle, it cannot possibly be enriched as much as other organelles, such as the plasma membrane or the Golgi, which need to be enriched >100-fold in order to consider them as reasonably pure. The 23.3- to 29.6-fold enrichment of ER markers, which we report, is unprecedented. The enrichment of other marker of the endomembrane system between ~3.0 (for lipid droplets) and 5.4 for peroxisomes from individual replicates does not mean that the resulting lipidome is substantially affected by such very minor contamination.

To further underscore this point, let me directly quote from the seminal review 'Isolation and Biochemical Characterization of Organelles from the Yeast, *Saccharomyces cerevisiae*' by Zinser and Daum (PMID: 7645343).

"The following example will illustrate how cross-contamination of preparations is influenced by the abundance of organelles in the cell (Figure 1). In mitochondria isolated from aerobically grown yeast, a specific mitochondrial marker is six-fold enriched over the homogenate. In addition, markers of the plasma membrane and microsomes in this fraction are five- and three-fold enriched, respectively. Although all three markers exhibit enrichment factors of nearly the same magnitude, these results should be interpreted with caution when assessing the quality of the preparation. The first conclusion that can be made is that mitochondria are sufficiently enriched, because optimal enrichment factors of this organelle are in the range of 4 to 7. Secondly one can conclude that this fraction is only slightly contaminated with plasma membrane. This conclusion is based on the fact that the plasma membrane comprises only about 1-2% of the total protein of a yeast cell. Highly enriched plasma membrane exhibits an enrichment factor of 100. The third conclusion is that the preparation of mitochondria is, in contrast to the low contamination with plasma membrane, heavily contaminated with microsomes because optimal enrichment factors of 6 are typically observed for purified microsomal fractions.

When taking this consideration into account, it should be clear that **1) a ~25-fold enrichment of ER markers over the cell lysate demonstrates the unprecedented purity of our preparation, 2) the sample is not substantially contaminated by other organelles** (this is especially noteworthy for mitochondria, which are not as abundant as in the example by Zinser and Daum because the cells are grown on glucose and they are only enriched by a factor of maximally 1.3, while the ER is enriched ~25-fold, **and 3) a very minor contamination of the final isolate with other organelles cannot account for the level of sphingolipids observed in ER membrane preparations** (because the plasma membrane is enriched only 4.2-5.3-fold in individual replicates, while the ER is not only much more abundant, but also much more enriched).

I want to illustrate this further with a back-of-an-envelope calculation that assumes that IPC would be only found in the plasma membrane and not in other organelles. To account for the concentration of IPC, which we report in our ER preparations, the contaminating plasma membrane would have to contain the following molar concentration of IPC:

2 mol% (IPC concentration in our isolates) * 20 (to account for the lower abundance in the initial cell lysate * 25/5 (to account for the enrichment of the ER over the enrichment of the PM = 1000 mol%). Obviously, this is impossible because the limit is 100 mol%.

We have taken the purity of our samples very seriously. We will extend our discussion in the revised manuscript to avoid any confusion of the reader on this important point.

Accordingly, Fig 3B-control showing almost 2-fold less IPC and M(IP)2C than Fig 2B reflects the batch-dependence of the contamination level.

See our answer above.

Also: it is known that the level of sphingolipids can rapidly change during cellular cultivation. Because the control in Fig. 3B is based on a slightly different cultivation (for providing an optimal inositol-depletion experiment), the data in Fig 3B and Fig 2B cannot be expected to show the same lipid composition. We have no evidence that contaminating membranes make a significant contribution to any of our isolated ER membranes. To avoid any confusion of the reader, we will highlight the fact of the difference in cultivation between Fig 3B and Fig 2B even more clearly in the revised manuscript.

The suggestion of substantial retrograde flux of complex sphingolipids from the Golgi complex to the ER (p.4, p.14-15, p.22) is far-fetched and lacks experimental support.

We respectfully disagree and are happy to provide more information in the revised manuscript.

According to the yeast genome database, the estimated copy number of the HDEL receptor (Erd2) is $\sim 11,600 \pm 3,100$. The estimated total number of 12 ER-luminal, soluble HDEL proteins (Pdi1, Ero1, Kar2, Lhs1, Yos9, Mpd2, Cpr5, Eug1, Mpd1, Scj1, Jem1, Sil1) sums up to 108,400.

Unless these proteins are not sequestered and kept in the ER by other means, they must constantly be retrieved from 'leaking' into the secretory pathway. And because they outnumber the HDEL receptor by an order of magnitude, and because there is substantial forward traffic (*S. cerevisiae* must double all membranes of the secretory pathway in 90 min) which would make them 'leak' from the ER there must be a substantial retrograde traffic via the COP-I machinery. We are happy to explain our line of thoughts better in the revised manuscript.

Likewise, the small but significant amounts of cardiolipin (and PG) in pure ER fractions (0.2-0.3% CL in Fig. 2B; 0.5% in Fig. 3B; 0.6% in Fig. 4B) are in agreement with batch-dependent enrichment of mitochondrial markers in the ER fraction vs. cell lysate (Fig S1C).

We have no indication for strongly batch-dependent differences in the enrichments of mitochondrial markers. The difference between Figure 2B, 3B and 4B are more likely due to differences in the degree of cultivation.

Given our arguments above regarding the enrichment of the ER relative to mitochondria, we doubt that the possible, minimal contaminations of our preparation with mitochondrial membranes alone can be responsible for the concentrations of CL observed in our isolated ER membranes. We will discuss this point more clearly in the revised manuscript. However, given that (often polyspecific) lipid transfer proteins have been identified in virtually any subcellular membrane, we find highly likely that at least some leakage from inner

mitochondrial membrane to the ER can occur. We will provide a more balanced discussion regarding this point in our revised manuscript.

It is recommended to insert an additional Y-axis in Fig S1C (0 to 30-fold) for the non-ER ontology genes (and provide it with a legend that properly explains the presentation of the data).

We thank the reviewer for this point and have updated Figure S1C, which we will use in the revised manuscript. This motivated us to take another look at the 'unexpected outliers' of our proteomic analysis (labeled).

Tcb3, Lam4, Ist2, Erg25, Slp1, Sed4, Gyp8, Osh6 all have a GO term annotation 'ER' or 'ER membrane', but they have other annotations. Some of them have (Osh6, Tcb3, Ist2, Lam4) been firmly implicated in contact sites between the ER and other organelles.

Seg2 has the go term annotations plasma membrane, cell periphery.

Fmp27, Csf1 share the GO annotation 'mitochondrion'. **Fmp27** was recently renamed to Hob1 and located to ER-mitochondrial contact site (PMID: 36354737).

Csf1 was recently localized to ER-mitochondria contact site (PMID: 34415038; PMID: 35015055).

Hence, revisiting the proteomic data further underscores the quality of our preparations and indicate a remarkable predictive power to find even new ER membrane proteins.

"The near perfect correlation of lipid abundances reported in four independent experiments" (Fig 2D p.15) may turn out less perfect if lipid species are summed per class.

Following the reviewer's advice, we have summed up lipid species per class and replotted the graph. We still find a remarkable correlation between individual replicates both for high and low abundant lipid classes. We feel that both representations have their own justification and would consider of using both (correlation between replicates for species and classes) in the revised manuscript.

5. The interpretation of the inositol depletion data (Fig 3, p.16) ignores the existence of lipid turnover. PI is known to exhibit rapid turnover compared to other lipid classes including the sphingolipids. Therefore the statement "This implies a strict prioritization for sphingolipid biosynthesis over PI synthesis when inositol becomes limiting" is not justified. The inositol depletion-induced increase of PA confirms a vast body of literature data, since it is a key feature of the transcriptional regulation of phospholipid biosynthesis genes by Opi1-Ino2/4. Note that the increase of MIPC vs. M(IP)2C may reflect the reduced PI level (Jesch et al JBC 2010).

We will extend our discussion on the role of PI turnover and conversion to PIPs in the revised manuscript. Nevertheless, it is known *de novo* PI synthesis is dramatically reduced upon inositol depletion as evidenced by radiolabeling experiments by the Susan Henry lab (PMID: 16777854). Given that the sphingolipid levels are maintained in the ER membrane, we still find our statement justified even if some fraction of PI would be converted into PIPs (which are an order of magnitude less abundant in the ER).

We see that ER PI decreases whereas ER SLs remain relatively stable. This indicates that the cells maintain the SL levels in the ER: This is supported and consistent with *ORM2* being regulated by Ino2-Ino4.

6. The induction of bilayer stress by prolonged exposure to DTT or TM reduces the growth rate (p.17). Please show the corresponding data.

We have used optimized conditions, which we have worked out in a previous, open access publication (PMID: 32850859). We will refer to the impact of DTT or TM on the growth rate more prominently in the revised manuscript.

The response of the ER lipidome, i.e. decreases in PE and PI, increased acyl length and decreased desaturation, is reminiscent of the changes occurring in cell lysates of yeast cells entering stationary phase (co-author Klose et al PLOS One 2012; p.23), where UPR is not activated.

Indeed, there are some changes in the lipid composition, which are reminiscent of the changes occurring in whole cells (dominated by the ER) upon entering later growth stages. However, the statement that the UPR is not activated during this transition is not correct. In the transition from the logarithmic phase to the stationary phase the UPR is activated (PMID: 31484935). Our new whole cell lipidomic data (**Figure R2**) suggest that cells undergo a phase of acute inositol depletion when cultivated like DTT- and TM-treated cells, but in the absence of a proteotoxic agent. Similar observations were made by the Kimata laboratory and reproduced in laboratory.

We will include the new whole-cell lipidomic data and will rewrite the results section related to (Figure 4). We will explain that our prime motivation is understanding, which lipid compositions drive the UPR.

To what extent do the molecular fingerprints of ER bilayer stress reflect slower growth or growth arrest?

This answer is almost impossible given the complexity of the lipid metabolic network and the wide pleiotropic effects that DTT and TM have. In the revised manuscript, we will discuss this point by also referring to the new data shown in **Figure R2**.

Did the authors consider UPR triggers other than specific, static ER lipid compositions, e.g. rates of changing lipid composition or desynchronization of protein and lipid synthesis?

We did not consider a role of the rate of changing lipid compositions, because these rates are several orders of magnitude slower the protein-lipid interactions that control the activating of Ire1 by lipid bilayer stress.

Naturally, we have considered the role of a perturbed protein-to-lipid ratio as a driving force of the membrane-based UPR. In fact, we have published the idea that UPR transducers act as sensor for the protein-to-lipid ratio as an hypothesis (PMID: 30075144). The development of the MemPrep technology represent an important milestone in our efforts to learn how cells sense and control the protein-to-lipid ratio in cellular membranes. In fact, this work is funded by an ERC consolidator grant in the context of the research proposal MemDense.

The proposed (un-)stressed ER-like lipid compositions may be preliminary.

We do not understand this point.

7. To demonstrate the purity of the immune-purified vacuolar fraction in the absence of protein MS data, a blot analyzing a fixed amount of total protein per fraction (as in Fig 1B) is more revealing and informative than the blot showing 0.2% of each fraction (Fig 6A).

For the revised manuscript, we would perform a quantitative proteomics experiment to analyze the enrichment of vacuolar proteins in our preparation.

8. The methods section and legends need extensive proofreading to make sure that experimental details and sources of materials are correct and complete. Several methods are currently missing, including the construction of the tag, protein concentration assay, gel electrophoresis and immune-blotting, quantitation of blots, K-means clustering, as well as source and genotype of strains and plasmids used.

Following the reviewer's advice, we will proofread the experimental details and source of materials and control for completeness. We will add all missing information.

Other remarks

a. Provide legends for Tables S3, 4, 5 on a separate Excel sheet, detailing the experimental conditions (S3) and the parameters shown (S4, 5).

This will be done for the revised manuscript.

b. Does ER-stress affect purity of the ER fractions obtained (e.g. due to an increase of MCS)? TableS5 includes a.o. contaminating mitochondrial proteins, which may provide insight.

The possibility that ER-Mito contact sites are remodeled during ER stress would be consistent with the available literature. We can discuss this point in the revised manuscript.

c. The clarity of the description of the untargeted protein MS procedures on p.8-9 could be improved by referring to the two datasets produced (Table S4 and S5).

d. Tables S4 and S5, and Fig S2D-F are not referred to in the Results section.

We now refer to Supplementary Table S4, Supplementary Table S5, and Figure S2D-F in the results section.

e. Explain the membrane topology of (tagged) Rtn1.

Following the reviewer's advice, we describe the membrane topology of Rtn1 in the revised manuscript.

f. P.12 states "small organelle fragments are less likely to form contacts ..." What is this based on? One could argue that increasing the number of fragments promotes chances of interaction.

We based this statement on the fact that only a certain fraction of the ER surface is intact with other organelles. The bigger an ER fragment the more likely it is that somewhere in this ER fragment there is a contact sites to a different organelle. In other words, it is more likely to find a membrane contact site in a ER-derived vesicle with the surface of $1 \mu\text{m}^2$ than on a vesicle with a surface area of $0.1 \mu\text{m}^2$.

We do not fully understand the argument that increasing the number of fragments promotes the changes of interaction. Increasing the number of fragments does not neither increase the total surface area, nor does it increase the number of proteins that form membrane contact site.

Increasing the number of fragments, however, increases the number of vesicles that lack membrane contact site altogether. Because sonication is essential for an optimal enrichment of ER-derived membranes, we consider our line of argumentation as more correct.

In the revised manuscript, we explain this point better.

g. In Fig 1C, the eluate also seems to contain GFP-negative vesicles. What is the conclusion regarding loss of luminal proteins (Kar2 is also lost in Fig 1D)? What is the size of the scale bar? Does it apply to all 6 panels?

We thank the reviewer for spotting this. The scale bar applies to all 6 panels and corresponds to $5 \mu\text{m}$. We have added this information to the Figure legend 1C.

h. Lipid vesicles obtained by extrusion using 0.2um filters are denoted as large unilamellar vesicles (LUV) (p.13).

The reviewer is correct. We realized that we made a mistake in the materials and methods section. We extruded liposomes subsequently with 0.4 µm filters, 0.2 µm filters, and 0.05 µm filters. In the first version of the manuscript the information regarding the 0.05 µm filter was missing. We have corrected this point in the revised manuscript.

i. Include panels depicting average acyl chain length or length distribution in Figs 2, 3, S2, S3, S6.

j. Cki1 phosphorylates choline at the expense of ATP (Fig S3D).

We have added this additional information, which we had omitted in our first submission because we were focusing on the subsequent steps of PI biosynthesis.

k. Lipidome of whole cell lysate in Fig 6 is identical to that in Fig 2. Was 1 culture used for isolating both ER and vacuole? If so, please indicate.

The reviewer is right and we are happy that this point was spotted. We make these changes in the revised manuscript.

Prof. Robert Ernst
Saarland University
Medical Biochemistry and Molecular Biology
Kirrberger Str. 100
Building 61.4
Homburg, Saarland 66421
Germany

24th Nov 2022

Re: EMBOJ-2022-112605R-Q

A new technology for isolating organellar membranes provides fingerprints of lipid bilayer stress

Dear Robert,

Thanks you again for your detailed response letter to the referee comments on your recent submissions, and for having discussed possible ways of addressing/clarifying the most substantive concerns with me last week. In light of these considerations, we would be happy to consider a revised version of the manuscript for eventual EMBO Journal publication, and I am herewith inviting you to start preparing such a revision along the lines discussed. As mentioned, please also note our specific guidelines for Resource Articles, in particular re. structured Methods and Material sections.

More detail information regarding formatting and uploading revised manuscripts can be found below and in our author guidelines - happy to answer any additional questions you may have in this regard.

Thank you for the opportunity to consider your work for publication. I look forward to your revision.

With kind regards,

Hartmut

9) Digital image enhancement is acceptable practice, as long as it accurately represents the original data and conforms to community standards. If a figure has been subjected to significant electronic manipulation, this must be clearly noted in the figure legend and/or the 'Materials and Methods' section. The editors reserve the right to request original versions of figures and the original images that were used to assemble the figure. Finally, we generally encourage uploading of numerical as well as gel/blot image source data; for details see: embopress.org/page/journal/14602075/authorguide#sourcedata

At EMBO Press, we ask authors to provide source data for the main manuscript figures. Our source data coordinator will contact you to discuss which figure panels we would need source data for and will also provide you with helpful tips on how to upload and organize the files.

In the interest of ensuring the conceptual advance provided by the work, we recommend submitting a revision within 3 months (22nd Feb 2023). Please discuss the revision progress ahead of this time with the editor if you require more time to complete the revisions. Use the link below to submit your revision:

Link Not Available

Response to reviewers
Manuscript EMBOJ-2022-112605

We thank all reviewers and the editor for their efforts in identifying strengths of our manuscript and points that can be improved/clarified. I am convinced that by addressing the reviewer's concerns and questions has improved the manuscript significantly.

Referee #1:

In this manuscript submitted to EMBO Journal (EMBOJ-2022-1126050), Reinhard et al. describes a new, forward-thinking technology known as MemPrep, for isolating different organellar membranes in yeast. The authors thoroughly vetted the purification process and took careful consideration in obtaining a pure organellar fraction via use of quantitative proteomics and immunoblotting with specific organellar markers. The authors demonstrated that the isolated organellar membranes can be used in a variety of downstream applications such as lipidomics and proteomics in different stress conditions; all contributing to a comprehensive understanding of how cells adapt their lipid composition at the organellar level. Many studies are emerging which demonstrates how membrane protein function is influenced by the lipid environment and vice versa. Thus, this technology is significant and can be broadly appreciated in many fields including protein quality control, protein translocation, lipid homeostasis, and stress. Overall, this manuscript is well-written and presented in logical fashion and the quality of the data is sufficient to make arguments convincing. This paper presents a new technology, which will be an excellent addition to many fields. To improve the manuscript, I have included suggestions below, which the scientific community could benefit from, if they were using MemPrep.

We thank the reviewer for highlighting the potential of this technique. In fact, we received numerous requests for collaborations from leading labs focusing on various aspects of protein quality control, the ERAD-system, organelle biology, lipid metabolism, and membrane biophysics after sharing the MemPrep manuscript as a preprint on bioRxiv and presenting the isolation approach at various conferences. One of the initiated collaborations has already led to a publication (Reinhard et al., Biophys J, 2023; DOI: 10.1016/j.bpj.2023.01.009). We shared our protocols and materials with the team of Hannes Herrmann (TU Kaiserslautern), who applied a variation of the MemPrep protocol to study the role of the ERMES complex (forming ER-mitochondria. contact sites) in the delivery of mitochondrial preproteins to the import machinery via the surface of the ER (DOI: 10.1101/2023.08.10.552816). These two examples highlight that the MemPrep technology is indeed of interest for the fields of organelle biology and membrane biophysics.

Experimental concerns:

1. Because this is a technology paper, it would be beneficial for the community if you have a section discussing limitations of this approach, if any. For example, what is the criteria for choosing an ideal bait protein? Do they have to be abundant like the reticulon used in this study? Perhaps you have tested several bait proteins for different organelles already? Would be nice to state what they are (if not, no need to add). Moreover, many in the field are interested in separating ER vs nuclear membrane or smooth ER vs. rough ER. Will be useful, to discuss limitations for separating membranes that appear contiguous.

The reviewer raises important points, which were also (in part) mentioned by reviewer 3 (points 1 and 2). For the revised manuscript. we provide an extended description of our rationale for choosing suitable bait proteins. We state: '*An ideal bait is a highly abundant transmembrane protein, feature an accessible C-terminus, and localize exclusively to a single organelle.*'

We also performed extensive new experiments to identify and discuss specific limitations of the MemPrep strategy in a new paragraph. Specifically, we isolated ER membranes via two bait proteins (Rtn1-bait and Elo3-bait) localizing to distinct ER subdomains (**new Figure 2F, and Figure EV2**). It turns out that both the proteome and the lipidome of these isolates are virtually identical. We discuss the implications of this finding (both pro's and con's) in the revised manuscript.

The fact that immuno-isolates via the Rtn1-bait and the Elo3-bait reflect rather the global ER composition than the composition of a specific ER subdomain may not be a weakness. Instead, it opens the opportunity for a range of applications. When combining the MemPrep approach with quantitative proteomics to study for example the substrate spectrum of a particular membrane protein insertase or a specific route of inter-organelle transport (e.g. transport between ER and mitochondria as demonstrated by the Herrmann laboratory in a manuscript employing the MemPrep approach (DOI: 10.1101/2023.08.10.552816)), it is much easier to analyze the proteome of a sample, which reflects several (potentially all) ER subdomains rather than only a single one. Hence, this limitation of the MemPrep procedure can lower the number of costly and work-intensive immuno-isolations and lipidomic/proteomic experiments required to address questions on the remodeling of the ER composition. We extended the discussion of this point in the revised manuscript. See: '*Quantitative proteomics validates performance of MemPrep with distinct bait proteins*' in the revised manuscript.

Notably, when applying a variation of the MemPrep approach for the isolation of ER membranes from mammalian cells (not part of the manuscript), we can isolate membrane subdomains as validated by proteomics (*manuscript in preparation*). Despite different proteomes, we find identical lipid compositions in immuno-isolates via REEP5-bait and a SEC63-bait. We are currently performing more immunoisolations using a CLIMP63-bait. We have invested significant efforts in figuring out the reason, why we can selectively isolate ER membrane subdomains from mammalian cells, but not from yeast. We are convinced that the earliest step of the procedure, the cell lysis, is one of the main reasons for the differences between yeast and mammalian cells. Another contributing factor is likely the different organelle architecture, which is different between HEK293 cells and yeast. Hence, we cannot exclude that mixing of ER subdomains occurs during the early steps of isolation from yeast, but we have no indication that ER membranes mix with other organelles: Our proteomics and lipidomics data, as well as extended control experiments argue against this.

We added a paragraph to the discussion on the limitations of the MemPrep technology. We state: '*There are, however, also limitations to the MemPrep approach, which has been optimized for highest purity. The yields of the preparation are low, which can make the isolation of membranes from low abundant organelles less feasible. While MemPrep provides a comprehensive snapshot of a specific organelle membrane, the associated preparative efforts are significant, and the benefit of extensive time-course experiments should be carefully evaluated. Because lateral specializations of the ER membrane and membrane contact sites are disrupted during the preparation, the MemPrep approach is not suitable to isolate membrane contact sites or other lateral membrane specializations such as the outer or inner nuclear membrane. For these purposes, proximity labeling approaches and a selective solubilization of membrane proteins are promising developments (Kwak et al, 2020; van 't Klooster et al, 2020).*'

In summary, we substantially extended our discussion and included new data, which support the original conclusions but also highlight limitations and possible applications of the MemPrep strategy.

2. In regards to smooth ER vs. Golgi ER statement above, by pulling down Rtn1-baited ER (localized mainly in smooth ER) if you look at other ER proteins enriched in Fig. 1E, do you

also see enrichment of smooth vs. rough ER membrane proteins? If so, would be invaluable to add this analysis to the manuscript. Of course, this isn't needed if you don't see a difference and can be included in your discussion within the "technical limitation" section.

Very good point. See our answer to point 1. In brief, we performed an experiment as suggested by the reviewer and isolated ER membranes via two bait proteins localizing to distinct ER subdomains (**new Figure 2F, and Figure EV2**). As anticipated by the reviewer, we do not see a specific enrichment of ER proteins in any of the isolations based either on the Rtn1- or the Elo3-bait. We discuss this finding as 'technical limitation' in the revised manuscript. Notably, upon performing a variation of the MemPrep approach on mammalian cells, we observe clearly distinct proteomes for the pulldowns of sheet vs. tubular ER (not part of this manuscript; *manuscript in preparation*).

3. For Figure 1D, was a nuclear membrane protein marker used as another control? Alternatively, in Figure 1E, it appears nuclear membrane proteins are slightly depleted in the purified ER fraction. I suggest stating the fold difference in the results section to make it clear to the readers.

We have not used a nuclear membrane marker for the immunoblots in Figure 1D but we use a variety of nuclear proteins in mass spec experiments. We have followed the reviewer's advice for the revised manuscript and provide numbers to clarify the fold difference (**modified Figure 1E**). We extended our analysis to distinguish between nuclear membrane proteins and protein from the nucleoplasm (**modified Figure 1E**). Also, we have updated the GO-term annotation leading to slightly different numbers compared to the previous version of the manuscript. We clearly state in the 'limitations' section that '*the MemPrep approach is not suitable to isolate membrane contact sites or other lateral membrane specializations such as the outer or inner nuclear membrane.*'

Following the suggestion of the review, we provide the fold difference in the revised Figure 1E and discuss the implications. Most proteins of the endomembrane system are enriched in our preparation including those with the annotation 'nuclear membrane'. Nevertheless, given that the ER membranes is probably one of the most abundant membrane to begin with, and given that ER markers are substantially enriched in the final isolate (>20-fold), we conclude that the resulting sample is dominated by ER membranes.

We now state: '*A moderate enrichment of markers from other organelles of the endomembrane system (Golgi apparatus, vacuole, etc.) was found as expected since they pass through the ER on their route to their subcellular destination and because the efficient removal of soluble proteins alone causes an enrichment of organelle membrane markers. In line with the procedure that is intended to enrich for the membrane fraction, ER membrane proteins were substantially more enriched than ER luminal, soluble proteins or proteins from other organelles (Appendix Figure S1D).*'

We proceed with: '*Only a few proteins annotated to other organelles are enriched >20-fold over the lysate (Appendix Figure S1D) and for most of these there is evidence that they in fact localize to the ER or the nuclear envelope, which is continuous with the ER membrane. Hence, Osm1, Yur1, Ist2, Ygr026w, Pex30, Pex29, Slc1, Uip6, Brr6, and She10 were falsely annotated as non-ER proteins (Appendix Figure S1D). A dual localization including the ER and another organelle has been reported for Osm1, Yur1, Pex31, Slc1, Cst26, Svp26, Ept1, and Cbr1. Likewise, there is evidence for an ER localization for the non-annotated proteins Ybr096w, Gta1, Msc1, and Hlj1. This suggests that MemPrep and quantitative proteomics can even predict ER membrane localization.*'

For a further discussion, see our answer to reviewer 3, point 4.

4. It is very interesting that DTT and TM treatment leads to similar changes in lipid composition and yet has different impact on the ER proteome. Was DTT and tunicamycin treated at different times other than 4 hours? Would be interesting to see if 2 hour or longer time points led to distinct lipid composition in DTT-treated cells vs. tunicamycin treated cells. If this was done, can just state as data not shown.

This is indeed an interesting observation, which we discuss prominently in the revised manuscript. We even extended this type of analysis by also providing the proteome of the ER from inositol-depleted cells (**new Figure 3A**), which differs dramatically from the one observed upon DTT and TM treatments.

Yet, we have not yet performed a more extensive, time-dependent experiment using DTT and TM. We believe that the complex changes of the proteome in the context of these prolonged forms of stress will make a straightforward interpretation of the underlying mechanisms extremely challenging. We have used the single 4 hour time point, because previous data by others and us suggest that the UPR is driven predominantly by ER membrane signals (PMID: 21775630; PMID: 34196665).

Nevertheless, we have substantially rewritten the section '*Lipid bilayer stress caused by proteotoxic agents Dithiothreitol (DTT) and Tunicamycin (TM)*' and added new controls and extensive data. These involve a **new Figure EV4** (qPCR data) and a **new Figure 4A,B** (whole cell lipidomics with WT and *ire1Δ* cells lacking a functional UPR). These data lead to

the important conclusion that the complex remodeling of the cellular lipidome in stressed cells is independent of UPR signaling. In other words: The UPR is activated by a membrane-based mechanism at this timepoint and previous UPR signaling was insufficient in adapting lipid metabolism to handle the metabolic challenge associated with the prolonged stress from by DTT and TM.

We also have rewritten the proteomics section '*DTT and TM have similar yet distinct impact on the ER proteome*' and discuss the proteomes and lipidomes induced by inositol-depletion (**new Figure 3A**) with those induced by TM, and DTT (**Figure 5; Figure EV5D,E; Figure S5**).

We believe that rewriting these sections and adding new data was very important and improved the manuscript substantially. We thank all reviewers for their important input.

Text/Figure concerns:

1. Line 51, should state Ire1 is from *S. cerevisiae* here as opposed to sentence below(line 54).

Thank you. We corrected this for the revised manuscript.

2. Line 371, add comma after "In the past"

Corrected.

3. For Figure 3D, I don't see difference in number of double bonds between control and inositol depletion conditions. Please clarify as this difference was stated in line 556.

We changed the text to: *'We further dissected the compositional changes of the ER membrane lipidome upon inositol-depletion at the level of the lipid acyl chains and observed a minor, non-significant trend toward more saturated glycerophospholipids (Figure 3E). While these changes are likely to fine-tune the physicochemical properties of the ER membrane, it is unlikely that they alone are sufficient to trigger the UPR by activating Ire1 (Halbleib et al, 2017).'*

For the revised manuscript, we show a trend towards more saturated lipid acyl chains, which is observed throughout the major glycerophospholipid classes (**new Appendix Figure S5A**). We state: *'Upon inositol depletion, a nuanced shift toward shorter and more saturated acyl chains is observed across major glycerophospholipid classes (**Appendix Figure S5A**). This contrasts with the impact of an increased PC-to-PE-ratio enforced by choline supplementation, which barely leaves any marks in the lipid acyl chain composition (**Appendix Figure S5B**). Moreover, prolonged proteotoxic stress induced by DTT or TM elicits distinctive impacts on the composition of lipid acyl chains (**Appendix Figure S5C, D**). In this context, the acyl chains demonstrate a tendency throughout most lipid classes to become slightly longer and more saturated.'*

Referee #2:

In this work, the authors describe a method to enrich ER-derived membranes and vacuolar membranes from microsomes prepared from cell extracts and differential centrifugation. The method depends on expression of a set of tagged ER membrane-associated proteins for which membrane fragments bearing the tagged protein can be enriched based on trapping the tagged protein. The isolated membranes were analyzed for protein and lipid contents by state-of-the-art proteomics and lipidomics methodologies. The data supported the conclusion that the ER membranes were enriched to a degree never achieved in previous reports

making it possible to assign lipid composition of organelles in a more definitive manner. Moreover, the new method permitted analyses to examine the correlation between lipid composition and membrane protein insertion, folding, and sorting in response to stress and phospholipid lipid precursor availability. The work is expertly performed and appropriately quantified. The "MemPrep" method described here, along with the data sets generated, will be of great value to the yeast community with interests in organelle isolation and the metabolic processes occurring therein. Minor criticisms of the work and manuscript are noted below.

Comments:

1. The inositol-mediated effects on lipid synthesis and the unfolded protein response pathways may be compromised by the fact that the inositol concentration in the synthetic media is ~ 11 micromolar. Studies by the Henry laboratory in the 1980s and 1990s have shown that the regulatory effects of inositol occurs at concentrations > 50 micromolar. Perhaps the authors supplemented their growth media with inositol, but I didn't see this.

This is an important point. We are aware of the beautiful work by the Henry laboratory. Indeed, UAS-INO target genes are fully repressed only at inositol concentrations >50 μM . Hence, reaching the full dynamic range of the UAS-INO regulation would require higher inositol concentrations in our medium. Yet, inositol at 10 μM already substantially lowers the *INO1* mRNA level (Hirsch et al., 1986; PMID: 3025587) and is sufficient to rescue inositol-auxotrophy mutants at 30°C.

Our rationale for choosing a medium with 11 μM of inositol was not to demonstrate the full repression of UAS-INO regulation, but to warrant a rapid, full activation of the UPR upon removing inositol from the medium and to provide a better comparison to previous studies on the UPR.

We made sure to clearly state the inositol concentration of our medium in the revised manuscript. We state: '*Because standard SCD medium contains only 11 μM inositol and because the BY4741 is particularly dependent on inositol for normal growth (Hanscho et al, 2012) it is possible that prolonged cultivation of this strain leads to a 'natural' inositol depletion.*

We again refer to the inositol concentration in the medium in the discussion when referring to newly added whole cell lipidomics data (**new Figure 4A**); **new Figure EV4**) We state: '*A general role of negatively charged lipids as attenuators of the UPR would have important physiological implications, because the level of PI and other anionic lipids change substantially in different growth stages (Casanovas et al, 2015). In fact, the availability of inositol is limiting for optimal growth of the commonly used strain BY4741 (Hanscho et al, 2012) and its prolonged cultivation in synthetic medium containing only 11 μM causes UPR activation even in the absence of exogenous stressors (Figure EV4) when the cellular level of anionic PI lipids is low (Figure 4A).*

In future studies, I suggest supplementing the inositol-free media with 50-75 micromolar inositol for comparing effects of this precursor.

We thank the reviewer for this suggestion.

Additionally, the effects of choline on lipid synthesis are dependent on the higher concentration of inositol. This information is found in the primary literature cited in the review (Henry SA, Kohlwein SD & Carman GM (2012) Metabolism and regulation of glycerolipids in the yeast *Saccharomyces cerevisiae*. Genetics 190: 317-49) that is cited in this manuscript. If I'm correct about this issue, I don't request repeating the experiments, but instead, discuss

this caveat. Parenthetically, I'm disappointed that many do not read nor cite the primary literature.

Following the advice of the reviewer, we refer to two relevant studies related to the role of choline and inositol in the revised manuscript (Hirsch and Henry, 1986; Gaspar *et al.*, 2006).

We state that '*Expectedly, the ER of choline-challenged cells features substantially higher levels of PC lipids at the expense of PE and causes an increase of the PC-to-PE ratio from ~1.1 to ~2.4 (Appendix Figure S3A). [...] Lipid metabolism and the PC-to-PE ratio may have been more affected if different concentrations of choline and inositol had been used (Hirsch & Henry, 1986; Gaspar et al, 2006).*'

Gaspar *et al.* (PMID: 16777854) provided evidence for a distinct impact of either inositol and choline alone or in combination on lipid metabolism. Notably, they investigated the impact of these supplementations on the lipid class composition at steady state. Figure 2 in this reference (Gaspar *et al.*; PMID: 16777854) indicates that the PC-to-PE ratio is -within error-barely affected when choline is supplemented either in the presence or absence of additional inositol.

In our experiments, we have supplemented choline to the medium to increase the level of PC and to increase the PC-to-PE ratio. While our supplementation may not have been the optimal, it clearly increased the PC level and increased the PC-to-PE level.

2. The membrane enrichment protocol is dependent on strains expressing organelle-specific tagged proteins. While the authors are willing to make the strains freely available, depositing the strains in widely available repository is advisable for the long term.

This is a very good point. We will carefully consider this option.

3. Investigators will want to isolate highly enriched membrane fractions from cells with specific mutations. Have the authors introduced any mutations in the strains expressing organelle-specific tagged proteins. This might be mentioned in the text.

The reviewer indicates an important application of MemPrep technology. We have already performed such experiments in collaboration with other laboratories and anticipate additional manuscripts in the future. In the discussion of the revised manuscript, we now state '*While changes in lipid abundance can be readily identified by whole cell lipidomics, it is not possible to study their redistribution in cells from one organelle to another unless individual organelle membranes can be isolated and analyzed. This is now becoming feasible. Naturally, it also possible in this context to combine gene deletions with organelle-specific bait strains for studying the role of a specific gene on the lipid composition of an individual organelle.*'

4. On lines 44 and 45, the authors cite preprints which have not been peer reviewed. I hope that this journal does not condone this practice.

Formal citations of preprints are encouraged by The EMBO Journal.

5. On line 656, the authors use the much overused phrase "It is tempting to speculate", and then they go on to make the speculation. Just state "we speculate" that

We removed the phrase from the manuscript as suggested by the reviewer.

Referee #3:

This manuscript reports an improved protocol "MemPrep" for purifying yeast ER that is based on the immune isolation reported by Klemm et al. (2009). The extensively optimized procedure yields an unprecedented enrichment of ER membranes as judged by immunoblot and untargeted protein mass spectrometry. The full lipidome of the purified ER fraction is presented. To identify ER compositions that trigger UPR, the effect of several conditions including prolonged proteotoxic stress, on the ER proteome and lipidome is analyzed. Based on the data, mixtures of synthetic lipids are proposed that mimic ER lipid composition under these conditions, to facilitate future mechanistic in vitro studies addressing UPR activation. To demonstrate broader application, MemPrep is used for establishing the lipidome of the yeast vacuole. The discussion is predominately speculative in line with the descriptive nature of the study. The enormous effort and rigor invested in this work is impressive, as is the quality of the proteomics and lipidomics data presented. The data provide a useful resource for future research. However, this reviewer cannot help thinking that the authors got carried away by the wealth and quality of the data, because of the often biased interpretation of results.

We would like to thank the reviewer for the positive, general assessment, but even more so for the critical comments, which helped us to substantially improve the manuscript.

Comments

1. The ER is generally considered a heterogeneous organelle comprising (specialized) subdomains such as SER, RER, tubular, sheet, perinuclear, cortical, PAM, MAM. Surprisingly, data are not discussed or interpreted in this context. Is the variation in ER marker enrichment factors (Fig S1C, S1D, Table S4) related to ER heterogeneity?

Following the advice of the review, we have extended our discussion on the tubular and sheet ER. We also included a discussion on the feasibility of the MemPrep approach for isolating ER subdomains.

We have added an entire section and new data regarding this important point in the revised manuscript. (***Quantitative proteomics validates performance of MemPrep with distinct bait proteins***). Related to this, see our answer to point 1 by reviewer 1.

When we isolate the ER using two different bait proteins (Rtn1- and Elo3-) localizing to two different regions of the ER, we obtain almost identical lipid and protein compositions (**new Figure 2F; new Figure EV2**). The implications of these findings (both the pro's and the con's) are discussed in the revised manuscript.

Based on these data and elaborated in our answer to reviewer 1, point 1 and 2, we discuss limitations of the MemPrep in a new paragraph added to the discussion of the revised manuscript. We state: *'Because lateral specializations of the ER membrane and membrane contact sites are disrupted during the preparation, the MemPrep approach is not suitable to isolate membrane contact sites or other lateral membrane specializations such as the inner nuclear membrane. For these purposes, proximity labeling approaches and a selective solubilization of membrane proteins are promising developments (Kwak et al, 2020; van 't Klooster et al, 2020).'*

2. Is the purified ER fraction representative for "the ER"?

This is an important question, which cannot be answered with 100% certainty. We refer to this point in the revised manuscript after discussing the lipidome and proteome of two immunoisolations using the Rtn1- and Elo3-bait (**new Figure 2F; new figure EV2**). We state: *'This suggests that MemPrep yields preparations, which rather represent the 'entire' ER membrane than a specific ER subdomain. Yet, we cannot exclude that the portion of the ER that is lost/discarded during the preparation may have a different composition. We speculate*

that the harsh mechanical disruption of the cell, which is required to break the cell wall, causes a fragmentation of the ER network that disrupts lateral specializations.'

Specifically, we provide new data from immunoisolations with the Rtn1-bait and the Elo3-bait yielding identical lipid and protein compositions (**new Figure 2F, and Figure EV2**). To us, this is indicative that our immunoisolations report rather on "the ER membrane" than on a specific subdomain. Nevertheless, given that a major fraction of the ER is lost during the preparation in P12, we cannot formally exclude that those portions of the ER have an identical composition.

We also added the following sentence to the results section: *'We cannot formally rule out the possibility that the discarded ER membrane vesicles in the P12 fraction have a different composition than the rest of the ER, which we subsequently isolate.'*

Notably, we can isolate ER subdomains from cultured mammalian cells using a variation of the MemPrep approach (*manuscript in preparation*). As discussed with the editor, we refer to our experience with mammalian cells in the revised manuscript. We state: *'While we know from our work with mammalian cells that ER subdomains can be isolated via MemPrep, we are convinced that preparations representative of the 'entire' organelle membranes have practical advantages for studying inter-organelle transport processes by lowering the minimally required sample number for proteomics and lipidomics experiments as indicated by first applications using the MemPrep technology and variations thereof (Reinhard et al, 2023; preprint: Koch et al, 2023).'*

Is the ER fraction that ends up in P12 (p.13) expected to be similar to the purified ER fraction?

This question is related to the previous one. It should be possible to use the P12 fraction for isolating ER membranes or even to omit the entire subcellular centrifugation procedure. However, based on our extensive experience from years of optimization, we expect that the resulting isolates would not be pure enough and much more contaminated with other organelles. We do not think that it is possible to distinguish between a specific, local ER composition and contaminating organelles from such preparations.

To address the concern of the reviewer and to highlight the potential caveat that the discarded ER membranes in fraction P12 may have a different composition than the rest of the ER, we added disclaimer to the results section. We state: *'We cannot formally rule out the possibility that these discarded ER membranes in the P12 fraction have a different composition than the rest of the ER, which we subsequently isolate.'*

Although the prime goal was purity, the protein yield of the procedure remains an important parameter in organelle isolation. Is it possible to estimate the yield of pure ER from the cell lysate based on marker analysis?

We thank the reviewer for raising this point and have included relevant information to the revised manuscript. We state: *'Overall, the MemPrep procedure provides high purity of organelle-derived membranes at the expense of low yields (83 mg protein in the cell lysate yields ~30 µg of protein in the isolate via the Rtn1-bait). Assuming the ER accounts for 20% of the total cell protein (Zinser & Daum, 1995), we estimate that >98% of ER protein is lost during the isolation.'* We also state: *'Quantitative proteomics reveals that fusing the bait tag to either Rtn1 or Elo3 has no impact on the overall cellular proteome (Figure EV1B) and that ER proteins can be enriched by MemPrep using either of the two baits (Figure EV1C, D) with an estimated yield of 1.9% and 0.8% of the input material for the Rtn1- and Elo3-bait, respectively.'*

How did we get there? We isolate the ER from 2L of cells with an $OD_{600} = 1$ (2000 ODunits). Typically, 2000 ODunits of cells yield a total protein concentration of the lysate of $\sim 83000 \mu\text{g}$ protein.

For an isolation from 2000 OD units via the Rtn1-bait, we obtained on average $31 \mu\text{g}$ isolated ER membrane protein.

For an isolation from 2000 OD units via the Elo3-bait, we obtained on average $14 \mu\text{g}$ isolated ER membrane protein.

Assuming that the ER accounts for $\sim 20\%$ of the total cell protein (PMID: 7645343), this would correspond to a yield of 1.9% for the ER isolated via the Rtn1-bait, and 0.84% for the Elo3-bait.

3. Does the sonication step in the procedure affect the sidedness of ER vesicles?

The review is correct. Sonication can affect the sidedness of ER vesicles. We now state the in the Materials and Methods section: '*While sonication may affect the sidedness of ER-derived vesicles, this step was crucial for maximizing the purity of the preparation.*'

Notably, if the sidedness of ER-microsomes would be substantially affected by sonication, the bait tag for the subsequent affinity-purification would not be accessible to the anti-FLAG antibodies. While omitting the sonication step may thus help increasing the yield, it certainly would lower the purity and result in non-interpretable results. Because the MemPrep procedure (including the sonication step) also works well with mammalian cells allowing even for the isolation of ER subdomains (*manuscript in preparation*), we consider the sonication step as a crucial element of the MemPrep procedure.

A FRET assay is used to exclude the occurrence of extensive organelle fusion or lipid exchange during sonication. The results (Fig S1B) do not convincingly support the conclusion. The assay conditions chosen may underestimate lipid exchange or fusion due to insufficient sensitivity. With PEG and Ca^{2+} present relative FRET efficiency is only decreased to 80%, indicative for a 1:1 dilution of the probes (Struck et al., 1981); with SDS added (concentration?) relative FRET is still 50%. Results suggest that the fold-excess of P100 over liposomal lipids is less than 8.2, which may originate from the peculiar scattering method used for quantifying liposomes and P100 that requires similar sizes of the vesicular structures to be compared. Quantitation of lipid content after extraction is more straightforward and reliable.

The reviewer highlighted important limitations of the FRET assay as performed and described in the original manuscript. For the revised manuscript **we have quantified lipids, optimized the assay, improved its dynamic range** and data processing (by correcting for bleed-through from filter settings and fluorophore cross-talk), **and performed more controls (new Appendix Figure S1B).**

With respect to the quantification of lipid, we state in the revised manuscript: 'LUVs were mixed with a 15.4 ± 1.3 -fold excess of unlabeled P100 microsomes as quantified by a total phosphate determination after lipid extraction and acid hydrolysis based on classical protocols (Chen et al, 1956; Bligh & Dyer, 1959; Rouser et al, 1970). In brief, lipids from 100 μ l of LUV or microsome solution were mixed with 100 μ l MP buffer without protease inhibitors and benzonase (25 mM HEPES pH 7.0, 600 mM mannitol, 1 mM EDTA) and extracted in two steps by addition of first 750 μ l chloroform:methanol (1:2) and then 750 μ l chloroform and 250 μ l ABC buffer (155 mM ammonium bicarbonate) at RT. The organic phase was recovered, and the solvent was evaporated using a centrifugal evaporator. Dried lipids were resuspended in 100 μ l chloroform:methanol (1:2). 25 μ l of this lipid extract were transferred to a pyrex glass tube, and the solvent was evaporated. The resulting lipid cake was treated with 300 μ l 70 % perchloric acid at 180 °C. Subsequently, 1 ml Milli-Q water, 0.4 ml of 5% (w/v) ascorbic acid, and 0.4 ml of 1.25% (w/v) ammonium molybdate were added and the resulting sample was boiled for 15 min. Absorbance at 797 nm was measured at RT after the sample cooled down.'

We also described the FRET assay itself (see Materials and Methods section) and added a **new Appendix Figure S1B**.

*Fluorescence spectra and calculation of relative FRET efficiencies in mixtures of labeled liposomes and a ~15.4-fold excess of unlabeled P100 microsomes after sonication as performed during MemPrep procedure (10 s sonication), after extensive sonication (100 s sonication), or upon incubation with 18 mM methyl beta cyclodextrin (M β CD), 40% (w/v) polyethylene glycol 8000 (PEG 8000), or 1% sodium dodecyl sulfate (SDS). A low relative FRET efficiency is the result of decreased average proximity of the two FRET-pair fluorophores and indicative for either fusion of labeled liposomes with unlabeled P100 microsomes or lipid exchange between vesicles. Error bars indicate standard deviations. Statistical significance was tested using an unpaired parametric t test with Welch's correction. ns: not significant; ***p < 0.001.*

We dedicated this important control experiment an extended paragraph in the results section: 'Sonication transiently disrupts lipid bilayers and can theoretically induce lipid mixing or a transient fusion of adjacent lipid bilayers. Because this would obscure our measurement of the ER membrane composition, we performed control experiments to rule this out. We utilized small unilamellar vesicles of ~100 nm containing POPC, NBD-PE, and Rho-PE at a ratio of 98:1:1. The two fluorescent lipid analogs form a Förster resonance energy transfer (FRET) pair (Appendix Figure S1B). We sonicated these synthetic liposomes in the presence of a ~15.4-fold excess of microsomal membranes (P100). Because fusion between the synthetic liposomes and microsomal membranes would 'dilute' the fluorescent lipid analogs, a decrease of the relative FRET efficiency would be expected upon membrane mixing or upon the exchange of individual fluorescent lipid molecules. However, the 10 cycles of sonication as used during MemPrep procedure for dissociating vesicle aggregates do not lead to a significant change of the FRET efficiency. Lower FRET efficiencies indicative for

lipid exchange or membrane fusion, was only observed after 100 cycles of sonication, which also leads to sample warming, and upon incubation for 30 min at RT with either 18 mM Methyl- β -cyclodextrin, which facilitates lipid exchange, or with 40% w/v PEG 8000, which supports membrane fusion (Appendix Figure S1B) (Lentz, 1994; Cheng et al, 2009). Expectedly, we observed a dramatic drop of the relative FRET efficiency upon the addition of SDS, which dissolves both the liposomal and microsomal membranes (Appendix Figure S1B). These data suggest that the sonication as used in the MemPrep procedure does not cause a significant degree of membrane mixing from fusion and/or lipid exchange.'

Our original interpretation that sonication does not neither induce a substantial membrane mixing nor any other exchange of lipids between membranes from different organelles remains valid.

4. The immuno-isolated highly purified ER prep is concluded to be largely devoid of cytosolic, inner nuclear membrane and mitochondrial markers (p.14). The significant enrichment of numerous markers of PM, vacuole, and Golgi in the ER fraction vs. P100 (Fig. 1E) and vs. cell lysate (Fig S1C, Table S4) goes unmentioned. Yet the contamination with these organelles most likely accounts for the unexpected finding of IPC, MIPC and M(IP)2C in the ER fraction.

The reviewer highlights a critical point. Are the preparations of ER membranes sufficiently pure? In the revised manuscript, we discuss the enrichment of non-ER markers of the endomembrane system more extensively. We also **updated Figure 1E** (as suggested by reviewer 1) and provide a **revised Appendix Figure S1D**. Yet, we respectfully disagree with the statement that the reported IPC, MIPC and M(IP)2C levels are most likely due to a contamination in the ER fraction. This cannot be the case (see below).

It is true that organelle markers of the endomembrane system are enriched in our preparation, but less so than ER membrane proteins. In the revised manuscript, we now state: '*A moderate enrichment of markers from other organelles of the endomembrane system (Golgi apparatus, vacuole, etc.) was found as expected since they pass through the ER on their route to their subcellular destination and because the efficient removal of soluble proteins alone causes an enrichment of organelle membrane markers. In line with the procedure that is intended to enrich for the membrane fraction, ER membrane proteins were substantially more enriched than ER luminal, soluble proteins or proteins from other organelles (Appendix Figure S1D).'*

We extended our analysis of the untargeted proteomics data using the Rtn1-bait and added a **new Appendix Figure S1D**.

We now state in the results section: *'Only a few proteins annotated to other organelles are enriched >20-fold over the lysate (Appendix Figure S1D) and for most of these there is evidence that they in fact localize to the ER or the nuclear envelope, which is continuous with the ER membrane. Hence, Osm1, Yur1, Ist2, Ygr026w, Pex30, Pex29, Slc1, Uip6, Brr6, and She10 were falsely annotated as non-ER proteins (Appendix Figure S1D). A dual localization including the ER and another organelle has been reported for Osm1, Yur1, Pex31, Slc1, Cst26, Svp26, Ept1, and Cbr1. Likewise, there is evidence for an ER localization for the non-annotated proteins Ybr096w, Gta1, Msc1, and Hlj1. This suggests that MemPrep and quantitative proteomics can even predict ER membrane localization.'*

We have modified our discussion on the observation of complex sphingolipids in the ER. We now state: *'Significant levels of complex sphingolipids in the ER, on the other hand, have previously been observed and are not surprising, even though the de novo biosynthesis of these lipids occurs in the Golgi complex (Hechtberger et al, 1994; Schneiter et al, 1999) (Figure 2B). In fact, complex sphingolipids can be transported to the ER at substantial rates, in part for their degradation to ceramide by the phospholipase Isc1 in the context of a sphingolipid salvage pathway (Matmati & Hannun, 2008).'*

We do not think that contaminations with other organelles 'most likely account to the unexpected finding of IPC, MIPC, and M(IP)2C in the ER fraction'. The reviewer argues that the observed enrichments of the plasma membrane and other organellar markers in our preparations indicate a significant contamination with these organelles, which in turn account for the substantial quantities of complex sphingolipids, which we report as ER lipids.

Firstly, the proteomics experiment is normalized to the protein concentration in each sample and it is not possible to use this type of normalization when trying to assess the degree of lipid contaminations. Enrichment of proteins are normalized considering both membrane proteins and soluble proteins. Membrane proteins are enriched, when soluble proteins are removed and when the sample is normalized to the protein content. Lipids are not enriched, when soluble proteins are removed, and the samples is normalized to the lipid content. Hence, washing of a membrane pellet leads to an enrichment of membrane proteins, but not to an enrichment of lipids. In other words, a five-fold enrichment of PM proteins in the final isolate does not mean that PM lipids are also five-fold enriched. Hence, we now state: *'A moderate enrichment of markers from other organelles of the endomembrane system (Golgi apparatus, vacuole, etc.) was found as expected since they pass through the ER on their route to their subcellular destination and because the efficient removal of soluble proteins alone causes an enrichment of organelle membrane markers.'*

Secondly, it is important to account for the fact that the ER is already one of the most abundant in glucose-grown cells (according to Zinser and Daum (PMID: 7645343). ER markers cannot possibly be enriched as much as other organelle markers, e.g. from the plasma membrane or the Golgi. Plasma membrane marker proteins or Golgi marker proteins need to be >100-fold enriched to be considered as reasonably pure. The >24-fold enrichment of ER markers, which we report, is unprecedented. The enrichment of other markers of the endomembrane system between 3-fold (for Golgi markers) and 6-fold (for peroxisome markers) does not mean that the resulting lipidome is substantially affected by such minor contamination.

This point is further explained by a direct quote from the seminal review 'Isolation and Biochemical Characterization of Organelles from the Yeast, *Saccharomyces cerevisiae*' by Zinser and Daum (PMID: 7645343).

"The following example will illustrate how cross-contamination of preparations is influenced by the abundance of organelles in the cell (Figure 1). In mitochondria isolated from

aerobically grown yeast, a specific mitochondrial marker is six-fold enriched over the homogenate. In addition, markers of the plasma membrane and microsomes in this fraction are five- and three-fold enriched, respectively. Although all three markers exhibit enrichment factors of nearly the same magnitude, these results should be interpreted with caution when assessing the quality of the preparation. The first conclusion that can be made is that mitochondria are sufficiently enriched, because optimal enrichment factors of this organelle are in the range of 4 to 7. Secondly one can conclude that this fraction is only slightly contaminated with plasma membrane. This conclusion is based on the fact that the plasma membrane comprises only about 1-2% of the total protein of a yeast cell. Highly enriched plasma membrane exhibits an enrichment factor of 100. The third conclusion is that the preparation of mitochondria is, in contrast to the low contamination with plasma membrane, heavily contaminated with microsomes because optimal enrichment factors of 6 are typically observed for purified microsomal fractions.”

Taking this into account, it should be clear that **1) the ~24-fold enrichment of ER markers over the cell lysate suggests an unprecedented purity of ER preparations**, and that **2) the sample is not substantially contaminated by other organelles** and that **3) the minor contamination of the final isolate with other organelle markers cannot account for the level of complex sphingolipids observed in ER membrane preparations.**

In the revised manuscript, we dedicated an entire paragraph to the discussion of our lipid data (regarding sterols and complex sphingolipids) and highlight more clearly that our new data are not in conflict with previous observations while potentially transforming our interpretation of the role of lipids at different sites of the secretory pathway.

We state: ‘Our lipidomic data are fully consistent with a gradual increase of lipid saturation along the secretory pathway (Van Meer et al, 2008; Bigay & Antonny, 2012), thereby complementing previous work on the composition of the trans-Golgi network / endosomal (TGN/E) system, secretory vesicles, and the plasma membrane in yeast (Klemm et al, 2009; Surma et al, 2011). Likewise, the ergosterol level of 9.7 mol% in the yeast ER membrane is consistent with previous estimations (Zinser & Daum, 1995; Schneiter et al, 1999; Van Meer et al, 2008) and parallels findings in mammalian cells, for which a resting cholesterol level in the ER between 7 and 8 mol% has been reported along with a switch-like regulation of sterol response element binding protein (SREBP) processing, when the level of cholesterol drops below 5 mol% in the ER (Radhakrishnan et al, 2008; Sokoya et al, 2022). From a cell biological viewpoint, our data suggest that the sterol gradient along the secretory pathway is rather flat from the ER (9.7 mol%) to the TGN/E system (9.8 mol%) (Klemm et al, 2009). If this is indeed the case, our findings support the view that active transport of sterols by lipid transfer proteins (Mesmin et al, 2013) aids lipid and protein sorting at the level of the TGN (Klemm et al, 2009). A flat sterol gradient in the early secretory pathway and a step-wise increase at the level of the TGN has important implications for the sorting of transmembrane proteins (Sharpe et al, 2010; Herzig et al, 2012; Quiroga et al, 2013; Lorent et al, 2017) and is consistent with recent models that favor sterol-enriched vesicular carriers (Borgese, 2016) or sterol-based, selective diffusion barriers for membrane proteins in the early secretory pathway (Weigel et al, 2021). This would be reminiscent of ceramide-based diffusion barriers for membrane proteins in the ER of dividing cells, which limits the access for ‘old’, potentially damaged membrane proteins to the daughter cell (Clay et al, 2014; Megyeri et al, 2019). Significant levels of complex sphingolipids in the ER, on the other hand, have previously been observed and are not surprising, even though the de novo biosynthesis of these lipids

occurs in the Golgi apparatus (Hechtberger et al, 1994; Schneiter et al, 1999) (Figure 2B). In fact, complex sphingolipids can be transported to the ER at substantial rates, in part for their degradation to ceramide by the phospholipase Lsc1 in the context of a sphingolipid salvage pathway (Matmati & Hannun, 2008).'

Accordingly, Fig 3B-control showing almost 2-fold less IPC and M(IP)2C than Fig 2B reflects the batch-dependence of the contamination level.

Complex sphingolipids show dramatic variations throughout a growth curve of batch-cultivated yeast (see e.g. Casanovas et al.; PMID: 25794437). Because the control in Fig. 3B is based on a slightly different protocol of cultivation (for providing an optimal UPR induction from inositol-depletion), the data in Fig 3B and Fig 2B cannot be expected to show the same lipid composition. We have no evidence that contaminating membranes make a substantial significant contribution to our isolated ER membranes.

For clarity, we added the following sentence to the legend of Figure 3: '*Note that the culturing conditions for the unstressed control was different from that used to determine the steady-state ER lipid composition in Figure 2.*'

The suggestion of substantial retrograde flux of complex sphingolipids from the Golgi complex to the ER (p.4, p.14-15, p.22) is far-fetched and lacks experimental support.

We respectfully disagree. According to the yeast genome database, the estimated copy number of the HDEL receptor (Erd2) is $\sim 11,600 \pm 3,100$. The estimated total number of 12 ER-luminal, soluble HDEL proteins (Pdi1, Ero1, Kar2, Lhs1, Yos9, Mpd2, Cpr5, Eug1, Mpd1, Scj1, Jem1, Sil1) sums up to 108,400. Hence, unless these HDEL-containing proteins are not sequestered and kept in the ER by other means, they must constantly be retrieved to avoid leakage and transport along the secretory pathway. Because they outnumber the HDEL receptor by an order of magnitude, and because there is substantial forward traffic (*S. cerevisiae* must double all membranes of the secretory pathway in 90 min) there must be substantial retrograde traffic via the COP-I machinery. We decided against the inclusion of this back-of-an-envelope calculation to the manuscript, because we do not consider it of central relevance for the manuscript.

We believe that the reliability of our ER lipid data is suitably addressed by our previous answers to reviewer 3. Furthermore, we have rewritten several sections of the manuscript related to this point. For example, we state in the discussion: '*Significant levels of complex sphingolipids in the ER, on the other hand, have previously been observed and are not surprising, even though the de novo biosynthesis of these lipids occurs in the Golgi complex (Hechtberger et al, 1994; Schneiter et al, 1999) (Figure 2B). In fact, complex sphingolipids can be transported to the ER at substantial rates, in part for their degradation to ceramide by the phospholipase Lsc1 in the context of a sphingolipid salvage pathway (Matmati & Hannun, 2008).*'

Likewise, the small but significant amounts of cardiolipin (and PG) in pure ER fractions (0.2-0.3% CL in Fig. 2B; 0.5% in Fig. 3B; 0.6% in Fig. 4B) are in agreement with batch-dependent enrichment of mitochondrial markers in the ER fraction vs. cell lysate (Fig S1C).

Following the advice of the reviewer, we have toned down our statements regarding the purity of our preparation.

We believe that the difference in the reported CL lipids between Figure 2B, 3B and 4B are more likely due to differences in the cultivation (cultivation is different for Figure 2B, 3B and 4B). The CL levels are very low in all examples.

Given our arguments regarding the enrichment of the ER relative to mitochondria, we doubt that contaminations of our preparation with mitochondrial membranes alone can be responsible for the variable concentrations of CL observed in our isolated ER membranes. However, given that (often polyspecific) lipid transfer proteins have been identified in virtually every subcellular membrane, we cannot exclude that some minor leakage of inner mitochondrial membrane lipids to the ER can occur.

We mention this in the 'limitations' section of the discussion in the revised manuscript. We state: *'While we cannot rule out the possibility that certain lipids redistribute during cell disruption and organelle isolation, our proteomic and lipidomic data identify organelle-specific compositions that exclude a global mixing of organelle membranes or a broad equilibration of lipids during these procedures.'*

It is recommended to insert an additional Y-axis in Fig S1C (0 to 30-fold) for the non-ER ontology genes (and provide it with a legend that properly explains the presentation of the data).

We have updated the respective figure (**new Appendix Figure S1D**) and had to maintain the Y-axis for optimal readability.

Following the advice of the reviewer, we provide a more elaborate figure legend: *'Mean enrichment (fold change over lysate) of each individual identified protein in our untargeted TMT-labeling proteomics data (Figure 1E, n=3). The bait protein 'Rtn1-bait' via which MemPrep was performed is indicated. Several proteins that were highly enriched in our ER membrane preparation via the Rtn1-bait are uniquely annotated to other organelles. Annotations of subcellular location were retrieved from UniProt (accessed 27.01.2023). A subsequent consultation of the Saccharomyces Genome Database gene description and GOterms (accessed 03.01.2024) reveals that many of those highly enriched proteins annotated to other organelles were previously observed in the ER or the nuclear envelope (Ist2, Pex30, Pex29, Brr6, She10, YBR096W, YEL043W, Hij1, YGR026W), or feature a dual localization to the ER and other organelles (Osm1, Slc1, Svp26, Yur1, Ept1, Pex31, Msc1). 'NA' stands for no annotation for subcellular location.'*

"The near perfect correlation of lipid abundances reported in four independent experiments" (Fig 2D p.15) may turn out less perfect if lipid species are summed per class.

We have summed up lipid species per class and replotted the graph. When grouping by lipid classes instead of lipid species, there is still a remarkable correlation between individual replicates both for high and low abundant lipid classes. We feel that both types of representations are justified. The remarkable reproducibility of our data is demonstrated by both types of representation. Note that the variability of CL lipids is due to their very low abundance, which can make a robust detection and quantification of all CL species challenging (for details see source data related to Figure 2).

5. The interpretation of the inositol depletion data (Fig 3, p.16) ignores the existence of lipid turnover. PI is known to exhibit rapid turnover compared to other lipid classes including the sphingolipids. Therefore the statement "This implies a strict prioritization for sphingolipid biosynthesis over PI synthesis when inositol becomes limiting" is not justified. The inositol depletion-induced increase of PA confirms a vast body of literature data, since it is a key feature of the transcriptional regulation of phospholipid biosynthesis genes by Opi1-Ino2/4. Note that the increase of MIPC vs. M(IP)2C may reflect the reduced PI level (Jesch et al JBC 2010).

Following the advice of the reviewer, we have carefully revised the relevant section of the manuscript. We now state: *'This implies distinct rates of PI and sphingolipid metabolism under this condition.'*

6. The induction of bilayer stress by prolonged exposure to DTT or TM reduces the growth rate (p.17). Please show the corresponding data.

We followed the suggestion of the reviewer by referring to the relevant literature. In fact, the growth curves were reported in a previous, open-access publication (Reinhard *et al.*; PMID: 32850859; Figure 1).

We now state: *'The stressed ER features a lower content of unsaturated membrane lipids (Figure 4C) and exhibits higher levels of storage lipids (EEs and TAGs) (Figure 4D), possibly due to reduced growth rates (Reinhard et al, 2020)'. We also state: 'Lipidome remodeling may be related to the strong growth defect that is induced by both drugs (Reinhard et al, 2020).'*

The response of the ER lipidome, i.e. decreases in PE and PI, increased acyl length and decreased desaturation, is reminiscent of the changes occurring in cell lysates of yeast cells entering stationary phase (co-author Klose et al PLOS One 2012; p.23), where UPR is not activated.

This is an important observation, which we dedicated a lot of attention. There are indeed some similarities in the ER (and whole cell) lipidomes of untreated cells and cells treated with DTT and TM for prolonged periods of time, yet the lipidomes are distinct (see **new Figure 4A,B**; compare untreated versus DTT and TM). We discuss these data in the revised manuscript in a newly written paragraph.

However, the statement that the UPR is not activated during the transition from log phase to stationary phase is not correct (see e.g. PMID: 31484935). We have included an entire new section and new data (**new Figure 4A,B; new Figure EV4A-C**) regarding this point, which includes UPR assays. See '*Lipid bilayer stress caused by proteotoxic agents Dithiothreitol (DTT) and Tunicamycin (TM)*' in the results section of the revised manuscript.

For example, we state in the revised manuscript: '*Instances of acute proteotoxic stress disrupt protein folding in the ER and activate the UPR without causing substantial alterations to cellular lipidomes (Reinhard et al, 2020). However, prolonged proteotoxic stress triggers the UPR through a membrane-based mechanism (Promlek et al, 2011; V  th et al, 2021), with the underlying molecular basis remaining largely unexplored. Even prolonged yeast cultivation without external stressors transiently triggers the UPR around the time of the diauxic shift (Tran et al, 2019a).*'

To what extent do the molecular fingerprints of ER bilayer stress reflect slower growth or growth arrest?

We agree with the reviewer that the reduce growth/growth arrest is a likely contributing factor even though it is challenging to prove this point given the complexity of the lipid metabolic network and the wide pleiotropic effects that DTT and TM. In fact, we have previously speculated on such possibility (see Reinhard et al.; PMID: 32850859): '*We speculate that these marked differences are at least partly due to the growth rate, which is substantially higher for YPD-cultured cells. This interpretation is consistent with the previous observation that storage lipids are dynamically regulated in a growth-stage dependent fashion (Klose et al., 2012; Kohlwein et al., 2013; Casanovas et al., 2015) and suggests an increased flux of fatty acids into membrane lipids in rapidly growing cells.*'

In the revised manuscript we state: '*Thus, lipidome remodeling cannot be attributed to UPR signaling in this context but may be involved in UPR induction (Figure EV4A-C). Lipidome remodeling may be related to the strong growth defect that is induced by both drugs (Reinhard et al, 2020).*'

Did the authors consider UPR triggers other than specific, static ER lipid compositions, e.g. rates of changing lipid composition or desynchronization of protein and lipid synthesis?

We have considered the role of a perturbed protein-to-lipid ratio as a driving force of the membrane-based UPR. In fact, we have published the hypothesis that UPR transducers may act as sensors for the protein-to-lipid ratio (Covino and Ernst; PMID: 30075144). In fact, the development of the MemPrep technology represents an important milestone in our efforts to

learn how cells sense and control the protein-to-lipid ratio in cellular membranes. We refer to this point by stating in the discussion: '*Therefore, it comes as no surprise that the compressibility and thickness of the ER membrane is continuously monitored by the UPR for regulating the relative rate of protein and lipid biosynthesis (Halbleib et al, 2017; Schuck et al, 2009; Covino et al, 2018).*'

We did not consider a role of the rate of changing lipid compositions, because these rates are several orders of magnitude slower the protein-lipid interactions and protein-protein that control the activation of Ire1 by lipid bilayer stress.

The proposed (un-)stressed ER-like lipid compositions may be preliminary.

We do not understand this point. However, we have made major changes to the section '*Lipid bilayer stress caused by proteotoxic agents Dithiothreitol (DTT) and Tunicamycin (TM)*', which we believe have substantially improved the manuscript.

7. To demonstrate the purity of the immune-purified vacuolar fraction in the absence of protein MS data, a blot analyzing a fixed amount of total protein per fraction (as in Fig 1B) is more revealing and informative than the blot showing 0.2% of each fraction (Fig 6A).

For the revised manuscript, we added new, quantitative proteomics data of the isolated vacuole membranes (**new Figure 6B**), which -in our view- provides a more comprehensive view on the performance of the MemPrep procedure in enriching for vacuole membranes.

8. The methods section and legends need extensive proofreading to make sure that experimental details and sources of materials are correct and complete. Several methods are currently missing, including the construction of the tag, protein concentration assay, gel electrophoresis and immune-blotting, quantitation of blots, K-means clustering, as well as source and genotype of strains and plasmids used.

Following the suggestion of the reviewer, we have proofread the relevant sections and added information wherever necessary.

'Manual generation of bait-tag strains

DNA sequences of HRV 3C site-GSG and myc-GSG were introduced via primers in two consecutive steps between 6xGLY and 3xFLAG encoding regions on plasmid pFA6a-6xGLY-3xFLAG-kanMX6 (Funakoshi & Hochstrasser, 2009) (Addgene plasmid #20754) using Gibson assembly to yield pRE866. The DNA sequence of the bait-tag cassette (Appendix Supplementary Methods) was amplified by PCR from plasmid pRE866 using primers RE1012/RE1013, RE1018/RE1019, RE1014/RE1015 for tagging Rtn1, Elo3, and Vph1, respectively. Primers contain homologous regions for C-terminal tagging at the endogenous loci (RTN1, ELO3, VPH1). PCR products were used to transform wildtype yeast strain BY4741 by the lithium-acetate method (Ito et al, 1983).'

'Protein concentration determination, SDS-PAGE, and immunoblotting

Protein concentrations of all fractions from the immuno-isolation were determined using the micro BCA protein assay (Thermo Fisher Scientific #23235) following the manufacturers recommendations. For SDS-PAGE, 1 volume membrane sample buffer (8 M urea, 0.1 M Tris-HCl, pH 6.8, 5 mM EDTA, 3.2% (w/v) SDS, 0.05% (w/v) bromphenol blue, 4% [v/v] glycerol, and 4% (v/v) β -mercaptoethanol) was mixed with 2 volumes of the immuno-isolation fraction, incubated at 60 °C for 10 min, and loaded onto 4-15% mini-PROTEAN TGX precast protein gels (Bio-Rad). Proteins were separated at 185 V for 35 min. After separation, proteins were transferred by semi-dry Western blotting onto nitrocellulose membranes (Amersham Protran Premium 0.45 μ m). The proteins of interest were detected using specific primary antibodies and fluorescent secondary antibodies (see Reagents and Tools Table) on

a fluorescence imager (LI-COR, Odyssey DLx). Signal intensities on immunoblots were quantified using ImageStudioLite.'

'Enrichment calculation based on untargeted proteomics and clustering

IsobarQuant output data were analyzed on a gene symbol level in R (<https://www.R-project.org>) using in-house data analysis pipelines. In brief, data were filtered to remove contaminants and proteins with less than 2 unique quantified peptide matches. Subsequently, log₂ transformed protein reporter ion intensities ('signal sum' columns) were first cleaned for batch effects using the 'removeBatchEffects' function from the limma package (Ritchie et al, 2015) and further normalized within the TMT set using the vsn package (Huber et al, 2002). Proteins were tested for differential expression using the limma package. The replicate information was added as a factor in the design matrix given as an argument to the 'lmFit' function of limma. A protein was annotated as a hit with a false discovery rate (fdr) smaller 5 % and a fold-change of at least 100 % and as a candidate with a fdr below 20 % and a fold-change of at least 50 %. Proteins classified as hits and candidates as well as a positive fold-changes of the "pre" condition were clustered into 9 clusters (method kmeans) based on the Euclidean distance between normalized TMT reporter ion intensities divided by the median of the "pre-stress" condition. UniProt annotations of subcellular location were retrieved from UniProt (accessed 27.01.2023).'

Source and genotype of strains and plasmids are provided in the reagent table file.

Other remarks

a. Provide legends for Tables S3, 4, 5 on a separate Excel sheet, detailing the experimental conditions (S3) and the parameters shown (S4, 5).

We have carefully assembled all relevant source data and believe that a clear connection between the materials and methods section and the source data can be made.

b. Does ER-stress affect purity of the ER fractions obtained (e.g. due to an increase of MCS)? TableS5 includes a.o. contaminating mitochondrial proteins, which may provide insight.

In the revised manuscript, we address the purity of our immunoisolates of the stressed ER experimentally (**new Figure EV5A-C**), discuss the observations, and refer to relevant literature.

We state: *'Membrane contact sites are remodeled during ER stress in both yeast and mammalian cells (Vevea et al, 2015; Liu et al, 2017; Kwak et al, 2020; Liao et al, 2022). To exclude that any of the observed proteomic changes are due to increased contaminations with other organelles, we first assessed the impact of DTT and TM on the quality of our ER preparations by immunoblotting. Expectedly, the GPI-anchored cell wall protein Gas1 accumulates as a ~105 kDa precursor in the ER of TM-stressed cells, but not in the ER or DTT-stressed cells (Figure EV5A) (Fankhauser & Conzelmann, 1991; Wang et al, 2023). The purity of the ER isolations was assessed by determining the enrichment of marker proteins for the ER, the vacuole, mitochondria, and endosomes (Figure EV5A-C). While the Rtn1-bait and the ER membrane protein Dpm1 were several-fold enriched in the immuno-isolation step starting from microsomes, the markers for the vacuole (Vph1), mitochondria (Por1), and endosomes (Pep12) were depleted relative to the microsomal P100 fraction (Figure EV5A-C). Hence, MemPrep allows for the isolation of both the unstressed and the stressed ER.'*

c. The clarity of the description of the untargeted protein MS procedures on p.8-9 could be improved by referring to the two datasets produced (Table S4 and S5).

Thank you. In the revised manuscript, we refer to the relevant source data.

d. Tables S4 and S5, and Fig S2D-F are not referred to in the Results section.

Thank you. In the revised manuscript, we refer in the results section to the relevant source data and data in Appendix Figures.

e. Explain the membrane topology of (tagged) Rtn1.

Following the reviewer's advice, we describe the membrane topology of Rtn1 in the revised manuscript. We state: '*Rtn1 is a small (~33 kDa) and highly abundant reticulon protein (~37,100 copies per cell), which stabilizes membrane curvature in the tubular ER (Ghaemmaghami et al, 2003; Voeltz et al, 2006). Rtn1 has four predicted transmembrane helices with both N- and C-terminus facing the cytosol and a C-terminal amphipathic helix, which inserts into the cytosolic leaflet of the ER membrane to generate a high spontaneous membrane curvature (De Craene et al, 2006; Hu et al, 2008).*'

f. P.12 states "small organelle fragments are less likely to form contacts ..." What is this based on? One could argue that increasing the number of fragments promotes chances of interaction.

We rephrased the relevant section of the manuscript to avoid confusions: '*Secondly, after a differential centrifugation to obtain a crude microsome fraction, organellar fragments were disrupted by sonication. This is because a major fraction of the ER membrane surface forms physical contacts to other organelles (Phillips & Voeltz, 2016), which would hamper the subsequent ER isolation. We reasoned that large ER fragments are more likely to contain such contact sites, while enough small ER fragments might facilitate higher purity if membrane mixing can be avoided.*'

g. In Fig 1C, the eluate also seems to contain GFP-negative vesicles. What is the conclusion regarding loss of luminal proteins (Kar2 is also lost in Fig 1D)? What is the size of the scale bar? Does it apply to all 6 panels?

We thank the reviewer for spotting this. The scale bar applies to all 6 panels and corresponds to 5 μm . We have added this information to the legend of Figure 1C.

h. Lipid vesicles obtained by extrusion using 0.2 μm filters are denoted as large unilamellar vesicles (LUV) (p.13).

The reviewer is correct. We realized that we made a mistake in the materials and methods section. We extruded liposomes with consecutively with 0.4 μm filters, 0.2 μm filters, and 0.05 μm filters. In the first version of the manuscript the information was incomplete and hence incorrect. We have corrected this point in the revised manuscript.

i. Include panels depicting average acyl chain length or length distribution in Figs 2, 3, S2, S3, S6.

While we have decided against including representation of these lipid data to the individual main Figures of the manuscript, we now provide extensive information regarding the lipid acyl chains in the **new Appendix Figure S5**. All relevant data are represented here are available with the source source data.

j. Cki1 phosphorylates choline at the expense of ATP (Fig S3D).

Thank you. We have added this information in the revised manuscript. It is now depicted in the **revised Appendix Figure S3D**.

k. Lipidome of whole cell lysate in Fig 6 is identical to that in Fig 2.

The reviewer is correct. We are grateful that this point was spotted. We clarified this point for the revised manuscript for Figure 6 and all other relevant Figures.

Was 1 culture used for isolating both ER and vacuole? If so, please indicate.

No. We used separate cultures using different bait strains for isolating ER and vacuole membranes.

Prof. Robert Ernst
Saarland University
Medical Biochemistry and Molecular Biology
Kirrberger Str. 100
Building 61.4
Homburg, Saarland 66421
Germany

9th Feb 2024

Re: EMBOJ-2022-112605R1

A new technology for isolating organellar membranes provides fingerprints of lipid bilayer stress

Dear Robert,

Thank you again for submitting your revised manuscript to The EMBO Journal. I sent it back to all three original referees, and I am happy to say that all of them found the study significantly improved, and are now in principle supportive of publication. However, referee 3 still has one important concern regarding a newly added experiment, which would seem important to clarify prior to acceptance. I would therefore appreciate if you could get back to me with some thoughts on these issues, happily via Zoom call in the first half of next week, so we could discuss how best to address them.

In addition, there are several other editorial and presentational points to incorporate at this stage:

- Please incorporate the various minor points listed by referee 3.
- On the abstract page of the manuscript, please include 4-5 general keyword terms to enhance searchability.
- Please include a dedicated "Data Availability" section at the end of the Material and Methods (suggested wording: "The [structural coordinates | microarray | mass spectrometry] data from this publication have been deposited to the [name of the database] database [URL] and assigned the identifier [accession | permalink | hashtag]."). Should this not apply, this should still be stated as "This study includes no data deposited in external repositories."
- Please adjust the header and the format of the Disclosure and competing interests statement (next to the Acknowledgment section) as specified in our Guide to Authors - for details, see <https://www.embopress.org/competing-interests>
- As we are switching from a free-text author contribution statement towards a more formal statement based on Contributor Role Taxonomy (CRediT) terms, please remove the present Author Contribution section and instead specify each author's contribution(s) directly in the Author Information page of our submission system during upload of the final manuscript. See <https://casrai.org/credit/> for more information.
- Please reorder the manuscript sections in the following order: Title page - Abstract & Keywords - Introduction - Results - Discussion - Materials & Methods - Data Availability - Acknowledgments - Disclosure & Competing Interests Statement - References - Figure Legends - Tables with legends - Expanded View Figure Legends.
- Please carefully go through the reference list to check its completeness and adherence to EMBO Journal format guidelines; I noted that for several citations, volume and/or page/eLocator numbers are still missing. For articles with more than 10 authors, only the first 10 should be listed, followed by "et al.". Finally, while most preprints have been referenced in the correct style in the text and the reference list, some (esp. Wang et al) have been not yet been completed.
- Please double-check that the correct email has been entered for co-author John Reinhard, as our acknowledgement email could not be delivered to John.Reinhard@uks.eu
- Please compile all Appendix Figures in a single APPENDIX PDF, headed by a brief Table Of Contents with page numbers of the included items. Please also move the Appendix Figure Legends from the main text into the Appendix, each of them below the respective figure. Finally, please also move the brief Appendix Supplementary Methods from the main text into the Appendix
- During our routine pre-acceptance checks, our data editors have raised the following queries regarding figures, data, and legends. In order to facilitate our checking, please correct them in the final manuscript file with "Track Changes" option activated:
 1. Please note that a separate 'Data Information' section is required in the legends of figures 2a-c; 3a-c, e; 4c-e; 6c-d.
 2. Please note that the figure EV 2a-c does not contain any statistical parameter, kindly rectify the statistics test and p value

related information in the figure legend appropriately.

3. Please indicate the statistical test used for data analysis in the legends of figures 5a-c; EV 1b-e; EV 5e.

4. Please note that in figures 3b-c; 6c-d; EV 1a; there is a mismatch between the annotated p values in the figure legend and the annotated p values in the figure file that should be corrected.

5. Please note that information related to n is missing in the legends of figures 2c; 3a-c, e; 4c-e; 5a-b; 6c-d; EV 1b-e; EV 2a-c; EV 3a-c; EV 4a-d.

6. Please note that the error bars are not defined in the legend of figure EV 5b.

7. Please note that the measure of center for the error bars needs to be defined in the legends of figures 2a-c; 3b-c, e; 4c-e; 6c-d; EV 2a-c; EV 3a-c; EV 4a-d; EV 5c."

- Finally, please provide suggestions for a short 'blurb' text prefacing and summing up the conceptual aspect of the study in two sentences (max. 250 characters), followed by 3-5 one-sentence 'bullet points' with brief factual statements of key results of the paper; they will form the basis of an editor-written 'Synopsis' accompanying the online version of the article. Please also upload a synopsis image, which can be used as a "visual title" for the synopsis section of your paper. The image (maybe a simplified version of Figure 1A?) should be in PNG or JPG format, and please make sure that it remains in the modest dimensions of (exactly) 550 pixels wide and 300-600 pixels high.

I am therefore returning the manuscript to you for a final round of minor revision, to allow you to make these adjustments and upload all modified files. Once we will have received them and referee 3's remaining criticism has been clarified, we should be ready to swiftly proceed with formal acceptance and production of the manuscript.

With kind regards,

Hartmut

9) Digital image enhancement is acceptable practice, as long as it accurately represents the original data and conforms to community standards. If a figure has been subjected to significant electronic manipulation, this must be clearly noted in the figure legend and/or the 'Materials and Methods' section. The editors reserve the right to request original versions of figures and the original images that were used to assemble the figure. Finally, we generally encourage uploading of numerical as well as gel/blot image source data; for details see: embopress.org/page/journal/14602075/authorguide#sourcedata

At EMBO Press, we ask authors to provide source data for the main manuscript figures. Our source data coordinator will contact you to discuss which figure panels we would need source data for and will also provide you with helpful tips on how to upload and organize the files.

In the interest of ensuring the conceptual advance provided by the work, we recommend submitting a revision within 3 months (9th May 2024). Please discuss the revision progress ahead of this time with the editor if you require more time to complete the revisions. Use the link below to submit your revision:

Link Not Available

Referee #1:

The authors have addressed substantial concerns raised by the reviewers. Importantly, this study provides compelling data that emphasizes an unprecedented enrichment of ER that is useful for many downstream applications. The revised manuscript incorporates additional data that robustly supports the overarching conclusions. In sum, Memprep has garnered attention from vast laboratories in membrane and organelle biology, making a strong case for the consideration of this manuscript for publication.

Referee #2:

This is a revised manuscript that reports a novel MemPrep method to isolate enriched ER organelle. The method is used to evaluate several regulatory mechanisms as a proof of concept the value to the method. Moreover, through these studies, important new information has been obtained to advance the field. The original submission was reviewed by three persons who felt the work was important, but identified concerns. By and large, the authors have addressed all concerns with new data and/or extended discussion. In particular, the authors have addressed all limitations. I have no further concerns.

Referee #3:

The revision including the new data and control experiments has considerably improved the manuscript. My previous comments 1-5 and 7-8 have been (more than) satisfactorily dealt with in the revised manuscript. The new data included in response to comment 6 raise a new concern.

The new data presented for the untreated control in Fig. 4 and EV4 may explain part of the DTT- and TM-induced lipid fingerprints, which has been overlooked by the authors.

As stated on p. 22, prolonged cultivation for 4h shows the hallmarks of cells running out of inositol (PI-level drops, UPR is induced, CDP-DAG and PA increase). During the 4h of prolonged cultivation, the culture's OD600 increased from 0.8 to 3.5 (Reinhard 2020, Fig 1F). Surprisingly, the authors do not consider the possibility that the lipidome's response to 4h of DTT or TM treatment, may be confounded by inositol becoming limiting as OD values increase from 0.8 to 2 (Reinhard 2020). The

decreased levels of PI and LPI, and increased levels of CDP-DAG and PA (Fig. 4D,E) strongly point in that direction. If true, i.e. if the observed response to prolonged proteotoxic stresses includes a contribution from (early stage) inositol depletion, the interpretation of the dataset and the follow-up experiments should be reconsidered. Alternatively, the effects of DTT and TM could be tested with excess inositol present.

Follow-up on previous points 2-4:

(2) Do note, that based on the numbers provided, protein yields of MemPrep are an order of magnitude lower than stated in the authors' response (manuscript p. 17 and 18):

$(31/[0.2 \times 83,000]) \times 100\% = 0.19\%$,

inferring that an estimated 99.8% of ER was lost during the procedure.

(3) p. 16, "We sonicated these synthetic liposomes in the presence of a ~15.4-fold excess of microsomal membranes (P100)." For clarity, please add "based on membrane phospholipid content".

(4) Although the authors have built a strong case that sphingolipids are genuine ER membrane lipids, at the end of the day the data do not exclude that (part of) the sphingolipids recovered in the pure ER isolate derive from cross-contamination with other organelles, however small.

Small remarks a-k have been resolved. With regard to h, please note that the text on p.8 does not match with the response letter.

Response to reviewers
Manuscript EMBOJ-2022-112605R2

We thank all three reviewers for their time and efforts that helped us tremendously improving the manuscript.

Referee #1:

The authors have addressed substantial concerns raised by the reviewers. Importantly, this study provides compelling data that emphasizes an unprecedented enrichment of ER that is useful for many downstream applications. The revised manuscript incorporates additional data that robustly supports the overarching conclusions. In sum, Memprep has garnered attention from vast laboratories in membrane and organelle biology, making a strong case for the consideration of this manuscript for publication.

We thank reviewer 1 for the invested time and efforts.

Referee #2:

This is a revised manuscript that reports a novel MemPrep method to isolate enriched ER organelle. The method is used to evaluate several regulatory mechanisms as a proof of concept the value to the method. Moreover, through these studies, important new information has been obtained to advance the field. The original submission was reviewed by three persons who felt the work was important, but identified concerns. By and large, the authors have addressed all concerns with new data and/or extended discussion. In particular, the authors have addressed all limitations. I have no further concerns.

We thank reviewer 2 for the invested time and efforts.

Referee #3:

The revision including the new data and control experiments has considerably improved the manuscript. My previous comments 1-5 and 7-8 have been (more than) satisfactorily dealt with in the revised manuscript. The new data included in response to comment 6 raise a new concern.

The new data presented for the untreated control in Fig. 4 and EV4 may explain part of the DTT- and TM-induced lipid fingerprints, which has been overlooked by the authors. As stated on p. 22, prolonged cultivation for 4h shows the hallmarks of cells running out of inositol (PI-level drops, UPR is induced, CDP-DAG and PA increase). During the 4h of prolonged cultivation, the culture's OD600 increased from 0.8 to 3.5 (Reinhard 2020, Fig 1F). Surprisingly, the authors do not consider the possibility that the lipidome's response to 4h of DTT or TM treatment, may be confounded by inositol becoming limiting as OD values increase from 0.8 to 2 (Reinhard 2020). The decreased levels of PI and LPI, and increased levels of CDP-DAG and PA (Fig. 4D,E) strongly point in that direction. If true, i.e. if the observed response to prolonged proteotoxic stresses includes a contribution from (early stage) inositol depletion, the interpretation of the dataset and the follow-up experiments should be reconsidered. Alternatively, the effects of DTT and TM could be tested with excess inositol present.

We thank reviewer 3 for the invested time and efforts.

We also thank the reviewer 3 for highlighting this important, and (as we believe) very valid point. We fully agree with the reviewer's assessment that inositol-depletion (and/or) an overall reduction of anionic lipids causing a decreased negative surface charge density may contribute to UPR activation under DTT or TM treatments.

We had not overlooked this point, but probably this has not become sufficiently clear in the previous version of the manuscript especially unstressed and stressed conditions have been discussed separately.

We wrote:

'Prolonged cultivation in the absence of supplemented proteotoxic agents also resulted in significant remodeling of the cellular lipidome and much lower EE and PI levels compared to DTT- or TM-treated cells (Figure 4A; untreated versus TM or DTT). Because standard SCD medium contains only 11 μ M inositol and because the BY4741 is particularly dependent on inositol for normal growth (Hanscho et al, 2012) it is possible that prolonged cultivation of this strain leads to a 'natural' inositol depletion. Overall, our observations suggests that DTT or TM treatments induce a complex and characteristic remodeling of the cellular lipidome, which is distinct from that of prolonged cultivation. This conclusion is corroborated by a principal component analysis of whole cell lipidomic data showing distinct clustering of the data derived from pre-stressed, untreated, and the DTT- or TM-stressed cells (Figure 4B).'

To address the possibility that inositol-depletion may contribute to the membrane-based stress during prolonged DTT and TM treatments, we now state in the revised manuscript:

*'Prolonged cultivation in the absence of supplemented proteotoxic agents also resulted in significant remodeling of the cellular lipidome and much lower EE and PI levels compared to DTT- or TM-treated cells (Figure 4A; untreated versus TM or DTT). Because standard SCD medium contains only 11 μ M inositol and because the BY4741 is particularly dependent on inositol for normal growth (Hanscho et al, 2012) it is possible that prolonged cultivation of this strain leads to a 'natural' inositol depletion. **The increased abundances of precursors of PI biosynthesis, PA and CDP-DAG, also point in this direction.** While it is likely that the mildly reduced levels of PI contribute to the membrane-based ER stress upon prolonged DTT or TM treatments, our data suggest **a more complex remodeling** of the cellular lipidome, **which is distinct from the effects of inositol depletion and prolonged cultivation.** This conclusion is corroborated by a principal component analysis of whole cell lipidomic data showing distinct clustering of the data derived from pre-stressed, untreated, inositol-depleted, and the DTT- or TM-stressed cells (Figure 4B).'*

We also state in the revised manuscript with respect to the composition of the DTT or TM stressed ER: *'The stressed ER features a lower content of unsaturated membrane lipids (Figure 4C) and exhibits higher levels of storage lipids (EEs and TAGs) (Figure 4D), possibly due to reduced growth rates (Reinhard et al, 2020) and increased fatty acid flux into storage lipids, **or a gradual depletion of lipid metabolites such as inositol from the medium** (Listenberger et al, 2003; Vevea et al, 2015; Henne et al, 2018; Reinhard et al, 2020). The elevated levels of lipid metabolic intermediates CDP-DAG and DAG in the stressed ER supports both possibilities (Figure 4D, E). **In fact, inositol-depletion is a possible contributor to the membrane-based activation of the UPR under this condition.'***

With these changes, we feel that the consideration the possibility that inositol-depletion contributes to lipid bilayer stress during prolonged DTT or TM treatments. Of note, we are currently preparing another manuscript in which we reconstitute the membrane-based sensing mechanism of Ire1 *in vitro* for systematically characterizing the impact of anionic lipids on Ire1 dimerization/oligomerization.

Follow-up on previous points 2-4:

(2) Do note, that based on the numbers provided, protein yields of MemPrep are an order of magnitude lower than stated in the authors' response (manuscript p. 17 and 18):

$(31/[0.2*83,000])*100\% = 0.19\%$,

The reviewer is correct. We adjusted the relevant sections in the manuscript. We now state: *'Assuming the ER accounts for 20% of the total cell protein (Zinser & Daum, 1995), we estimate*

that >99.8% of ER protein is lost during the isolation.'. We also state: '*Quantitative proteomics reveals that fusing the bait tag to either Rtn1 or Elo3 has no impact on the overall cellular proteome (Figure EV1B) and that ER proteins can be enriched by MemPrep using either of the two baits (Figure EV1C, D) with an estimated yield of 0.19% and 0.08% of the input material for the Rtn1- and Elo3-bait, respectively.*'

inferring that an estimated 99.8% of ER was lost during the procedure.

(3) p. 16, "We sonicated these synthetic liposomes in the presence of a ~15.4-fold excess of microsomal membranes (P100)." For clarity, please add "based on membrane phospholipid content".

Following the reviewer's advice, we now state in the revised manuscript: '*We sonicated these synthetic liposomes in the presence of a ~15.4-fold excess of microsomal membranes (P100) based on membrane phospholipid content.*'

(4) Although the authors have built a strong case that sphingolipids are genuine ER membrane lipids, at the end of the day the data do not exclude that (part of) the sphingolipids recovered in the pure ER isolate derive from cross-contamination with other organelles, however small.

In response to the reviewer's point, we now state in the revised manuscript: '*While we cannot rule out that a minor fraction of these lipids may originate from contaminating organelles, these data suggest a significant retrograde transport of complex sphingolipids from the Golgi apparatus to the ER, likely via COPI vesicles together with ER-resident proteins bound to the HDEL-receptor (Aguilera-Romero et al, 2008).*'

Small remarks a-k have been resolved. With regard to h, please note that the text on p.8 does not match with the response letter.

The reviewer is correct. With respect to the point h, the text in the manuscript '*Multilamellar liposomes were extruded through 0.4 μm and then 0.1 μm filters with each 21 strokes to yield large unilamellar vesicles (LUVs).*' is (and was) correct.

The text in the previous response to the reviewer, was an outdated, incorrect answer from an earlier round of control experiments during the optimization of the FRET assay. While the text in the manuscript was updated, the answer to the reviewer was not. We apologize for the irritation.

Prof. Robert Ernst
Saarland University
Medical Biochemistry and Molecular Biology
Kirrberger Str. 100
Building 61.4
Homburg, Saarland 66421
Germany

26th Feb 2024

Re: EMBOJ-2022-112605R2
MemPrep, a new technology for isolating organelar membranes provides fingerprints of lipid bilayer stress

Dear Prof. Ernst,

Thank you for submitting your final revised manuscript for our consideration. I am pleased to inform you that we have now accepted it for publication in The EMBO Journal.

Yours sincerely,

Hartmut Vodermaier
